# Soil moisture-atmosphere coupling strength over Central Europe in the recent warming climate

Thomas Schwitalla[1], Lisa Jach[1], Volker Wulfmeyer[1], Kirsten Warrach-Sagi[1]

[1]Institute of Physics and Meteorology, University of Hohenheim, Garbenstrasse 30, 70599 Stuttgart, Germany

*Correspondence to*: Thomas Schwitalla (Thomas.Schwitalla@uni-hohenheim.de)

**Abstract**

In the last decades, Europe has experienced increasing periods of severe drought and heatwaves which have a major impact on agriculture and society. While soil moisture was found to be a crucial factor for enhancing the

duration and intensity of these events, their influence is typically quantified for climate periods or single events. To provide an overview of how surface conditions shape land-atmosphere (LA) coupling, this study evaluates the interannual variability of LA coupling strength for selected warm summer seasons between 1991-2022 over Central Europe by means of ERA5 data.

Especially the drought summer seasons 2003, 2018, and 2022 were particularly distinctive in respect to the

changing soil moisture-atmosphere coupling pattern which in turn leads to an increased lifted condensation level height thereby inhibiting local deep convection triggering. Summer 2021 was a special case as spring precipitation was consistent with the climatological average and a heavy rain event occurred during July, resulting in high moisture availability and a change in the LA coupling strength. The results obtained with respect to LA coupling strength reflect a shift in the coupling relationships toward reinforced heating and drying by the land surface under

heatwave and drought conditions, whose frequency is increasing with ongoing climate change.

## 1 Introduction

In the last decades, Europe experienced severe drought periods and heatwaves (WMO, 2015; C3S, 2018; Markonis et al., 2021; WMO, 2022a) with 2022 being the hottest summer ever recorded over Europe (WMO, 2022a). Precipitation exhibited a strong dry anomaly during summer over Central Europe in 2003, 2018 and 2022 (WMO,

2004, 2018; C3S, 2018; WMO, 2022b; Spensberger et al., 2020). At the same time, the soil experienced an exceptional dryness in the uppermost 25 cm (Boeing et al., 2022; Rakovec et al., 2022) as shown by the soil moisture index developed by Zink et al. (2016). This was also shown by Rousi et al. (2023) and Dirmeyer et al. (2021) for 2018, who suggest that these extreme conditions will be more likely under climate change conditions during 2020-2049 where two out of three summer seasons will experience hot and dry conditions in a +1.5°C

warmer world which is already the case. Rousi et al. (2022) identified Europe as a heatwave hot spot where heatwaves are three to four times more likely than in other areas of the midlatitudes due to the occurrence of a double-jet stream configuration associated with atmospheric blocking conditions (Kornhuber et al., 2017).Land-atmosphere (LA) coupling generally describes the co-variability of atmospheric conditions (e.g., planetary boundary layer (PBL) height, convective available potential energy (CAPE), lifted condensation level (LCL) and

the characteristics of the land surface (e.g., vegetation, soil moisture) (Findell and Eltahir, 2003b; Koster et al., 2004; Dirmeyer, 2011; Guo et al., 2006). In the context of extremes, LA coupling was identified as a driver and intensifier for the duration and intensity of heat waves and droughts (van Heerwaarden and Teuling, 2014; Ukkola

et al., 2018; Schumacher et al., 2022). Miralles et al. (2019) and Schumacher et al. (2022) showed the existence of a self-propagating mechanism of droughts. Meteorological droughts intensify due to increased water vapor deficit (VPD) inside the PBL which feeds back into an intensified depletion of surface moisture reservoirs.

One of these reservoirs is soil moisture, which plays a key role for the climate due to its influence on the partitioning between surface sensible and latent heat fluxes of the incoming solar energy (Seneviratne et al., 2010; Stephens et al., 2023). In vegetated areas the surface latent heat flux additionally depends on the atmospheric water vapor deficit (VPD), air temperature, incoming radiation, and vegetation properties (stomatal resistance, leaf area index (LAI) and rooting depth) (Miralles et al., 2019; Warrach-Sagi et al., 2022). In consequence of spatial and temporal variability in these influencing factors, LA coupling often shows regional, but also temporal variations, especially under climate change conditions (Seneviratne et al., 2006; Denissen et al., 2022; Jach et al., 2022).

Knist et al. (2017) investigated the long-term average relationship between root zone soil moisture and surface fluxes by means of different regional climate model (RCM) simulations for the period 1989-2008 for the European summer seasons. They identified a coupling hot spot region for the surface coupling of sensible and latent heat fluxes and latent heat flux and 2m temperature in South Europe while a transition zone is present over larger parts of Central Europe. Jach et al. (2022) performed a RCM LA coupling sensitivity experiment with respect to climate change signals of temperature and humidity for the period 1986-2015. Their results revealed a permanent coupling hot spot over Northeast and East Europe with the location being insensitive to changes in low level moisture and temperature. While there was only little sensitivity over the northern part of this area, Central Europe and the British Isles showed a change in the coupling regime based on the convective triggering potential and low-level humidity index (CTP-HI$_{low}$) framework (Findell and Eltahir, 2003a, 2003b). The combination of CTP and HI$_{low}$ allows for a determination whether convection is likely to occur (see Fig. 15 of Findell and Eltahir, 2003a). Jach et al. (2022) performed climate change sensitivity tests using the CTP-HI$_{low}$ framework. They found that Central Europe is in a transition zone where the development of convection is more likely to be solely controlled by a temperature increase.

Warrach-Sagi et al. (2022) evaluated the atmospheric coupling index (ACI; Guo et al., 2006; Dirmeyer, 2011) using an RCM and found a strong sensitivity between sensible heat flux and CAPE during the growing season 2005 over South Germany while Leutwyler et al. (2021) found a strong soil moisture – precipitation feedback over Central Europe during the summer seasons 1999-2008.

Several studies investigated the relation of soil moisture with recent European heat waves and droughts. Hauser et al. (2016) found that a soil moisture-temperature feedback was a key driver for the severe heat wave over Siberia in 2010, while Dirmeyer et al. (2021) and Orth (2021) found that it was a key driver for the European heatwave in 2018. García-Herrera et al. (2010) found that a strong soil moisture deficit was also one of the key drivers for the 2003 European heat wave. The study of Miralles et al. (2014) suggests that the heatwaves over Europe in 2003 and over Russia in 2010 were enhanced by a persistent large scale weather pattern associated with a strong soil moisture decay. The analysis of Dirmeyer et al. (2021) for the 2018 European heatwave revealed enhanced soil moisture – near-surface coupling under drought conditions. The exceptionally low soil moisture limited evapotranspiration and thus amplified the heat wave due to reduced evaporative cooling (Santanello et al., 2018). This led to one of the most severe heatwaves over Europe since 1979 (Becker et al., 2022). Wehrli et al. (2019) found that soil moisture and the large scale weather pattern are equally important for the duration and intensity of heatwaves around the globe. According to Ossó et al. (2022), Europe already faced an increase in climate extremes

since 2000 and will remain a hot spot for severe droughts (Huebener et al., 2017; van der Wiel et al., 2022) impacting not only summer's crop yields (Toreti et al., 2022) but also affecting the generation of renewable energy. Shifts in the hydrological conditions from energy- to moisture-limited conditions originating from droughts and heatwaves (Dirmeyer et al., 2021; Duan et al., 2020) or severe flooding (Lo et al., 2021) imply temporal variability in LA coupling at sub-seasonal to interannual time scales. Guo and Dirmeyer (2013)  also found interannual variability in soil moisture-precipitation coupling in consequence of diverging soil moisture availability. Additionally, the critical soil moisture thresholds (Dirmeyer et al., 2021; Rousi et al., 2023) suggest not only an intensification of the heat and drought conditions by LA coupling over Europe but also a strengthening of the coupling itself.

However, a quantification of the temporal variability in different coupling relationships and the associated impacts of the variability still lack, as LA coupling strength on other time scales than climate periods has been barely investigated over Central Europe so far. The same applies to shifts between coupling regimes due to variability in the climatic conditions.

In this study, we therefore assess the variability of LA coupling of selected European summer seasons 1991-2022 in dependence on temperature, soil moisture, precipitation and large-scale weather pattern by applying data from the fifth generation European Centre for Medium Range Weather Forecasting (ECMWF) atmospheric reanalysis (ERA5; Hersbach et al., 2020).

The paper is structured as follows: Section 2 describes the applied data sets and coupling indices. Section 3 describes the interannual variability of meteorological variables, the meteorological situation of the summer seasons chosen for evaluation followed by the LA coupling analysis. Section 4 discusses our results while section 5 summarizes our work and provides an outlook on potential future research.

## 2 Material and Methods

### 2.1 Datasets

For the analysis of the LA coupling, ERA5 was used. ERA5 is produced by the Copernicus Climate Change Service (C3S, http://climate.copernicus.eu/) at ECMWF. This data set provides hourly estimates of atmospheric, surface, and oceanic variables on a horizontal resolution of 0.25°. ERA5 clearly outperforms its predecessor ERA-Interim (Dee et al., 2011; Martens et al., 2020) and makes use of sophisticated atmospheric data assimilation including satellite derived soil moisture data (Albergel et al., 2012) to its land-surface model (LSM) HTESSEL (Balsamo et al., 2009).

ERA5 has been recently successfully applied in LA feedback studies over Europe (Rousi et al., 2023; Rousi et al., 2022) and other regions (Sun et al., 2021; Qi et al., 2023). Other reanalysis data sets like the Uncertainties in Ensembles of Regional ReAnalysis (UERRA), only available until 2019, are not recommended to use if surface fluxes are required for analysis (https://confluence.ecmwf.int/display/UER/Issues+with+data). The Consortium for Small-scale Modeling (COSMO) REA6 (Bollmeyer et al., 2015) data set is only available between 1995-2019 and does neither make use of a sophisticated data assimilation scheme nor of an ensemble approach. The Climate Forecast System Reanalysis (CFSR; Schneider et al., 2013) is only available until 2010 and thus does not cover the recent climate change period. Although a study of Beck et al. (2021) revealed that ERA5-Land (Muñoz-Sabater et al., 2021) outperformed ERA5 with respect to in-situ soil moisture measurements in the Carpathians and Southeast France during 2015-2019, data sets developed solely for land surface studies like ERA5-land and the

Global Land Evaporation Amsterdam Model (GLEAM; Miralles et al., 2011) lack atmospheric boundary layer variables required for studying land-atmosphere coupling and therefore were not considered in this study to avoid mixing different models for the investigation of the coupling chain.

To categorize the summer seasons during 1991-2022, seasonal mean anomalies of 2m temperatures and precipitation from ERA5 as well as precipitation from the ENSEMBLES daily gridded observational dataset for precipitation (E-OBS; Cornes et al., 2018) version V26.0e were calculated.

## 2.2 LA coupling indices

In our study we apply a subset of the statistical LA coupling metrics framework, namely the terrestrial coupling index (TCI) and atmospheric coupling index (ACI) (Guo et al., 2006; Dirmeyer, 2011; Santanello et al., 2018)., Additionally, the correlation between surface sensible heat flux (SH) and surface latent heat flux (LH) is calculated. To derive the different indices, we used a combination of the NCAR Command Language (NCL, Brown et al., 2012) together with the FORTRAN programs provided by Tawfik (2015).

For our analysis, we used volumetric root zone soil moisture $\eta$, defined as weighted sum of the soil moisture in the top three soil layers of ERA5 down to 1 m below the surface, LH and SH, CAPE, and PBL height (PBLH). In addition, we used the height of the lifted condensation level (HLCL) and the lifted condensation level deficit (LCL deficit), defined as difference between HLCL and PBLH. As HLCL was not available from ERA5, we used the approach from Georgakakos and Bras (1984) and Bolton (1980) which is based on surface pressure, 2m

temperature, and 2m dewpoint to derive HLCL which is also applied in Dirmeyer et al. (2014):

$$\text{HLCL} = \frac{R_d \, T_V}{g} * \log \frac{P_{SFC}}{P_{LCL}} \qquad (1)$$

$R_d$ is the gas constant for dry air, $T_v$ is the virtual temperature at 2 m above ground, $g$ is the acceleration due to gravity, $P_{SFC}$ is the surface pressure (hPa) and $P_{LCL}$ denotes the pressure of the lifted condensation level (hPa).

The strength of soil moisture-latent heat flux coupling (TCI $_{\eta\text{-LH}}$) between $\eta$ and LH is defined as


$$TCI_{\eta-LH} = \sigma(\eta) \frac{dLH}{d\eta} \qquad (2)$$

where $dLH/d\eta$ is the slope of the linear regression as described in Santanello et al. (2018) and $\sigma(\eta)$ describes the standard deviation of root zone soil moisture. Equation (2) describes the sensitivity of LH with respect to changes in the root zone soil moisture.

To derive the strength of the coupling between the land surface and the atmosphere (ACI), the standard deviation of $\eta$ can, e.g., be substituted by surface fluxes in Eq. 2 while LH in Eq. 2 can be substituted by PBLH or CAPE (Dirmeyer et al., 2014).

ACIs are computed 1) between LH and CAPE (ACI$_{LH\text{-CAPE}}$), and 2) between LH and HLCL (ACI$_{LH\text{-HLCL}}$):

$$ACI_{LH-CAPE} = \sigma(LH) \frac{dCAPE}{dLH} \qquad (3a)$$

$$ACI_{LH-LCL} = \sigma(LH) \frac{dHLCL}{dLH} \qquad (3b)$$

σ(LH) denotes the standard deviation of LH. The daily mean values, required for the indices, are calculated between 06 UTC and 18 UTC (Yin et al., 2023).

Water grid cells are not considered in our evaluation.

## 3 Results

### 3.1 Interannual variability of summer seasons 1991-2022


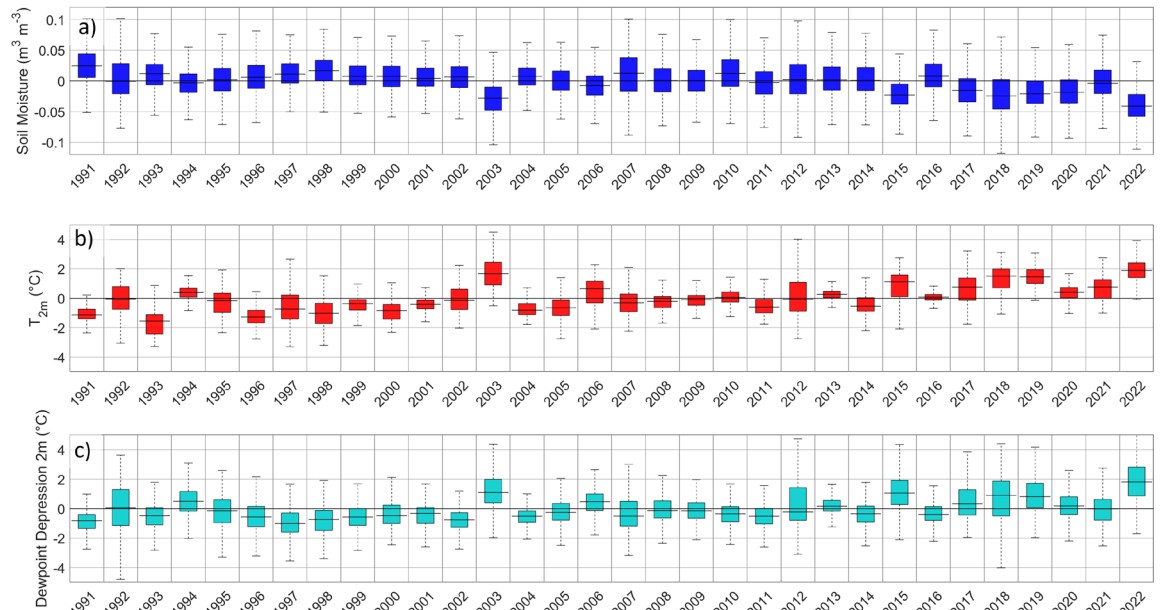

**Figure 1.** Interannual variability of anomalies of root zone soil moisture η (a), 2m temperature (b), and dewpoint depression (c) for the summer seasons between 1991-2022. The data are averaged over land grid cells in the region between 40°N-60°N and 5°W-25°E.

From the anomaly timeseries in Fig. 1a it is seen that from 2015 onwards the soil moisture content shows a tendency to decrease during summer except for 2016. The summer seasons 2003 and 2022 are the driest summer seasons since 1991. At the same time, a trend for a temperature increase of 0.5-1°C is observed from Fig. 1b since

160 2015.

Dewpoint depression anomalies (Fig. 1c) can be used as an indicator for the inhibition of cloud formation. A trend towards larger dewpoint depression is also observed here since 2015. As higher temperatures increase the evaporative demand of the atmosphere, this results in a further reduction of soil moisture and thus an enhanced dewpoint depression which is seen among the summer seasons after 2015 in Fig. 1. The anomaly spread of η and

2m temperatures do not increase during these years pointing towards a general warming and drying over our region of interest which will become more likely in the near future (Huebener et al., 2017; Rousi et al., 2022).

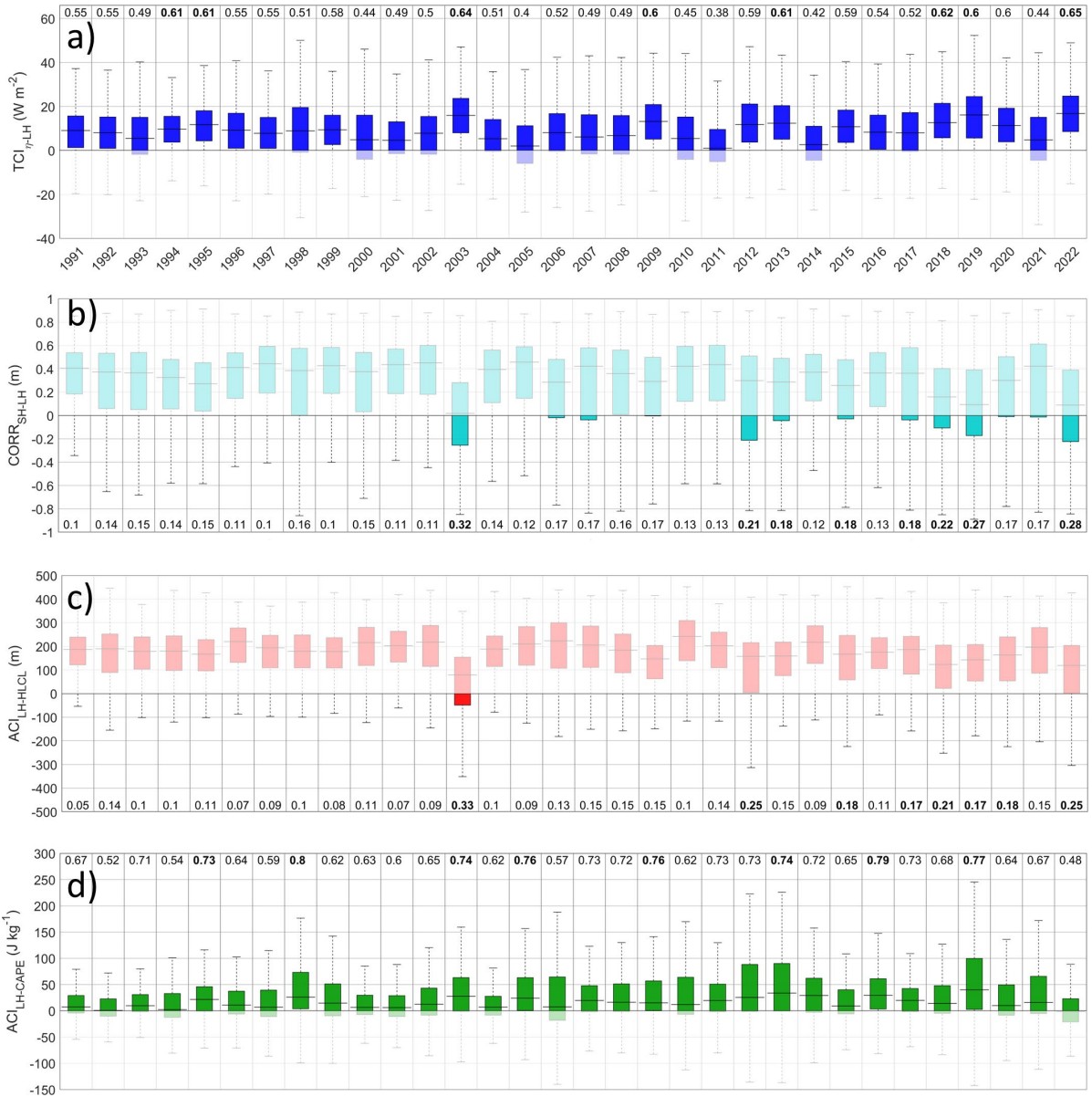

**Figure 2.** Interannual variability of the coupling indices TCI $_{\eta\text{-LH}}$ (a), correlation SH-LH (b), ACI$_{\text{LH-HLCL}}$ (c), and ACI$_{\text{LH-HLCL}}$ (d) for the summer seasons 1991-2022. The bold-faced numbers indicate the fraction of grid cells exceeding the 75[th] percentile of the respective index. Full colors denote the sign, at which the first variable of the index (e.g., η) drives the second variable (e.g., LH). The data are averaged over land grid cells in the region between 40°N-60°N and 5°W-25°E.

Figure 2 shows the interannual variability of the correlation between SH and LH, and the coupling indices TCI$_{\eta\text{-LH}}$, ACI$_{\text{LH-CAPE}}$, and ACI$_{\text{LH-HLCL}}$ during the summer seasons 1991-2022. The correlation between SH and LH during
the different summer seasons is mostly positive (Fig. 2b), however there are few exceptions for the very warm summer seasons 2003, 2018, 2019, and 2022 where the median of the correlation is less than 0.2 and the number of grid cells with negative correlations is increased. The median of TCI$_{\eta\text{-LH}}$ (Fig. 2a) shows higher values for the warm summer seasons (see Fig. 1b). Especially during the extremely warm and dry summer seasons 2003 and 2022 more than 90 % of the grid cells exceed the 75[th] percentile of the TCI$_{\eta\text{-LH}}$. The ACI$_{\text{LH-HLCL}}$ (Fig. 2c) does not
show a clear trend for an increase or decrease while usually only a small fraction of the grid cells exceeds the 75[th] percentile. However, during the warm and dry years a trend of ACI$_{\text{LH-HLCL}}$ approaching values around or below

zero is evident. For the $ACI_{LH-CAPE}$ (Fig. 2d) no clear trend for an increase or decrease can be observed which could give a hint that also the large-scale weather pattern can play a reasonable role in this case. It is worth noting that 2019 shows the largest variability of $ACI_{LH-CAPE}$ where 78 % of the grid cells exceed the 75th percentile.

Based on the interannual variabilities shown in Figs 1 and 2, we therefore decided to focus on summer seasons which have a median 2m temperature anomaly of more than 0.5°C which is proven to be a realistic estimate for changes of the maximum temperatures over land in the last decade (Forster et al., 2023). All anomalies were calculated using the Climate Data Operators (CDO) version 2.0.5 (Schulzweida, 2022).

| Year | 2003 | 2006 | 2015 | 2017 | 2018 | 2019 | 2020 | 2021 | 2022 |
|---|---|---|---|---|---|---|---|---|---|
| E-OBS Precipitation anomaly [mm] | -60.4 | -0.4 | -34.3 | -9.3 | -37.8 | -34.7 | 7.8 | -3.7 | -63.0 |
| ERA5 Precipitation anomaly [mm] | -59.4 | -8.7 | -38.9 | 0.2 | -36.1 | -32.4 | 17.0 | 15.1 | -37.9 |
| **Table 1.** Selected summer seasons based on a positive temperature anomaly larger than 0.5°C with respect to the climatological summer mean 1991-2020. The second row shows the median precipitation anomaly from E-OBS and the third row denotes the median precipitation anomaly from ERA5 | | | | | | | | | |

As seen from Fig. 1 and Table 1, the warm and dry summer seasons have become predominant since 2015. This has been associated with a strong reduction in annual and seasonal precipitation, combined with a reduced atmospheric water availability that led to a constant decline of the root zone soil moisture and, thus, to an agricultural drought. Although the median 2m temperature anomaly for summer 2020 was only 0.4 °C, it is considered in our analysis as this was the only summer with a moderate observed positive precipitation bias since

190  2015.

**3.2 Meteorological situation of the selected summer seasons**

This subchapter describes the synoptic conditions during each of the previously selected summers. The conditions comprise the 500 hPa geopotential, which informs about the large-scale weather pattern, the 2m temperature anomaly, the precipitation anomaly and the root zone soil moisture anomaly. A more detailed characterization of

the summers will be used for the interpretation of the coupling indices later.

**3.2.1 500 hPa geopotential**

Figure 3 shows the 500 hPa geopotential height anomalies for the selected summer seasons. The 500 hPa geopotential height helps to determine mid-tropospheric troughs and ridges describing the large-scale weather pattern. Most of the investigated summer seasons are characterized by positive 500 hPa geopotential anomalies

over large parts of Central Europe. The summer seasons 2003, 2019, and 2022 were characterized by a centric positive anomaly over central Europe with 2022 showing the highest positive anomalies of the investigated summer seasons. The summer seasons 2006 and 2017 were characterized by a meridional anomaly gradient around 50°N.  In summer 2006, positive anomalies were present over the British Isles and South Scandinavia while in 2017, positive geopotential anomalies were observed over South Europe. Summer 2018 was characterized by

strong positive anomalies north of 50°N and summer 2015 shows a moderate positive centric geopotential anomaly over Central Europe. During summer 2020, the 500 hPa geopotential shows a very weak zonal anomaly gradient so that it can be considered as an average summer compared with the climatology 1991-2020 (Fig. 3a). Summer

2021 was characterized by weak geopotential anomaly gradients while a higher anomaly was present over the British Isles.


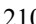

**Figure 3.** ERA5 500hPa geopotential anomalies [gpdm] for the selected summer seasons. The top left panel shows the mean summer 500 hPa geopotential 1991-2020.

### 3.2.2 Near surface temperature

The positive 500 hPa geopotential anomalies shown in Fig. 3 are associated with positive 2m temperature anomalies. The highest 2m temperature anomalies were observed during the summers 2003, 2018, 2019, and 2022 (Fig. 4b, f, g, j) and were spatially associated with strong positive geopotential anomalies over Central Europe.
During summer 2006, the 2m temperature anomalies are highest north of 51°N while during the summer seasons 2015 and 2017, the highest temperature anomalies were observed south of 50°N. This coincides with the fact that maximum positive geopotential anomaly is observed south of 51°N (Fig. 3d, e). Summer 2020 shows positive temperature anomalies over a wide area of our study domain. However, the 500 hPa anomalies were very moderate indicating a constant flow of cooler and moist airmasses from the West to Central Europe. Summer 2021 showed
a west-east anomaly gradient with temperatures slightly below the climatology over the western part of our investigation domain.

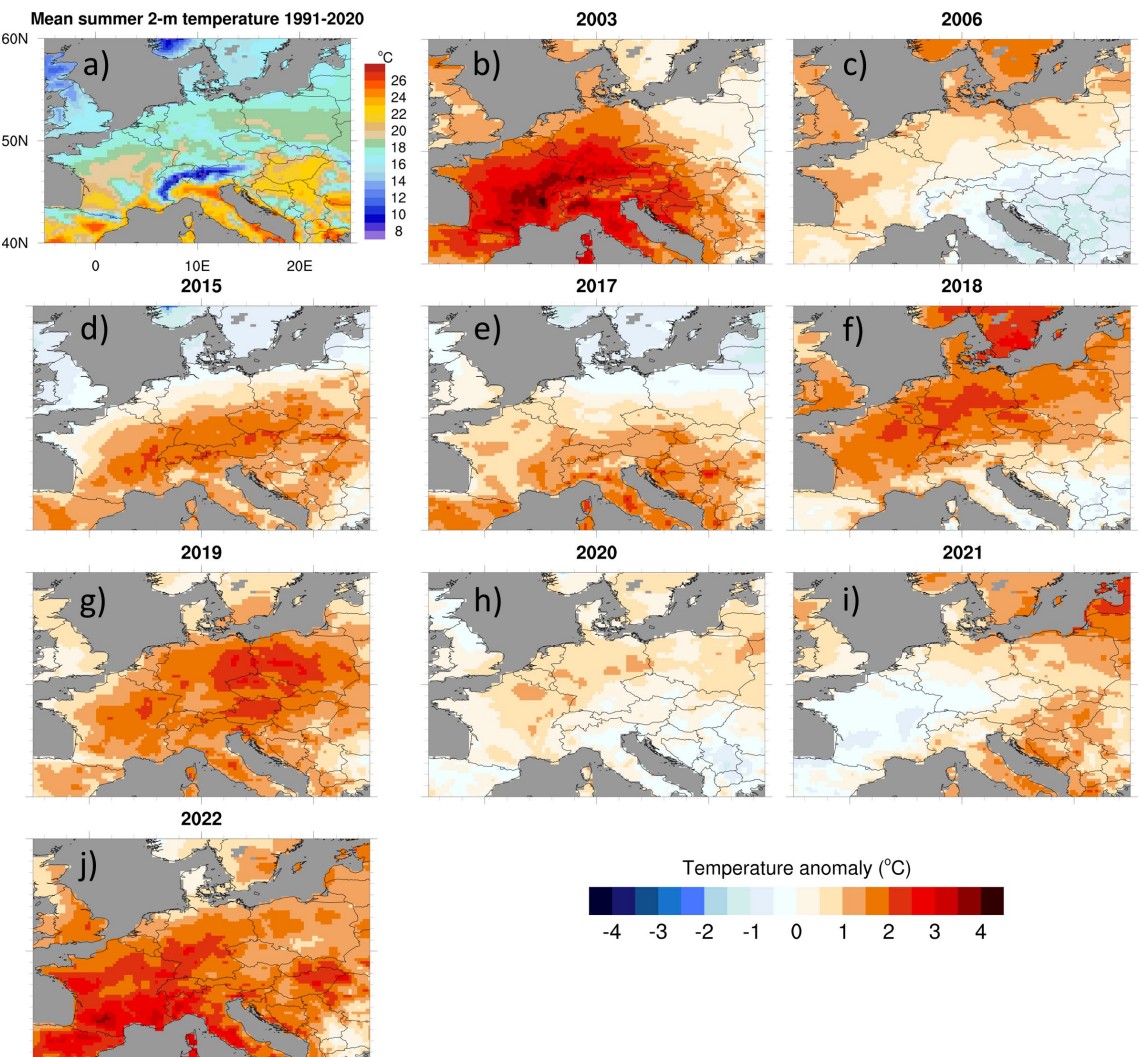

**Figure 4.** ERA5 2m temperature anomalies [°C]. The top left panel shows the mean summer 2m temperature 1991-2020 from ERA5.

### 3.2.3 Precipitation

Observed precipitation (Fig. 5 and Table 1) is often well below the climatological average 1991-2020. The summer seasons 2003, 2018, 2019 and 2022 were exceptionally dry (Rousi et al., 2023; Rousi et al., 2022) with a median precipitation anomaly between -34 mm and -63 mm. These extreme anomalies are also seen in the precipitation anomalies derived from ERA5 (Fig. 6) which reasonably catches these dry periods (Lavers et al., 2022). With respect to precipitation derived from E-OBS, 2006 can be seen as an average year with moderate precipitation anomalies over Central Europe. The summer season 2015 shows a strong dry anomaly associated with a warm temperature bias and positive 500 hPa geopotential anomalies. The summer season 2017 shows a strong wet bias over North Germany which is related to strong convective activity (e.g.,Caldas-Alvarez et al., 2022). Summer 2020 shows strong to moderate precipitation anomalies both in E-OBS and ERA5 over Germany, France, Poland, and Benelux while precipitation over Southeast Europe is above the climatological average resulting in an overall positive precipitation anomaly in both data sets. During summer 2021, precipitation over France, Benelux, and Germany was above average due to a small scale low-pressure system which caused the Ahr flood event (Mohr et al., 2023). This event was also simulated by ERA5 as indicated by the dark teal colors in Fig. 6j.

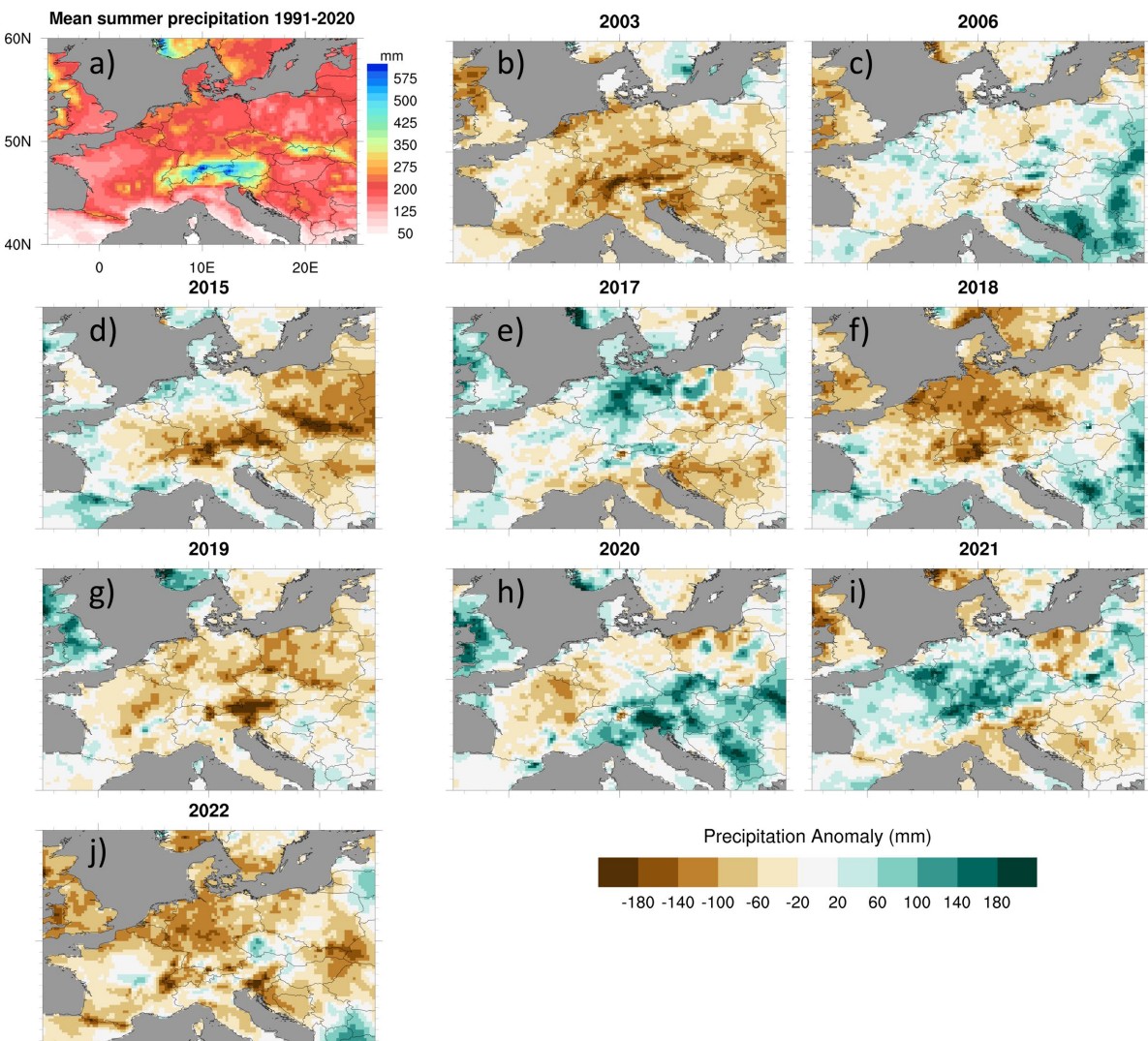

**Figure 5.** E-OBS precipitation anomalies [mm] for the selected summer seasons. The top left panel denotes the mean summer precipitation 1991-2020.

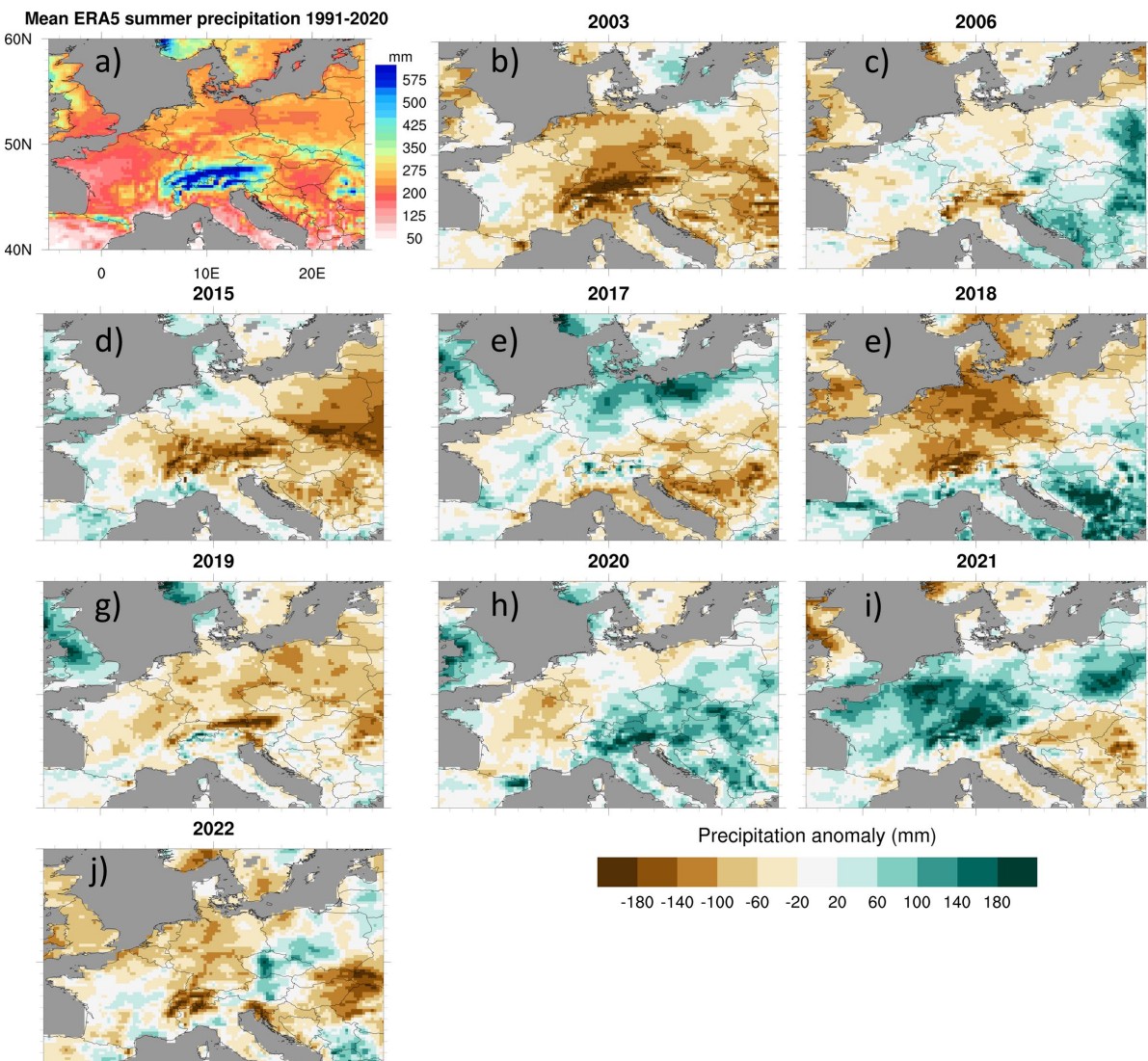

**Figure 6.** Same as Fig. 5 but for ERA5.

### 3.2.4 Soil moisture

Figure 7 displays the ERA5 derived root zone soil moisture anomalies. The summer seasons 2003, 2018, and 2022
show the lowest root zone soil moisture availability over Germany, Benelux, and France. This relates to the strong
positive temperature bias and the precipitation dry bias shown both by E-OBS and ERA5. An evaluation of the
median of the soil moisture anomalies over Central Europe revealed that summer 2006 is an average summer with
moderate positive anomalies over East Europe. The negative soil moisture anomaly during summer 2015 is related
to missing precipitation over large parts of Central Europe. Summer 2017 shows a strong positive soil moisture
anomaly over North Germany and North Poland related to the higher-than-average rainfall amount (see Figs. 5
and 6). Interestingly, although summer 2019 was among of the warmest and driest summers, the soil moisture dry
bias is less pronounced as in the other three hot and dry summer seasons 2003, 2018, and 2022 related to a higher
soil moisture content during spring (Fig. S2f). Summer 2020 shows drier than average soils over France and
Germany while soil moisture in the other regions is around or even above the climatological average. The summer

season 2021 shows strong positive soil moisture anomalies over Benelux and Germany which was related to colder than average April and May 2021 (C3S, 2022) as well as due to the Ahr flood event (Mohr et al., 2023).

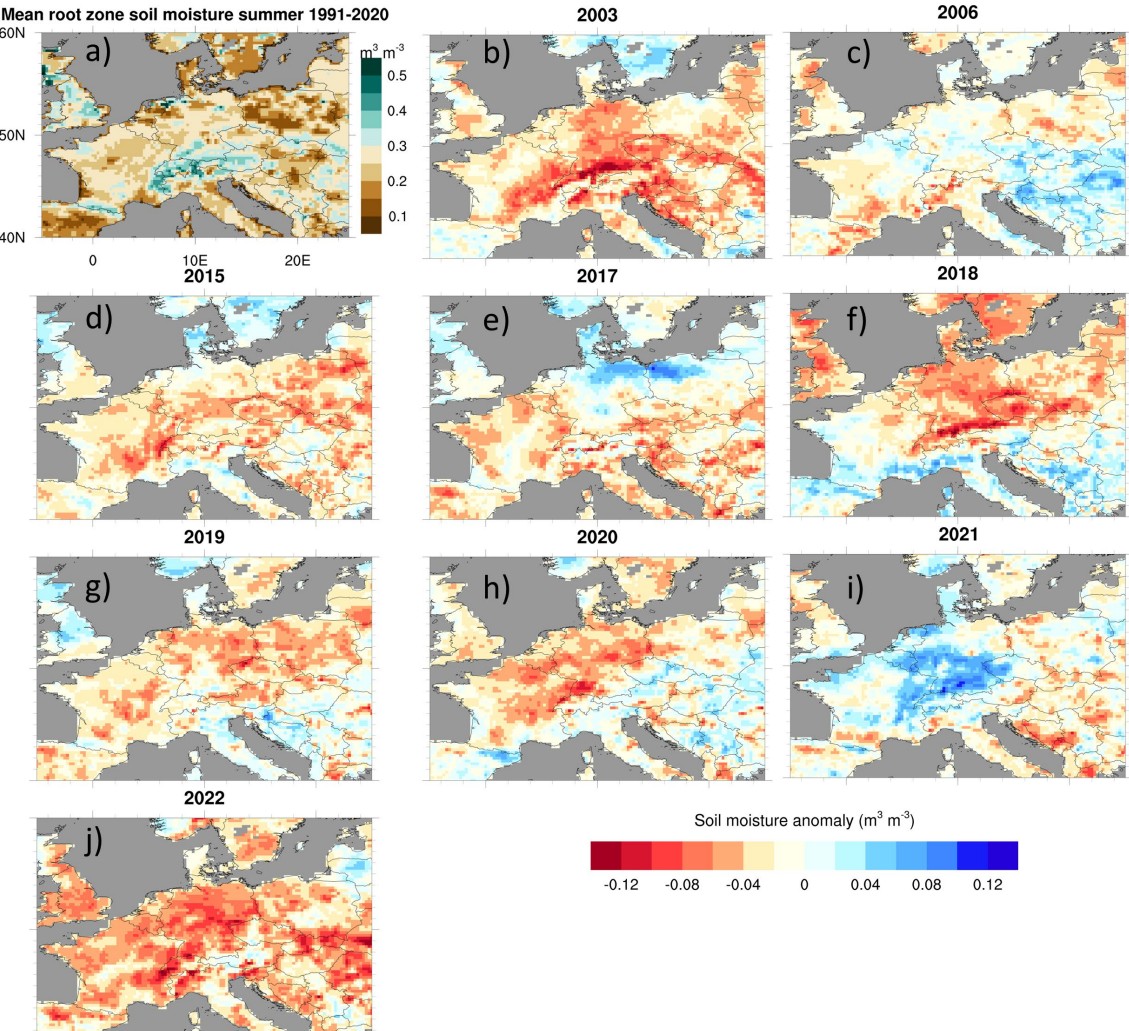

**Figure 7.** ERA5 soil moisture anomalies [m³ m⁻³] for the selected summer seasons. The top left panel denotes the summer mean root zone soil moisture 1991-2020 from ERA5.

### 3.2.5 Categorization of evaluated warm summer seasons

Although all years showed that most of the cells faced a considerable warm anomaly, the years diverge in the spatial patterns and the spatial extent of warm or cool as well as moist or dry anomalies. By visual examination it is possible to identify three groups within the hot years. Firstly, 2003, 2015, 2018, 2019, and 2022 stand out the most. They are characterized by large temperature anomalies, dry anomalies in soil moisture and precipitation extent over most of the land areas in our study domain. Secondly, 2017 and 2021 were warm, but also comparatively wet years. Finally, 2006 and 2020 both exhibited moderate anomalies in all the meteorological

fields shown before. In the following chapters, the groups will be referred to as "warm and dry", "warm and humid", and "moderate".

## 3.3 Terrestrial coupling

### 3.3.1 Soil moisture-latent heat flux coupling

In this section, we present the η-LH coupling based on the terrestrial coupling index ($TCI_{\eta\text{-}LH}$) for the selected summer seasons. The $TCI_{\eta\text{-}LH}$ describes how changes in soil moisture coincides with variations in LH. A positive $TCI_{\eta\text{-}LH}$ denotes that LH is limited by the root zone soil moisture and the soil moisture variation results in LH variation while a negative $TCI_{\eta\text{-}LH}$ indicates that the development of LH is energy limited, i.e., the incoming energy determines the LH development. In case the absolute $TCI_{\eta\text{-}LH}$ is low, either there is too little soil moisture available for evaporation, close to the wilting point, or the soil is too wet and a further increase does not lead to considerable changes in evaporation (Müller et al., 2021). Since the land surface influence on the convective and nocturnal boundary layer differs considerably due to the presence or absence of incoming shortwave radiation, all analyses base on daytime means computed for the period 06 UTC and 18 UTC of each day (Yin et al., 2023)

Figure 8 shows the $TCI_{\eta\text{-}LH}$ of all warm summer seasons shown in Table 1 which became the dominant situation over Europe since 2015. The very warm and dry seasons show a strong positive $TCI_{\eta\text{-}LH}$ over the regions affected by low soil moisture (Germany, France, and Benelux; Fig. 7a,e,f,i). In summer 2015, which is overall very dry with respect to soil moisture and precipitation, $TCI_{\eta\text{-}LH}$ shows neutral values over North Germany while the rest of the investigation domain shows positive values. The warm and wet summers show the lowest values for the $TCI_{\eta\text{-}LH}$ of all warm years. In the wettest areas of both years a switch in the sign of the index occurs. The neutral to negative values indicate the availability of sufficient soil moisture (Fig. 7), which in turn suggests a decoupling of the LH flux variation from soil moisture variation in these regions and years (compare Fig. 6 and 7).

During 2021, when a positive η anomaly is observed over Germany, Benelux, eastern France (Fig. 7h), the $TCI_{\eta\text{-}LH}$ becomes moderately negative in these regions with values of about -20 W m$^{-2}$ (Fig. 8h). This can be explained by a moist spring season (Fig. S2i) and the heavy precipitation event that occurred in June 2021 (Mohr et al., 2023) leading to a soil moisture content close to field capacity (Fig. S1b). A similar behavior of the $TCI_{\eta\text{-}LH}$ is observed during the two cold and wet summer seasons (not shown).

During the moderate summers 2006 and 2020, the $TCI_{\eta\text{-}LH}$ shows a heterogenous pattern with neutral to slightly positive values of up to 20 W m$^{-2}$ over most parts of Central Europe. The only exception is the alpine area and in 2006 the eastern part of our study domain where the $TCI_{\eta\text{-}LH}$ gets slightly negative.

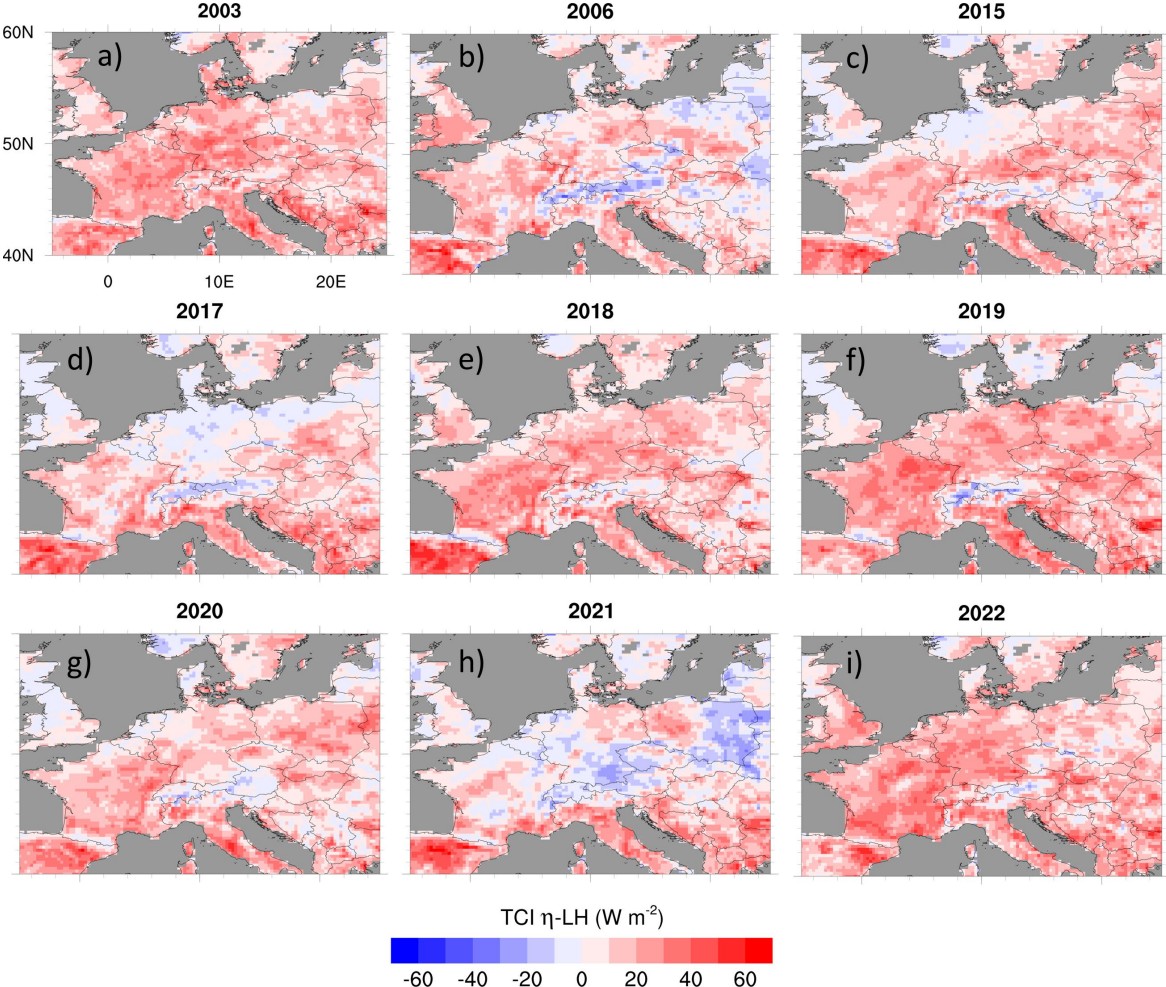

**Figure 8.** ERA5 based Terrestrial Coupling Index TCI $_{\eta\text{-LH}}$ between root zone soil moisture $\eta$ and LH for the selected summer seasons.

### 3.3.2 Correlation SH-LH

The majority of correlation coefficients are negative over the Iberian Peninsula and the Mediterranean, which is related to very low absolute evapotranspiration (Seneviratne et al., 2006). Over the British Isles, Scandinavia and the Atlantic coasts, the heat fluxes usually demonstrate a positive correlation.

During the warm and dry summers 2003, 2018, 2019 and 2022, the correlation LH-SH (Fig. 9) became negative over Germany, France, and Benelux. This is related to the anomalously warm and dry conditions in the atmosphere and a soil moisture deficit during these. The soil moisture deficit limits LH while SH is further increased. The SH increases due to a reduction of the evaporative cooling effect at the surface, and the consequent increase in the temperature gradient between land surface and atmosphere. During the warm and wet as well as the moderate years, the SH-LH correlations remain positive over Mid Europe and the patterns of the correlation coefficients largely resemble those of the TCI$_{\eta\text{-LH}}$ (see Fig. 8).

In 2017, the spring season showed a positive soil moisture anomaly over Germany, East Europe and the British Isles which is reflected in the strong correlation over these regions. The correlation pattern for summer 2021 is similar as during the cold and wet seasons 1997 or 2002 (not shown) where enough soil moisture is available for evapotranspiration.

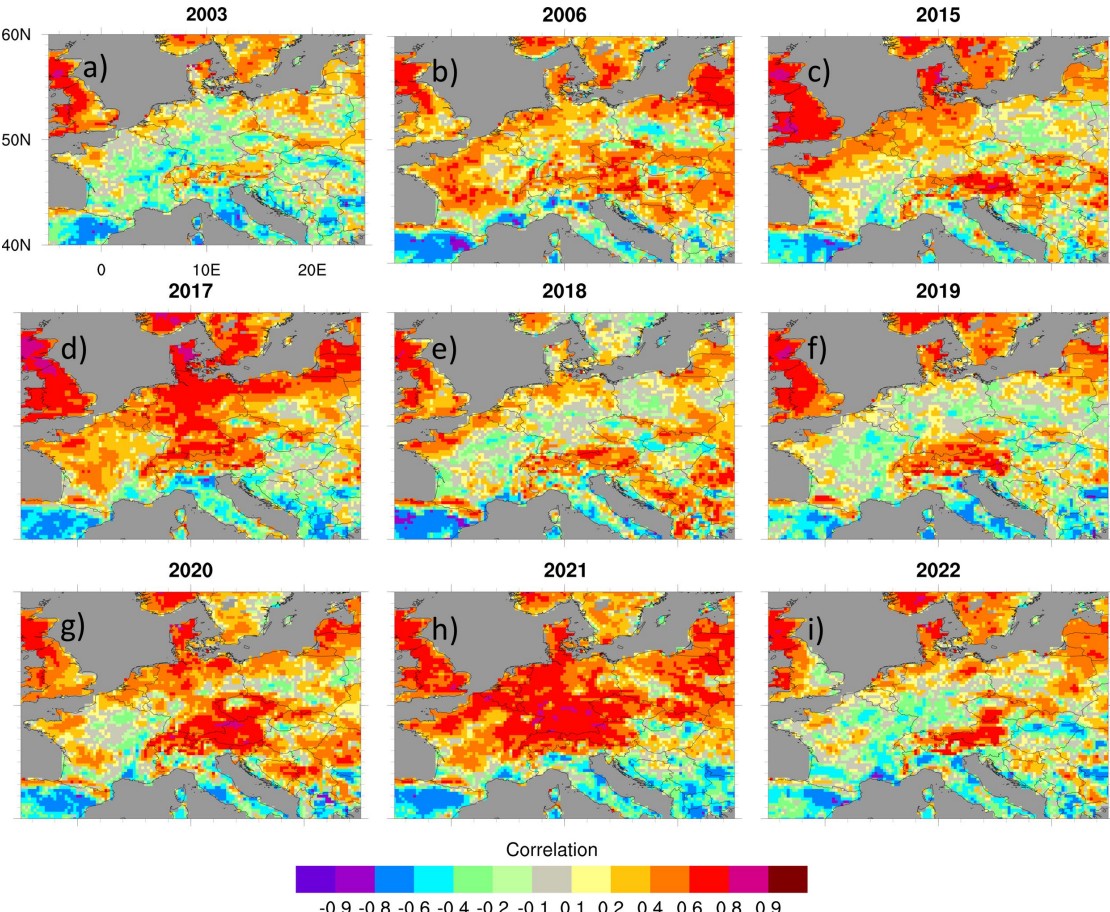

**Figure 9.** Pearson correlation coefficient between SH and LH for the selected summer seasons. Dark grey areas denote water grid cells.

## 3.4 Atmospheric coupling

### 3.4.1 Coupling LH-HLCL

This chapter explores the relationship between LH and HLCL and is complemented by an evaluation of the LCL deficit, building a bridge toward convective processes.

For the $ACI_{LH\text{-}HLCL}$ negative values are associated with a potentially physical relationship. An increase in LH means stronger PBL moistening by the land surface. Stronger moistening in turn suggests that saturation is reached faster and at a lower altitude meaning a lower HLCL The LCL deficit compares the heights of the PBL and the LCL (PBLH - HLCL). It can be employed as a proxy for the evolution of locally triggered deep convective processes. A positive LCL deficit means that the PBL top is above the LCL, when both heights are given in units of meters above ground. Hence, saturation occurs within the PBL, which is a prerequisite for locally triggered convective processes and cloud formation. Contrarily, a negative LCL deficit denotes an inhibition of convection developments (Santanello et al., 2011). Please note that Santanello et al. (2011) depict LCL and PBL on pressure levels, which leads to a switch in the sign in their interpretation.

The average patterns of the $ACI_{LH\text{-}HLCL}$ in the reference period indicate a physical influence of the LH on HLCL (negative values) over the South of the domain (Fig. 10). The negative values are limited to the Iberian Peninsula and the Mediterranean, where summers are typically strongly moisture limited. Simultaneously, the LCL deficit

is negative (Fig. 12) leading to a strong inhibition of the local formation of clouds and deep moist convection. Over France, Germany and the Balkan states, the patterns are patchy with negative or slightly positive values in valleys and strongly positive ones over mountain ranges. The LCL deficit is comparatively small with values of up to -300 m. This is the area in the study domain facing considerable interannual variability, which is reflected in sign changes, among other things. Over the rest of the domain, the values are primarily positive, which suggests no considerable influence of the LH on HLCL, although the LCL deficit has overall negative values throughout all summer seasons. This shows that saturation in the PBL primarily occurs in the North of our study domain.

During the warm and dry summers, Mid Europe experiences a switch in the sign from averagely positive to slightly negative values in the $ACI_{LH-HLCL}$ (Fig 11a, e, f, i). These areas mostly converge with those where the correlation between LH and SH also switched the sign (Fig. 8). At the same time, the moderately negative LCL deficit intensifies to up to -600m over Mid Europe to over -900m over the Iberian Peninsula (Fig. 11). This indicates that the very dry soil during these summers (Fig. 7) caused the low LH which in turn initiated a considerable increase of the HLCL (Fig. S5) and thus a higher LCL deficit as shown in Fig. 12. This is also shown by the negative values of the $TLCI_{\eta-LH-HLCL}$ (Fig. S3) showing feedback between $\eta$, LH and HLCL while only weak feedback between $\eta$, LH, and CAPE is present (Fig. S4). Please note that the SH is always positively correlated with the PBLH over land and doesn't experience strong interannual variability (not shown). This implies that a strong increase in the SH due to the LH limitation causes strong PBL heating and growth. This in turn pushes both the PBLH and the HLCL upward. Due to the combination of strengthened PBL heating and decreased PBL moistening the HLCL rises further, which leads to an intensification of the LCL deficit and thus inhibiting deep moist convection (Santanello et al., 2011). The areas with the strongest changes in the signal converge with the regions experiencing the strongest warm and dry anomalies (compare Fig. 3j, Fig. 5j, and Fig. 7j).

During the warm and humid as well as the moderate summers, the $ACI_{LH-HLCL}$ is positive over large parts of Central Europe indicating that LH variations are not the primary driver of the HLCL evolution. Further, the SH is not influencing the HLCL (not shown), which suggests a stronger atmospheric influence in the L-A system during moderate to humid periods. The pattern of 2006 overly corresponds to the pattern of the climatological average. During summer 2021, the positive soil moisture anomaly (Fig. 7) is connected to weak or negative coupling between $\eta$ and LH (Fig. 8).

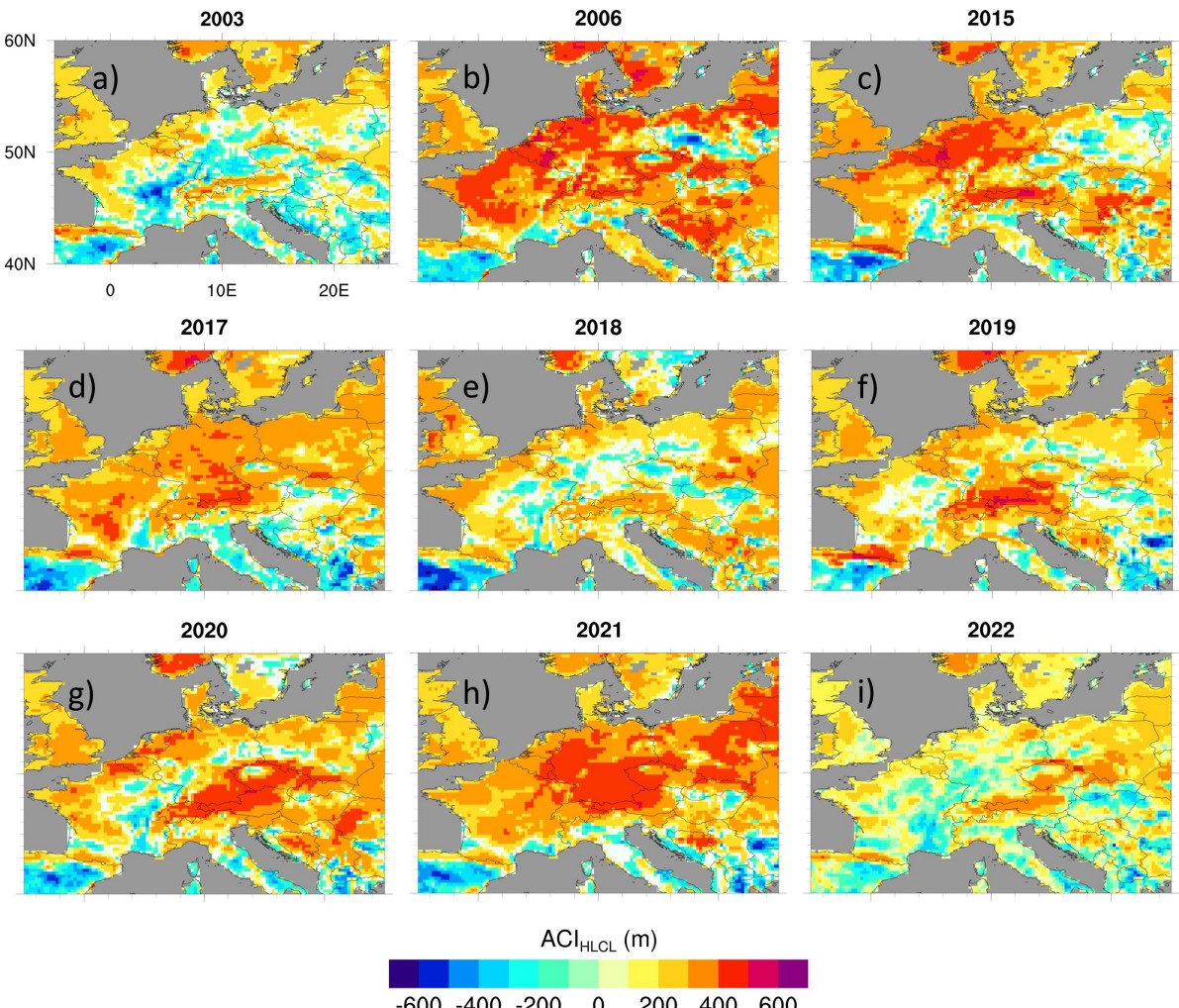

**Figure 10.** Atmospheric coupling index between LH and HLCL (ACI$_{LH-HLCL}$) for the selected summer seasons.

Over Germany, France, and Benelux, the ACI$_{LH-HLCL}$ shows low or negative values during the extreme warm and dry summer seasons 2003, 2018, and 2022 (Fig. 10a, e, i).

This indicates that the very dry soil during these summers (Fig. 7) caused the low LH which in turn initiated a considerable increase of the HLCL (Fig. S5) and thus a higher LCL deficit as shown in Fig. 12. This is also reflected by the negative values of the TLCI$_{\eta-LH-HLCL}$ (Fig. S3) pointing towards feedback between η, SH and HLCL while only weak feedback between η, LH, and CAPE is present (Fig. S4).

In summer 2006, 2015, and 2017 the ACI$_{LH-HLCL}$ is positive over large parts of Central Europe indicating that LH variations drive the evolution of HLCL. During summer 2021, the positive soil moisture anomaly (Fig. 7) is connected to weak or negative coupling between η and LH (Fig. 8). This implies that LH either has little variations or is high compared to other summer seasons and thus lowering HLCL (not shown, e.g., Wei et al., 2021) which is also reflected in a mostly neutral LCL deficit over Central Europe as shown in Fig. 11.

As the TCI$_{\eta-LH}$ is mostly positive over these regions during these summers, while the ACI$_{LH-CAPE}$ is neutral to slightly positive, this indicates that soil moisture variation impacts LH variations but with weak feedback to the atmosphere as indicated by the TLCI$_{\eta-LH-CAPE}$ (Fig. S4).

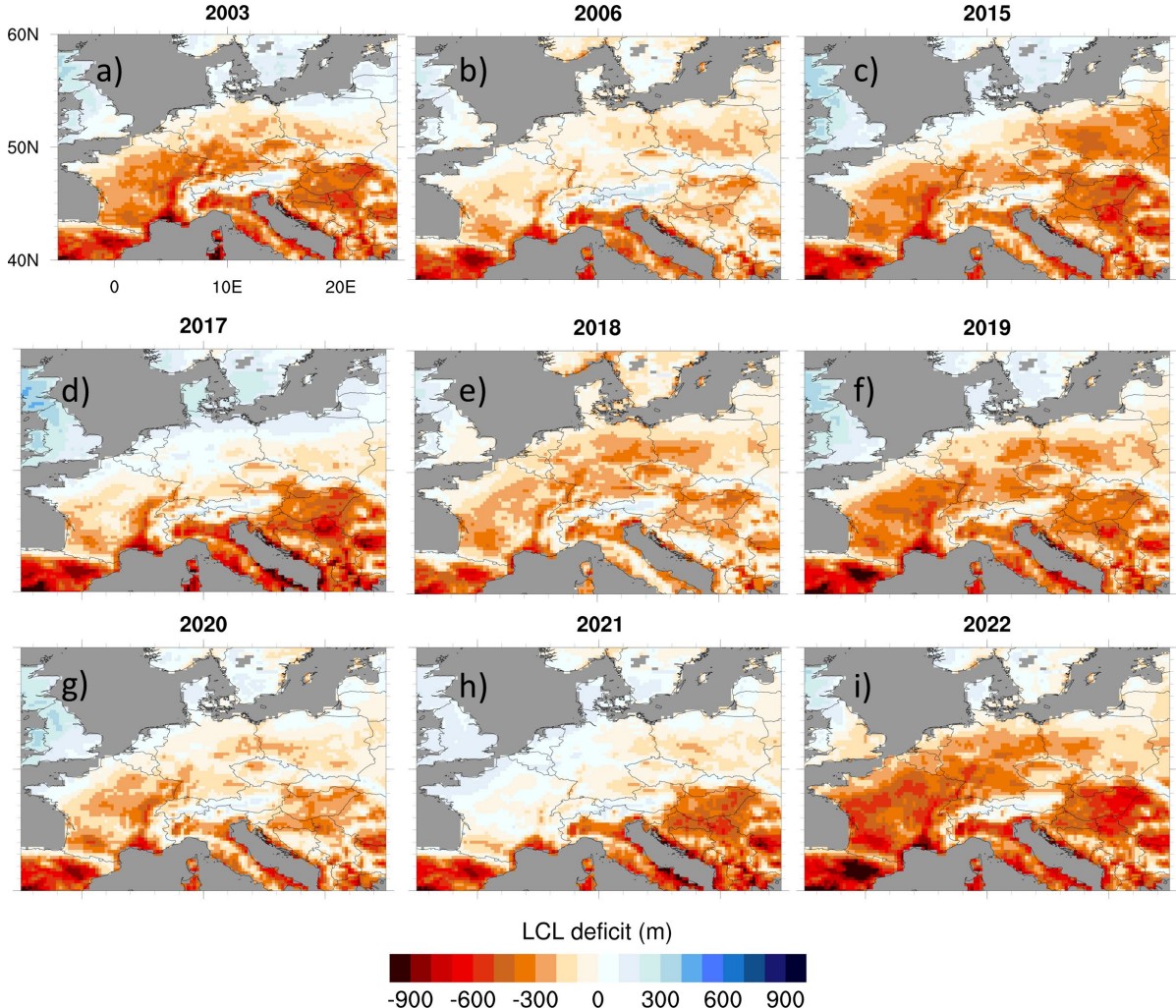

**Figure 11.** Mean ERA5 LCL deficit. Orange and reddish colors denote less favorable conditions for convection.

### 3.4.2 Coupling LH-CAPE

This section explores the results of $ACI_{LH\text{-}CAPE}$ for the warm summer seasons. This index aims at assessing the relationship between surface moistening of the PBL represented by LH and the energy in the atmosphere, which is potentially available for the development of deep moist convection (CAPE). CAPE represents the deviation of the atmospheric virtual temperature profile from the moist adiabat between the level of free convection and the equilibrium level.

This buoyant energy is typically stored a couple of hundred meters above the ground. It depends on both atmospheric humidity and the temperature gradient, which in turn are subject to surface influences through the surface heat fluxes. Through PBL moistening, an increase in LH can lead to an increase of CAPE which indicates the potential for convective developments and thus precipitation. CAPE depends on the atmospheric humidity which is, among others, related to LH while LH is related to the atmospheric temperature, humidity, soil moisture and LAI.

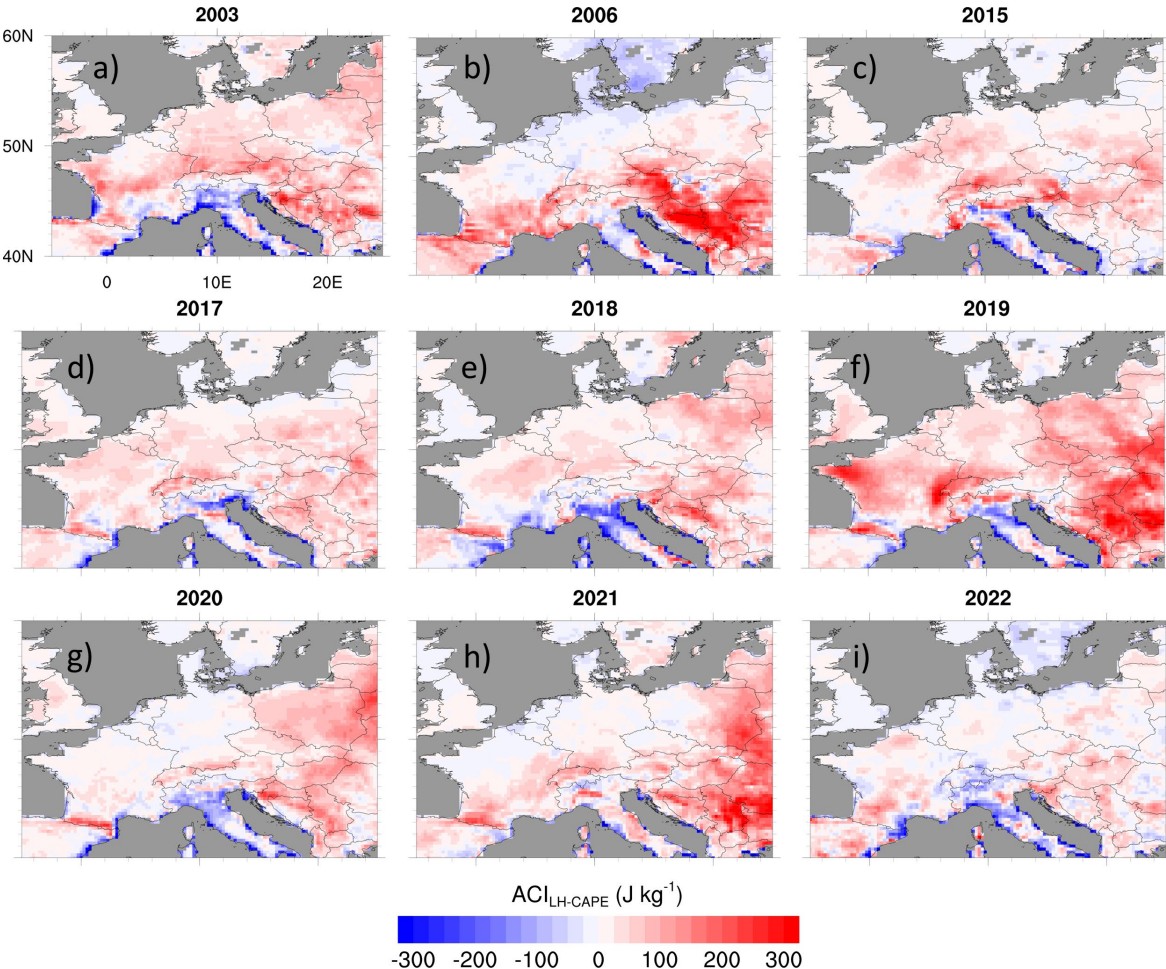

**Figure 12.** ERA5 based atmospheric coupling index between LH and CAPE (ACI$_{LH\text{-}CAPE}$). Grey areas denote water grid points.

A common feature is the negative ACI$_{LH\text{-}CAPE}$ along the coast of the Mediterranean. As the sea surface temperatures in this region can reach up to 26°C (García-Monteiro et al., 2022), this leads to high evaporation over the sea and thus high precipitable water values. Together with a temperature gradient of up to 30 °C or more in the Mediterranean between 850 hPa and 500 hPa (not shown), this leads to stronger atmospheric instability and thus reduced coupling to LH.

Coupling hot spots are observed over East and Southeast Europe with ACI$_{LH\text{-}CAPE}$ values of more than 250 J kg$^{-1}$ in summer 2006, 2019, 2020, and 2021 (Fig. 12). They are related to higher values of LH over these regions (not shown) due to neutral or positive root zone soil moisture anomalies (Fig. 7). These coupling hot spots were also observed in a climate sensitivity study of Jach et al. (2022) who added climate change perturbations of temperature and moisture to a RCM simulation. Over Germany and France, mostly only weak coupling is seen with stronger signals during 2003 and 2019. In case evapotranspiration is not limited by soil moisture, the incoming radiation is allowing for potential evapotranspiration and surface latent and sensible heat fluxes are partitioned accordingly.

In case evapotranspiration is not limited by incoming radiation but by available soil moisture, evapotranspiration is below the potential rate leading to higher Bowen ratios and a further increase in temperature. This enhances evapotranspiration and therefore a gradual decrease in soil moisture towards wilting point. According to Benson and Dirmeyer (2021) this ultimately leads to the situation that LH almost vanishes and the incoming radiation mainly transforms into sensible heat which can exacerbate heatwaves and droughts. The low values of $ACI_{LH-CAPE}$ over the British Isles and South Scandinavia suggest that these regions are more frequently impacted by large scale synoptic systems with a more stable atmosphere rather than localized precipitation events (Jach et al., 2020). This is also reflected by the positive LCL deficit shown in Fig. 11.

## 4 Discussion

Our objectives were to evaluate interannual variability of coupling strength over Central Europe (summer 1991-2022), and to further investigate the warmest nine summer seasons in the context of the prevailing temperature and humidity anomalies. We now discuss the key findings.

Our first finding is that interannual variability occurs in different coupling relationships throughout the summer seasons 1991-2022. A comparison of the variability in the indices reveals a connection with temperature and moisture anomalies on the interannual scale, which was also suggested for the climate time scale by Jach et al. (2022). This connection is particularly shown in indices showing a relation associated with the hydrological cycle ($TCI_{\eta-LH}$, correlation SH-LH, $ACI_{LH-HLCL}$). Guo and Dirmeyer (2013) for instance also showed interannual variability in the coupling of soil moisture with surface fluxes and temperature associated with soil moisture anomalies.

In this study, the $TCI_{\eta-LH}$ shows a year-to-year variability during the full period of summer seasons 1991-2022. However, there is a trend of an increased coupling during the warm and dry summers (Fig. 2a). This indicates that variations of $\eta$ drive LH as there is not enough soil moisture available for evapotranspiration. The average correlation SH-LH stays mainly positive, however especially in the warm and dry summer seasons the correlation became negative. This further suggests a moisture limited coupling regime. The interannual variability of $ACI_{LH-HLCL}$ shows a trend towards zero or even negative values during the warm and dry summer seasons which indicates less moistening of the PBL due to insufficient evaporation from the land surface and thus an increase of HLCL.

The average interannual variability of $ACI_{LH-CAPE}$ shows only little variations throughout the summer seasons. This could be related to a weaker direct connection between changes of LH impacting CAPE due to the present atmospheric stratification which is not only impacted by the surface conditions but also by the large-scale weather pattern and atmospheric stratification.

However, the $ACI_{LH-CAPE}$ shows coupling hot-spots over Southeast and East Europe as well as over the Baltic states which were also observed in the study of Jach et al. (2022) who applied a climate temperature change signal to an existing 30 year simulation.

From the interannual variability of the different variables shown in Figs. 2 and 3, it can be concluded that warm and dry summer seasons show, on average, a differing behavior of the coupling strength. This matches with the finding of Guo and Dirmeyer (2013), that areas with normally wet climate can experience a shift in coupling regimes under dry conditions. On the seasonal time-scale, Lo et al. (2021) also found regime shifts due to flood or drought conditions. As the current global warming trend reflects in more frequent hot and dry conditions over Central Europe, it was decided to focus only on nine hot summer seasons between 1991-2022 (sec, 3.2.5). Koster

et al. (2009) suggest that precipitation dry bias is the result of strong positive temperature anomalies because of reduced evaporative cooling and increased SH. This is confirmed by an evaluation of summer temperature and precipitation anomalies over our region of interest which yield correlations ranging from -0.25 to -0.65 between temperature and precipitation anomalies. This suggests that hot and dry conditions will often coincide in the future and the regime shifts as discussed below, will occur with a higher frequency. During the warm and humid and moderate summer seasons a switch of the regime is rarely visible.

The coupling signals remain stable throughout the summer seasons over North Europe and the Mediterranean region (Seneviratne et al., 2006; Knist et al., 2017; Jach et al., 2020; Jach et al., 2022). The correlation between SH and LH is mainly positive over the British Isles, indicating that evapotranspiration is limited by the incoming energy (Knist et al., 2017) which is also the case over France, Benelux, and Germany for summer 2021 (not shown).Over Central and East Europe changes in the coupling regimes occur between the individual summers which is indicated by switches in the sign of multiple indices. This area coincides with the transition zones which was also observed in the studies of Knist et al. (2017) and Jach et al. (2022).

A common feature of the warm and dry summer seasons is the anticorrelation of LH and SH south of 44 °N (Fig. 8). These regions are usually water-limited leading to limited evapotranspiration thus further reducing LH. As enough incoming solar energy is present in these regions, this further enhances SH and thus could further intensify drought periods (positive coupling). Together with the positive TCI $_{\eta\text{-LH}}$ the anticorrelation of SH-LH points to a strong limitation of evapotranspiration by insufficient root zone soil moisture. Though not yet represented in the model, in reality, this results in a low LAI which is often the case in South Europe (see Fig. S6c, d). Moisture-limitation of the LH in the warm and dry summers leads to a shift in the energy flux partitioning towards reduced PBL moistening and amplified PBL heating because of increased SH. This shift causes a drying throughout the PBL, which is shown by an increased HLCL (Fig. S5) and an intensified negative LCL deficit. Thus, the dry and warm conditions at the land surface propagate through the atmosphere and feed back in less favorable conditions for local convection.

As an example, the year 2018 started with an already warmer than average and slightly drier spring season over Germany (Xoplaki et al., 2023) turning into a severe drought due to a strong soil moisture depletion (Rousi et al., 2023) resulting in  an exceptionally low LAI (Fig. S6c). Dirmeyer et al. (2021) found that when the volumetric soil moisture content falls below a critical value, surface heating becomes extremely more sensitive to further surface drying amplifying the intensity of heatwaves.

According to Rousi et al. (2022) the frequency of the occurrence of heat waves is accelerating over Europe in the last 30-40 years where the large scale circulation pattern often features mid- and upper troposphere blocking situation leading to a split of the jet stream towards the Arctic and the Mediterranean. As the jet stream is an important feature for the European weather, it can also alter the near surface flow conditions in West and Central Europe (Laurila et al., 2021) while in other regions like the Mediterranean and East Europe, soil moisture preconditioning is more important as the impact of the jet stream becomes weaker (Prodhomme et al., 2022).

If enough moisture is in the local L-A system, (warm humid and moderate years) variations in the moisture at the land surface do not play a strong role locally. This is shown by a decoupling in several links along the local coupling (LoCo; Santanello et al., 2018) coupling chain: TCI$_{\eta\text{-LH}}$ negative, LH and SH co-vary, ACI$_{\text{LH-HLCL}}$ positive. However, during the humid or moderate summers the LCL deficit gets positive and locally triggered deep convection can occur.

As an example, in the warm and humid summer 2021 a strong SW-NE temperature anomaly gradient associated with a strong 500 hPa geopotential anomaly gradient around 55°N was evident. This led to a stronger westerly flow air which allows for more humid air masses from the Atlantic. A major event during this summer was the flood event mid of July 2021 which affected larger areas of West and Central Europe and lead to extreme precipitation of more than 150 mm d$^{-1}$ (Ludwig et al., 2023; Mohr et al., 2023). This heavy precipitation event, which was also captured by ERA5 (Fig. 6h), was caused by a slow moving small-scale low-pressure system over France and Benelux and led to a longer lasting positive soil moisture anomaly from mid of July onwards. The anomaly is directly reflected in negative TCI$_{\eta\text{-LH}}$ values and a strong positive correlation between LH and SH as enough surface moisture was available for evaporation. The pattern of the correlation SH-LH and the pattern of ACI$_{\text{LH-HLCL}}$ largely resembles each other which is also observed in the cold and wet summer seasons (not shown). The LCL deficit in Fig. 10 is mainly positive over Central and South Europe which is associated with a negative precipitation anomaly over the respective areas. On the other hand, the negative LCL deficit over the British Isles is directly connected with a positive precipitation anomaly (especially during summer 2019 and 2020) indicating that LA feedback processes were driven by low pressure systems. Although ERA5 is the most comprehensive reanalysis data set currently available (Hersbach et al., 2020), some limitations have to be acknowledged. Like many other numerical weather prediction (NWP) models, ERA5 applies a static LAI climatology (Fig. S6b) which was derived from the period 2000-2008 (Boussetta et al., 2013; ECMWF, 2016). However, under a changing climate the interannual variability of LAI is enhanced as shown by the satellite derived data from the Copernicus Global Land Service (CGLS) project (Fuster et al., 2020) (Fig. S6c,d). Data such as these could help to further improve, e.g., the simulated evapotranspiration. Vegetation-climate dynamics are presumed to intensify the response in the regimes, as dry conditions e.g. cause less vegetation growth or vegetation dying, which potentially further reduces the LH and exacerbate the effects described above.

On the other hand, Denissen et al. (2020) found that LSMs tend to overestimate the critical soil moisture (Hsu and Dirmeyer, 2023).  A recent study of Warrach-Sagi et al. (2022) using the LSM NOAH-MP (Niu et al., 2011) showed that, even on a convection permitting (CP) horizontal resolution, LA feedback strength tends to be underestimated when using a LAI climatology in numerical weather prediction (NWP) models as compared to including the dynamic vegetation model GECROS (Yin and van Laar, 2005). Since, this is in contrast to the results of Denissen et al. (2020) indicating the need for further enhancements of the applied LSMs (Hersbach and Bell, 2022; He et al., 2023) and the need to investigate the role of dynamic vegetation in the LA system.

However, Martens et al. (2020) evaluated LH from ERA5 against FLUXNET stations (Pastorello et al., 2020) for the period 1991-2014. Their analysis revealed that ERA5 surface fluxes perform well in a moderate temperature climate. ERA5 soil moisture shows reasonable correlations of up to 0.7 over Europe (Muñoz-Sabater et al., 2021) while LH in ERA5 tend to be overestimated on average by about 9 W m$^{-2}$. This could be related to an overestimation of wet days in combination with underestimated sub-daily precipitation rates (Beck et al., 2019). Hence, although limitations are present in the reanalyses data set, they suggest that the exact values of coupling indices can vary but the sign and magnitude of the indices are robust

## 5 Summary

This study provides an assessment of temporal variability in three coupling relationships during the summer seasons between 1991 and 2022 for Central Europe. The relationships under investigation are soil moisture-LH

coupling at the terrestrial leg of the local coupling chain, LH-CAPE as well as LH-HLCL coupling comprising two relationships of the atmospheric leg. Firstly, the interannual variability between all years of the period was examined in the context of prevailing temperature and moisture anomalies in the light of a warming climate and a projected increase in hot and dry periods until 2100 (Huebener et al., 2017). The second part of the analyses focused on the coupling during the nine warmest summers of the period.

Soil moisture availability during the summer seasons 1991-2022 show a decreasing trend while average 2m temperatures shows an increase of about 0.5°C since 2015. At the same time, the dewpoint depression anomalies show strong positive signals during the very warm and dry summer seasons 2003, 2015, 2018, 2019, and 2022 indicating a drier PBL and potentially leading to a suppression of the development of convection. The interannual variability of the correlation SH-LH as well the $TCI_{\eta\text{-}LH}$ also reflected the exceptional warm and dry summer seasons. Therefore, it was decided to further investigate the summer seasons exceeding a median temperature anomaly of +0.5°C based on the ERA5 summer mean value of 1991-2020 (WMO, 2017).

To enhance our analysis, anomalies of the 500 hPa geopotential, volumetric root zone soil moisture, and precipitation anomalies derived from ERA5 and E-OBS (Cornes et al., 2018), were considered for the interpretation of the results. Our results revealed that the investigated summer seasons are characterized by positive geopotential anomalies throughout Europe. Strong geopotential anomalies are linked to considerable positive 2m temperature anomalies strong dry soil moisture anomaly.

The analysis of the LA coupling strength was performed by means of different coupling indices like TCI, ACI (Dirmeyer, 2011; Santanello et al., 2018) as well as the correlation between SH and LH (Knist et al., 2017) by applying the coupling metrics framework provided by Tawfik (2015). All indices were calculated from ERA5 data using daytime values between 06 UTC and 18 UTC for each day (Yin et al., 2023). Reanalyses can be used as a reference for a further analyses and evaluation of climate simulations. However, these investigations requires high-frequency and high spatial resolution model output from NWP models (Findell et al., 2024) which is still a challenging task.

The interannual variability of the summer seasons revealed a temperature increase which is accompanied by a decline in soil moisture and an increase in the dewpoint depression which is most prominent in the especially warm and dry summers 2003, 2015, 2018, 2019, and 2022.

The warm and dry conditions lead to an intensification or even the onset of statistically measurable coupling in the various processes along the LoCo process chain. In wet years, LH does not depend on the soil moisture availability as sufficient transpiration of the leaves is possible (see Fig. S5d) and the HLCL is not primarily controlled by the lack of moisture at the surface.

The increasing frequency of warm and dry summers from 2015 onwards hints toward a trend of extended periods of reduced soil moisture available- for evapotranspiration and the likelihood of locally triggered convection. This leads to a growing influence of soil moisture variability on the meteorological conditions which was not as pronounced before 2003 due to cooler and moister conditions. Markonis et al. (2021) found a considerable increase in drought events over Central Europe since 2010 which they relate to increasing temperature and a lack of rainfall which together cause a soil moisture depletion due to excessive evapotranspiration.

The switches in the sign of the coupling indices imply that on the seasonal time scale local soil moisture and temperature anomalies can cause an exceedance of thresholds along the LoCo process chain. This has the potential to changes the role of the land surface as the driver for the local LA-system on the interannual time scale, and thus needs to be considered for sub-seasonal to seasonal (S2S) forecasts.

**Acknowledgements**

By the time of writing the manuscript, LJ was funded by the German Ministry of Education and Research (BMBF) project ClimXtreme (subproject LAFEP, grant number 01LP1902D). Copernicus Climate Change Service (2018) data was downloaded from the Copernicus Climate Change Service (C3S) Climate Data Store
https://cds.climate.copernicus.eu/cdsapp#!/dataset/reanalysis-era5-single-levels?tab=overview. The results contain modified Copernicus Climate Change Service information 2020. Neither the European Commission nor ECMWF is responsible for any use that may be made of the Copernicus information or data it contains. We thank the four anonymous reviewers for their valuable comments to further improve the quality of the manuscript.

*Code availability*

The code used in this study to calculate the coupling indices is obtained from https://github.com/abtawfik/coupling-metrics. The NCL software package can be downloaded from https://www.ncl.ucar.edu/current_release.shtml.

*Data availability*

E-OBS data were downloaded from https://surfobs.climate.copernicus.eu/dataaccess/access_E-OBS.php and the
ERA5 data are available at https://cds.climate.copernicus.eu/cdsapp#!/dataset/reanalysis-era5-single-levels?tab=overview.

*Author contributions*

TS, LJ, VW, and KWS conceived the idea for the LA feedback study presented here. TS processed the data and
graphics and performed the analyses together with LJ and KWS. The paper was written by TS with support of all coauthors.

*Competing interests*

The authors declare that they have no competing interests.

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
