# Peer review of "Soil moisture-atmosphere coupling strength over Central Europe in the recent warming climate"

_EGUsphere, 2023_

## Author Comment (AC1)

Revision on: Schwitalla et al., "Soil moisture-atmosphere coupling strength over Central Europe in the recent warming climate"

The manuscript provides an analysis of the inter-annual variability in land-atmosphere coupling focusing on the summers from 1991 to 2022 in Europe. The analysis uses the ERA5 reanalysis product and combines different coupling metrics and atmospheric and soil conditions to try to explain the drivers of summer conditions. Although the main idea of the manuscript sounds interesting, the text and explanation of the results are very confusing and in my opinion is not solid enough as it is to be published.

We thank you for carefully reviewing our manuscript. You suggested to focus on central Europe (see below), therefore we revised our investigation for the region of 5°W-25°E, 40°N-60°N. We refer from now on to this region unless stated otherwise. Please find our responses to your comments as marked in blue.

Specific comments:

The authors decide to use a reanalysis for the study of LA coupling, soil conditions, surface pressure and air temperatures but then they use the E-Obs product for precipitation. Despite the good performance of E-Obs in precipitation, if the aim of the study is to analyze the interconnection between variables and LA metrics in the "reality" of the ERA5 product, I do not agree with the use of E-Obs in precipitation, when all the rest of variables are related to the ERA5 simulated precipitation. The change of E-Obs by ERA5 precipitation may not change the results, since they may agree in the classification of dry and wet summers but in my opinion it is a required step for adding consistency to the analysis.

Different from temperatures, humidity, pressure, and soil moisture, precipitation is not assimilated into the ECMWF model for the ERA5 product and therefore is a pure diagnostic variable of the model with a strong dependence on the convection and microphysics parameterization.

For our study we require summer anomalies in precipitation to identify warm dry summers. The magnitude of the mean summer precipitation 1991-2020 is larger in ERA5 than in EOBS throughout the domain with maxima of 30 % in mountainous regions, especially the Alps. Much more important for our investigation are summer precipitation anomalies in each year. These show the same general patterns and magnitudes in ERA5 and EOBS. Below in figure R1 please find example plots of precipitation anomalies from ERA5 and E-OBS for the very wet and warm summer 2010 (not used in our study) and the extremely dry summer 2003 showing minor precipitation differences in extreme years.

[Figure]

[Figure]

Fig. R1: Precipitation anomaly for summer 2003 (top row) and 2010 (middle row) derived from ERA5 (left column) and E-OBS (right column). The bottom row shows the mean precipitation difference of the summer months between 1991-2020.

Therefore, we had decided to use E-OBS in this study. However, following your suggestion, we complete the analysis adding ERA5 precipitation.

The use of ERA5 product for the study of LA coupling is also controversial per se, since land surface models like the one employed in ERA5 have several difficulties in simulating LA interactions (e.g. Dirmeyer et al., 2018) and the authors have not validated the ERA5 simulation of the coupling metrics employed in the analysis.

Standalone land surface model simulations do not impact the atmospheric variables. Dirmeyer et al. (2018) refer the "difficulties in simulating LA interactions" of such stand-alone model simulations. Further it is not clear which specific difficulties in the LSM HTESSEL you are referring to in your comment. Fig. 9 and Table 2 in Dirmeyer et al. (2018) display reasonable results of the stand-alone land-surface model HTESSEL (EL) which is applied in ERA5. In section 2b they mention that this HTESSEL stand-alone simulation is forced by altitude corrected atmospheric variables from the coarse ERA-Interim product as only relation to a Reanalysis product.

Reanalyses data provide the three-dimensional consistent land and atmosphere gridded multidecadal timeseries of diurnal cycles closest to observations and therefore allow to study on the regional climate scale LA coupling beyond studying surface variables. It is not the goal of this study to evaluate reanalysis data or a model simulation. It is the goal to study the land-atmosphere interaction based on the best available 3-dimensional long-term gridded data set for Europe which is also applied in very recent studies of Rousi et al.(2022) and Rousi et al. (2023) to investigate feedback effects between the atmosphere and the land surface.

Coupling metrics are based on variables that are currently not available at the same temporal and spatial resolution from observations but from model simulations or reanalysis data (Findell et al., 2023). Therefore, the metrics themselves cannot be evaluated. The ERA5 variables used in this study (soil moisture, 2-m temperatures, surface sensible and latent heat flux) were successfully applied in LA feedback studies of Sun et al. (2021), Qi et al. (2023), and Rousi et al. (2023).

Given that the validation of the ERA5 product simulation of LA coupling would correspond to another article, I recommend adding another reanalysis product or model to the analysis and focus on the agreement between models. If the authors still consider that adding more products is too complicated then I would recommend to add another section to the article discussing the possible effect of ERA5 uncertainties (e.g. in precipitation, LH and SH) reported in the literature on the results (not in general as it was done in line 391).

Following your suggestion further below, we investigated the study of Martens et al. (2020) which relates to the study of Muñoz-Sabater et al. (2021). As shown in both studies, the representation of the surface fluxes, soil moisture and net radiation in ERA5 is very reasonable compared to in-situ measurements over Europe.
We added the following to the end of the discussion section:

"Martens et al. (2020) evaluated surface latent heat fluxes from ERA5 against FLUXNET stations (Pastorello et al., 2020) for the period 1991-2014. Their analysis revealed that ERA5 performs well in moderate temperature climate which is the case over Europe. ERA5 soil moisture over Europe shows a reasonable correlation of up to 0.7 over Europe (Muñoz-Sabater et al.,2021) while LH in ERA5 tend to be overestimated on average by about 9 W/m$^{-2}$ when compared to all stations. SH in general shows almost no bias. This overestimation could be related to an overestimation of wet days in combination with underestimated sub-daily precipitation rates (Beck et al., 2019). The overestimation of precipitation resulting in higher LH estimates could lead to an increased atmospheric instability and thus affecting the ACI and the LCL deficit."

Regarding your point of using other/additional reanalysis products. Of course, in the preparation of this work, we performed a detailed review of available data sets:
- UERRA is only available until July 2019. Its sensible and latent heat fluxes are only available in six-hourly intervals and cannot be used for our scientific analyses (https://confluence.ecmwf.int/display/UER/Issues+with+data).
- COSMO-REA6 (Bollmeyer et al., 2015) is only available from 1995-2019 and does neither make use of a sophisticated data assimilation scheme nor of an ensemble approach.
- CFSR (Schneider et al., 2013) is only available until 2010 and thus does not cover the recent climate change period.

Therefore, ERA5 provides the most advanced long-term high-resolution reanalyses data for Europe and there is no added value to our study by adding another reanalyses product. A short paragraph has been added to section 2.1 to clarify why we used ERA5 and no other reanalysis data sets.

The classification of summers in warm dry or wet was done using averages of the whole Europe as domain, although most of the time the authors only comment on the results over Germany and around countries (e.g. section 4.1.1). I think it would make more sense to classify the summer with the averages of an area centered in Germany avoiding the effect of the different regions in this classification and the different patterns that you obtain for the same category (e.g. warm and dry).

The selection of the domain also complicates the description of the results. The authors focus the analysis over Germany, thus in some part of the text they comment only results over Central Europe (e.g. section 4.1.1) and in other all Europe (e.g. section 3.2), which makes it very confusing because the patterns and results in Europe are specially diverse. I would recommend the authors to focus the analysis in central Europe, avoiding the discussion of north and south areas, or to divide the results section into paragraphs dedicated to the phenomena happening in each region.

Thank you for raising this point. We agree that it is beneficial to set the classification region to a smaller area. Therefore, we decided to exclude areas south of 40°N, north of 60°N, west of 5°W, and the areas east of 25°E in our analysis, reprocessed all the analyses, and modified the figures accordingly.

Right now, the structure of the results includes the explanation of each variable over the whole Europe is very difficult to follow, especially when the authors try to connect results between variables, because it is not clear to which region they are referring. Something similar happens with the selection of warm and dry summers. The chosen criterion leads to very different results for the same category (perhaps because of the spatial variability in the whole Europe used as average for the classification). And then the description of results seems incomplete since the explanations of the authors do not apply for all areas and all summers in the same category. The two results sections need to be revised and re-organize to improve the clarity of results for different regions and years. Perhaps a good idea is to show the results by year (including all maps of the same year in the same figure) and reduce the number of selected years so we can better follow the story that the authors are suggesting.

Following your suggestion, we reduced the number of years by setting a 2-m temperature anomaly median threshold of 0.5°C, i.e., for warm years the anomaly median of the domain needs to exceed the threshold. This together with focusing on central Europe slightly changes the selected of years in our analysis for the summer seasons. 1994 and 2013 dropped out and 2017 is added, i.e., 2003, 2006, 2015, 2017, 2018, 2019, 2020, 2021, and 2022 are classified as the warm and dry years. 2002 has now changed to a cold and wet year and is shown, together with 1997, in the supplement.

For clarity we reorganized section 4. We start with the terrestrial coupling strength. This is followed by the correlations between LH and SH in section 4.2. Section 4.3 shows our results of the ACI in connection with the LCL deficit.

The results (sections 3 and 4) are also not clear and not well supported. For example, the analysis is incomplete (e.g. the reader is sent to section 4.3.2 and 3.3 in lines 202 and 203 but they do not exist) or it includes wrong references (e.g. Figure 14 in line 300, line 303 referring to high SH in Fig. 10 when Fig10 shows correlations, and other missing references in the paragraphs like in lines 248 and 249 Fig 6 and Fig8 should be cited).

We carefully revised sections 3 and 4. This was also required due to the smaller domain of the analyses and focus on warm dry summers in the revision. We also added additional references the figures at the appropriate positions.

The results in section 3.0 are also supported by references mainly to abstracts in conferences (e.g. line 232) instead of on the results from the ERA5 product.

The references related to our C3S references are not conference contributions but press releases from official reports of the Copernicus Climate Change Service (C3S) published since 2017 online. Since 2019 they also have a doi. We therefore chose to change the references to the reports themselves 2021 and 2022 and refer to the report websites for 2017 and 2018.

The main justification for using ERA5 data for this analysis is that you have all the environmental variables more or less consistent with each other, so for example if in line 206 you are saying that the dry anomaly in summer is related to dry spring season you should support that with a map based on ERA5 in the supplementary information that supports that claim.

Thank you for your suggestion. We added the spring season soil moisture anomalies for the summer seasons to the supplementary material.

The same happens with other explanations of results based on some heavy precipitation or drought events (e.g. lines 169, 238, 353,357-359), are these events really represented in the ERA5 data? Because if not the results that we are seeing are not related to that event.

In figure R1 above we show exemplary for 2010 and 2003 the comparison of ERA5 and EOBS for large scale strong precipitation and drought summers. To show that ERA5 reasonably well simulates heavy precipitation and drought events, we added the ERA5 precipitation anomalies to section 3.3 of the revised manuscript.

[Figure]

**Figure R2.** ERA5 summer precipitation anomalies with respect to 1991-2020.

The patterns show that the events are presented in ERA5 in both, precipitation and soil moisture anomalies. In 2021 ERA5 shows stronger precipitation than EOBS in the region of Germany and Benelux that was hit by the severe large scale precipitation event in July (Mohr et al., 2023) .

Some of the claims based on the results are not easy to follow or see in the maps (e.g. line 200 "By comparing Fig. 6 and 7..."). Perhaps a statistical analysis of spatial correlations between variables could help to reach more robust conclusions.

We calculated the correlation coefficients between temperature and precipitation. In the analyzed reduced domain mentioned above, the correlation is always negative with values between -0.25 and -0.65 indicating that in most cases, a positive temperature and negative precipitation anomaly are associated with each other. This statement has been added to the new section 3.3.

Another example is in the paragraph starting in line 277, there are more coupling hot spots over central Europe for example in 2019 and 2006 but the soil moisture anomalies sometimes are negative and sometimes are positive. The authors should make an effort to explain the results that we are seeing or reduce the number of maps included in the manuscripts explaining the processes leading to warm conditions in particular years and areas.

The corresponding paragraph in section 4.3 now reads:

Coupling hot spots are observed over East and Southeast Europe with ACI values of more than 250 J kg$^{-1}$ (Fig. 8). They are connected to higher values of LH over these regions due to neutral or positive root zone soil moisture anomalies in 2006, 2019, 2020, and 2021 (Fig. 5). These coupling hot spots agree with the sensitivity between temperature and moisture change signals in Europe found by Jach et al. (2022).

Also the interpretation of land atmosphere coupling should be revised in the manuscript. I am not sure the authors explain clearly the role of soil moisture deficits in the restriction of latent heat flux and the induced increase in temperature. For example, this is the case in the mentioned paragraph (line 277), since both strong and weak coupling are associated with positive soil moisture anomalies. We need more information about what is happening there.

We added the following paragraph to our analysis in section 4.3:

"In case evapotranspiration is not limited by soil moisture, the incoming radiation is allowing for potential evapotranspiration and surface latent and sensible heat fluxes are partitioned accordingly. In case evapotranspiration is not limited by incoming radiation but by available soil moisture evapotranspiration is below the potential rate leading to higher Bowen ratios. The increasing Bowen ratio leads to an increase in temperature. This enhances evapotranspiration and therefore a gradual decrease in soil moisture towards wilting point. According to Benson and Dirmeyer (2021) this ultimately leads to the situation that latent heat fluxes almost vanish and the incoming radiation mainly transforms into sensible heat which can exacerbate heatwaves and droughts."

The conclusion sections could be reorganized, separating the discussion from the conclusions. In this way perhaps it is easier to identify the real conclusions of this study and the new information that we have learned about land-atmosphere coupling, which at the moment is not clear.

Following your suggestion, we separated the discussion from the summary to enhance readability of our manuscript.

Minor comments:

Line 111: "To complement our analysis, seasonal mean anomalies of 500 hPa geopotential (Lhotka and Kyselý, 2022) and ERA5 volumetric root zone soil moisture were calculated" This is not clear, do you mean geopotential height? Also are both variables calculated from ERA5 or do you use another database for the 500 hPa geopotential?

Both variables are calculated from ERA5. We will make it clear in the manuscript.

There are some minor spelling errors. The text should be revised (e.g. line 64 "suggests", line 75 "In the preceding…", line 213 no new paragraph, line 228 "The very…" is a very long sentence and the connectors are not used well, line 169 "caused by the severe'', line 189 "previous" no "precious", line 199 revise connectors, line 258 remove "in addition", line 294 "suggests", line 299 "is more often in…", line 302 "a considerable increase in the HLCL")

Thank you for spotting this. We will recheck grammar and spelling throughout the whole manuscript.

Please, correct the order of maps (2003 and 1994) in figure 4.

As the classification of the summer seasons has changed, Figure 5 now contains different years and in the correct order.

Line 236 "preventing a moisture limitation", please check this sentence. Do you mean here "leading to moisture deficits"?

We agree. We revised the sentence in the new section 4.2 according to your suggestion to: "During the most hot and dry summers 2003, 2018, and 2022, the correlation LH-SH became negative over Central Europe which is related to the anomalously warm and dry conditions during these seasons leading to a moisture deficit in the soil."

Line 254 "show non-significant values" please avoid the "significant" word if you did not apply a significant test. If you did apply a significant test, please provide the details on the text.

As we now focus our analysis on Central Europe, this part of the sentence relating to Spain and Portugal has been deleted. We did not apply any significance tests in our analysis.

Line 246 "almost weak" Please replace it by another expression e.g. "mostly weak" or just "weak".

As we removed the warm and wet summer seasons from our analysis, this error is not present anymore.

Line 249 "the average soil moisture availability" do you mean the "high soil moisture availability"?

Thank you for spotting this. Indeed, we meant "high soil moisture availability".

Fig 14, please add an explanation of why this selection of warm and dry years.

As our classification has changed to warm years only, we added all subfigures to this panel plot.

Section 6 should be section 5.

After the separation of the discussion and conclusions, the section numbering will be adjusted accordingly. The discussion is now section 5, and the summary is now the final section 6.

Paragraph starting in line 384, this is just the justification for using ERA5 in the analysis but not the discussion of how uncertainties in ERA5 like the overestimation of LH (Martens et al., 2020) could be affecting these results. This paragraph should probably be placed in section 2.1.

Following your suggestion, we moved the first half of this paragraph to section 2.1.

REFERENCES:

Dirmeyer, P. A., and Coauthors, 2018: Verification of Land–Atmosphere Coupling in Forecast Models, Reanalyses, and Land Surface Models Using Flux Site Observations. J. Hydrometeor., 19, 375–392, https://doi.org/10.1175/JHM-D-17-0152.1.

Martens, B., Schumacher, D. L., Wouters, H., Muñoz-Sabater, J., Verhoest, N. E. C., and Miralles, D. G.: Evaluating the land-surface energy partitioning in ERA5, Geosci. Model Dev., 13, 4159–4181, https://doi.org/10.5194/gmd-13-4159-2020, 2020.

Benson, D. O., and P. A. Dirmeyer, 2021: Characterizing the Relationship between Temperature and Soil Moisture Extremes and Their Role in the Exacerbation of Heat Waves over the Contiguous United States. J. Climate, **34**, 2175–2187, https://doi.org/10.1175/JCLI-D-20-0440.1.

Bollmeyer, C., Keller, J.D., Ohlwein, C., Wahl, S., Crewell, S., Friederichs, P., Hense, A., Keune, J., Kneifel, S., Pscheidt, I., Redl, S. and Steinke, S. (2015), Towards a high-resolution regional reanalysis for the European CORDEX domain. Q.J.R. Meteorol. Soc., 141: 1-15. https://doi.org/10.1002/qj.2486

Dirmeyer, P. A., Balsamo, G., Blyth, E. M., Morrison, R., & Cooper, H. M. (2021). Land-atmosphere interactions exacerbated the drought and heatwave over northern Europe during summer 2018. AGU Advances, 2, e2020AV000283. https://doi.org/10.1029/2020AV000283

Mohr, S., Ehret, U., Kunz, M., Ludwig, P., Caldas-Alvarez, A., Daniell, J. E., Ehmele, F., Feldmann, H., Franca, M. J., Gattke, C., Hundhausen, M., Knippertz, P., Küpfer, K., Mühr, B., Pinto, J. G., Quinting, J., Schäfer, A. M., Scheibel, M., Seidel, F., and Wisotzky, C., 2023: A multi-disciplinary analysis of the exceptional flood event of July 2021 in central Europe – Part 1: Event description and analysis, Nat. Hazards Earth Syst. Sci., 23, 525–551, https://doi.org/10.5194/nhess-23-525-2023

Muñoz-Sabater, J., Dutra, E., Agustí-Panareda, A., Albergel, C., Arduini, G., Balsamo, G., Boussetta, S., Choulga, M., Harrigan, S., Hersbach, H., Martens, B., Miralles, D. G., Piles, M., Rodríguez-Fernández, N. J., Zsoter, E., Buontempo, C., and Thépaut, J.-N.: ERA5-Land: a state-of-the-art global reanalysis dataset for land applications, Earth Syst. Sci. Data, 13, 4349–4383, https://doi.org/10.5194/essd-13-4349-2021, 2021.

Qi, Y., Chen, H., & Zhu, S. (2023). Influence of land–atmosphere coupling on low temperature extremes over southern Eurasia. Journal of Geophysical Research: Atmospheres, 128, e2022JD037252. https://doi.org/10.1029/2022JD037252

Rousi, E., Kornhuber, K., Beobide-Arsuaga, G. et al. Accelerated western European heatwave trends linked to more-persistent double jets over Eurasia. Nat Commun **13**, 3851 (2022). https://doi.org/10.1038/s41467-022-31432-y

Rousi, E., Fink, A. H., Andersen, L. S., Becker, F. N., Beobide-Arsuaga, G., Breil, M., Cozzi, G., Heinke, J., Jach, L., Niermann, D., Petrovic, D., Richling, A., Riebold, J., Steidl, S., Suarez-Gutierrez, L., Tradowsky, J. S., Coumou, D., Düsterhus, A., Ellsäßer, F., Fragkoulidis, G., Gliksman, D., Handorf, D., Haustein, K., Kornhuber, K., Kunstmann, H., Pinto, J. G., Warrach-Sagi, K., and Xoplaki, E.: The extremely hot and dry 2018 summer in central and northern Europe from a multi-faceted weather and climate perspective, Nat. Hazards Earth Syst. Sci., 23, 1699–1718, https://doi.org/10.5194/nhess-23-1699-2023, 2023.

Schneider, D. P., C. Deser, J. Fasullo, and K. E. Trenberth, 2013: Climate Data Guide Spurs Discovery and Understanding. Eos Trans. AGU, 94, 121–122, https://doi.org/10.1002/2013eo130001

Sun, G., Z. Hu, Y. Ma, Z. Xie, F. Sun, J. Wang, S. Yang, 2021: Analysis of local land atmosphere coupling characteristics over Tibetan Plateau in the dry and rainy seasons using observational data and ERA5, Science of The Total Environment, **774**, 145138, https://doi.org/10.1016/j.scitotenv.2021.145138.

---

## Author Comment (AC2)

Schwitalla et al. present a study on the coupling strength of moisture and energy fluxes, and atmospheric characteristics over the European continent. The goal is to understand interanual variability of the LA coupling sign and strength in different climatic regions under varying moisture and energy conditions in summer. Focus is placed on the drought conditions. As a reference they chose the climate period 1991 to 2020, while the time series investigated extends to 2022.

We thank you for reviewing our manuscript. Please find our responses to your comments as marked in blue. Following the suggestion of Reviewer #1, our investigation now focusses on the area between 5°W-25°E and 40°N-60°N.

In the quantification of the coupling sign and strength, they use standard indices and correlation coefficients based on linearity assumptions.

We apply the terrestrial and atmospheric coupling indices, which are no standard indices yet to study land-atmosphere coupling. So far, they were not applied on the regional scale in Europe except for one growing season in Germany by Warrach-Sagi et al., 2022. The indices consider, that coupling experiences thresholds, e.g., in case of soil moistures above field capacity, LH is not limited by soil moisture and the no terrestrial coupling due to soil moisture via LH is visible. The study is complemented by correlation of SH and LH, which is applied to gain more information on the energy partitioning role on the coupling (e.g., Knist et al., 2017). In case they are not correlated other processes than LA coupling via soil moisture determines the fluxes and thresholds in moisture or energy limitations may be reached.

A plethora of work has been on published on the sign and strength of LA coupling in the past couple of decades. The study corroborates previous findings; in my opinion there are no surprises, or perhaps I have missed them. In this case, the authors need to revise the manuscript and clearly point out the new findings.

Though multiple studies assessed land-atmosphere feedback during the past two decades, still a huge research gap, particularly for Europe and for the time periods, which are already affected by the climate crises such as the droughts and floodings in the summers since 2015. In the past decades for Europe on the regional scale below 50 km resolution land-atmosphere coupling studies were often based solely surface variables (e.g. Knist et al., 2017). Only recently the regional studies also considered the development and state of the atmospheric boundary layer (e.g. Jach et al., 2020). The existing studies focus on (Central) Europe up to 2015 (e.g., Knist et al., 2017, Jach et al., 2020; Leutwyler et al., 2021) or on a single summer season (e.g., Dirmeyer et al., 2021). Between 2015 and 2022 Europe experienced a strong increase in summer temperature, droughts and heavy precipitation, this is therefore a very interesting period to add in land-atmosphere feedback studies in central Europe. A review of the current state of the art and the studies is included in the introduction from line 33 to line 74.

The analyzed period includes the decades that have been studied previously with respect to land-atmosphere coupling based on regional climate model simulations (e.g. Knist et al., 2017; Jach et al., 2020; Miralles et al.,2019). All other studies were based on the coarse grids of global climate models. It would be questionable if our results would not support previous findings on the same period. Here we extend the study to the current rapid climate change of the past decade. Further the results with ERA5 are closer to observations than regional climate models (Sun et al.,2021, Qi et al.,2023, and Rousi et al.,2023).

Our results show that the extreme drought years 2003, 2018, and 2022 can be identified as changing soil moisture-atmospheric coupling pattern. This in turn leads to a decoupling between SH and LH as shown by the correlation between these two variables. Additionally, the LCL deficit considerably

increases during these years further enhancing and amplifying the heat and drought situation. As shown in Benson and Dirmeyer (2021) this can lead to a self-reinforcing mechanism which even further amplifies heat and drought conditions in changing climate.

In the summary and discussion section, the authors touch on the main the goal of the study and many interesting questions. For example, the authors state that "the hydroclimatological conditions during each summer drive considerable interannual variability in LA coupling…". This would indeed be an interesting finding indeed. However, in my opinion, the analyses does not show this in a rigorous way. The indices of different years are presented, without further analysess.

We have to admit that the wording "hydroclimatology" could be misleading here as we did not investigate the relation between temperature, precipitation, and streamflow. Here we meant that the interannual temperature and soil moisture (and thus precipitation) variability. This sentence has been changed and now reads:

"Our results show that the interannual temperature and soil moisture variability during the different summers considerably  drive the interannual variability in LA coupling over Central Europe."

They also suggest "a growing influence of soil moisture variability on the meteorological conditions…" in the second half of the study period, which was drier than the first half. Again, the presentation of the coupling indices and linear correlations for individual years does not afford this conjecture in my opinion.

Markonis et al. (2021) found a considerable increase of drought events since 2010 due to higher temperature, less precipitation and the resulting soil moisture depletion already occurring in spring. We added this reference to the second last paragraph of our summary.

Two additional points that caught my attention are the categorization of the different years and application of the ERA5 data set (which was also brought up by the other reviewer). In the former, the classification appears to be rather arbitrary.

The years 2021 may server as an example, which is categorized as a warm and dry year in the table, but exhibits a wet anomaly and is referred to as warm and wet in the text, if I am not mistaken. This type of confusion does not lend confidence in the results.

Indeed, the definition of 2021 in Table 1 was wrong. 2021 shows a precipitation dry bias connected to a warm bias. We will carefully check all summer seasons again to ensure a correct classification.

Following the suggestion of Reviewer #1, we decided to focus our analysis to a smaller region between 5°W-25°E and 40°N-60°N. In addition, we only consider only warm summer seasons where the median 2-m temperature anomaly exceeds +0.5 K.

In the latter, the issue of data assimilation in ERA5 in the diagnosis of LA coupling has to be discussed further. Also in my opinion, reanalyses are of limited value in feedback studies, which leads to the challenge of identifying feedbacks in simulations while reproducing real world weather conditions.

Reanalyses data provide the three-dimensional consistent land and atmosphere gridded multidecadal timeseries of diurnal cycles closest to observations and therefore allow to study on the regional climate scale LA coupling beyond studying surface variables. It is not the goal of this study to evaluate reanalysis data or a model simulation. It is the goal to study the land-atmosphere interaction based on the best available 3-dimensional long-term gridded data set for Europe which is also applied in very recent studies of Rousi et al., 2022 and Rousi et al., 2023 to investigate feedback effects between the atmosphere and the land surface.

Apart from the 12-hourly atmospheric data assimilation (Hersbach et al., 2020), a sophisticated Kalman Filter based soil moisture assimilation is applied hourly (de Rosnay et al., 2013) in ERA5 which connects the atmosphere with the soil during the subsequent forecasts. Martens et al. (2020) clearly showed that ERA5 outperforms its predecessor ERA-I with respect to surface fluxes which was confirmed by a study of Muñoz-Sabater et al. (2021).

As shown in Muñoz-Sabater et al. (2021), the representation of the surface fluxes, soil moisture and net radiation in ERA5 is very reasonable compared to in-situ measurements over Europe as well as in comparison with data from GLEAM project (Miralles et al., 2011). Dirmeyer et al. (2021) successfully applied ERA5 for the investigation of LA feedback processes during the severe summer drought over Europe in 2018. The ERA5 variables used in this study (soil moisture, 2-m temperatures, surface sensible and latent heat flux) were successfully applied in LA feedback studies of Sun et al. (2021), Qi et al. (2023), and Rousi et al. (2023).

To summarize, ERA5 data deliver the required 3D data to apply LA feedback metrics that combine the variables of our study. We added the discussion accordingly to the manuscript.

Perhaps one has to make a choice and accept that in case of feedback studies in order to identify mechanisms, internal model consistency is more important than reproducing past weather. It would be interesting to understand the perspective of the authors in a more in depth discussion.

We disagree with this statement. An improvement of the representation of metrics must directly propagate in improved forecasts and vice versa. As explained above, it is not the goal of this study to evaluate reanalysis data or a model simulation. It is the goal to study the land-atmosphere interaction based on the best available 3-dimensional long-term gridded data set with diurnal cycles for Europe. We study the feedback based on the variables, not in the process chain calculated in parameterizations of the surface and boundary layer and within the land surface model. That would require model simulations, high resolution vertical and horizontal observations and would answer different research questions (e.g., Milovac et al., 2016 and Bauer et al., 2023).

Further, global and regional climate model simulations suffer from biases in the representation of surface fluxes, e.g., due to an inaccurate simulation of precipitation and thus an improper simulation of (root zone) soil moisture (Diez-Sierra et al., 2022). Although the general climate change signal can still be present in the simulations (Ban et al., 2022), even a bias correction of precipitation would not be sufficient in this case as this does not impact other prognostic variables.

A short paragraph has been added to section 2.1 why we used ERA5 and no other reanalysis data sets.

In my opinion, the study requires much more work beyond major revisions in order to contribute new and interesting results addressing the important issue of interannual variability of LA coupling in summer.

As stated above, between 2015 and 2022 Europe experienced a strong increase in summer temperature, droughts and heavy precipitation. This is therefore an interesting period to add in land-atmosphere feedback studies over Central Europe. Studies on LA feedback that also use data in the atmospheric boundary layer are either based on reanalyses data or model simulations and limited for Europe. Europe in the past has not been a hot spot, however this may now change as our study shows for the last decade. The problems of the purely model simulation-based studies are outlined above. The reanalyses data is suitable for the metrics. The results of the metrics can be used to study the capability of regional and global models "to identify mechanisms, internal model consistency."

We are confident that we sufficiently addressed all your concerns mentioned above so that the manuscript can now be published.

References:

Ban, N., Caillaud, C., Coppola, E. et al. The first multi-model ensemble of regional climate simulations at kilometer-scale resolution, part I: evaluation of precipitation. Clim Dyn **57**, 275–302 (2021). https://doi.org/10.1007/s00382-021-05708-w

Bauer, H.-S., Späth, F., Lange, D., Thundathil, R., Ingwersen, J., Behrendt, A., & Wulfmeyer, V. (2023). Evolution of the convective boundary layer in a WRF simulation nested down to 100 m resolution during a cloud-free case of LAFE, 2017 and comparison to observations. Journal of Geophysical Research: Atmospheres, 128, e2022JD037212. https://doi.org/10.1029/2022JD037212

Benson, D. O., and P. A. Dirmeyer, 2021: Characterizing the Relationship between Temperature and Soil Moisture Extremes and Their Role in the Exacerbation of Heat Waves over the Contiguous United States. J. Climate, **34**, 2175–2187, https://doi.org/10.1175/JCLI-D-20-0440.1.

de Rosnay, P., Drusch, M., Vasiljevic, D., Balsamo, G., Albergel, C. and Isaksen, L. (2013), A simplified Extended Kalman Filter for the global operational soil moisture analysis at ECMWF. Q.J.R. Meteorol. Soc., 139: 1199-1213. https://doi.org/10.1002/qj.2023

Diez-Sierra, J., and Coauthors, 2022: The Worldwide C3S CORDEX Grand Ensemble: A Major Contribution to Assess Regional Climate Change in the IPCC AR6 Atlas. Bull. Amer. Meteor. Soc., **103**, E2804–E2826, https://doi.org/10.1175/BAMS-D-22-0111.1

Dirmeyer, P. A., Balsamo, G., Blyth, E. M., Morrison, R., & Cooper, H. M. (2021). Land-atmosphere interactions exacerbated the drought and heatwave over northern Europe during summer 2018. AGU Advances, 2, e2020AV000283. https://doi.org/10.1029/2020AV000283

Hartick, C., Furusho-Percot, C., Clark, M. P., & Kollet, S. (2022). An interannual drought feedback loop affects the surface energy balance and cloud properties. Geophysical Research Letters, 49, e2022GL100924. https://doi.org/10.1029/2022GL100924

Markonis Y., Kumar R., Hanel M., Rakovec O., Máca P., AghaKouchak A. (2021): The rise of compound warm-season droughts in Europe. Sci Adv.,Feb 3;7(6):eabb9668. doi: 10.1126/sciadv.abb9668. PMID: 33536204; PMCID: PMC7857689.

Milovac, J., K. Warrach-Sagi, A. Behrendt, F. Späth, J. Ingwersen, and V. Wulfmeyer (2016), Investigation of PBL schemes combining the WRF model simulations with scanning water vapor differential absorption lidar measurements, J. Geophys. Res. Atmos., 121, 624–649, doi:10.1002/2015JD023927.

Miralles, D.G., Gentine, P., Seneviratne, S.I. and Teuling, A.J. (2019), Land–atmospheric feedbacks during droughts and heatwaves: state of the science and current challenges. Ann. N.Y. Acad. Sci., 1436: 19-35. https://doi.org/10.1111/nyas.13912

Muñoz-Sabater, J., Dutra, E., Agustí-Panareda, A., Albergel, C., Arduini, G., Balsamo, G., Boussetta, S., Choulga, M., Harrigan, S., Hersbach, H., Martens, B., Miralles, D. G., Piles, M., Rodríguez-Fernández, N. J., Zsoter, E., Buontempo, C., and Thépaut, J.-N.: ERA5-Land: a state-of-the-art global reanalysis dataset for land applications, Earth Syst. Sci. Data, 13, 4349–4383, https://doi.org/10.5194/essd-13-4349-2021, 2021.

Jach, L., K. Warrach-Sagi, J. Ingwersen, E. Kaas, and V. Wulfmeyer, 2020: Land cover impacts on land-atmosphere coupling strength in climate simulations with WRF over Europe. J. Geophys. Res-Atmos. 125(18), 1-21. DOI:10.1029/2019JD031989

Leutwyler, D., A. Imamovic, and C. Schär, 2021: The Continental-Scale Soil Moisture–Precipitation Feedback in Europe with Parameterized and Explicit Convection. *J. Climate*, **34**, 5303–5320, https://doi.org/10.1175/JCLI-D-20-0415.1.

Martens, B., Schumacher, D. L., Wouters, H., Muñoz-Sabater, J., Verhoest, N. E. C., and Miralles, D. G.: Evaluating the land-surface energy partitioning in ERA5, Geosci. Model Dev., 13, 4159–4181, https://doi.org/10.5194/gmd-13-4159-2020, 2020.

Qi, Y., Chen, H., & Zhu, S. (2023). Influence of land–atmosphere coupling on low temperature extremes over southern Eurasia. Journal of Geophysical Research: Atmospheres, 128, e2022JD037252. https://doi.org/10.1029/2022JD037252

Rousi, E., Fink, A. H., Andersen, L. S., Becker, F. N., Beobide-Arsuaga, G., Breil, M., Cozzi, G., Heinke, J., Jach, L., Niermann, D., Petrovic, D., Richling, A., Riebold, J., Steidl, S., Suarez-Gutierrez, L., Tradowsky, J. S., Coumou, D., Düsterhus, A., Ellsäßer, F., Fragkoulidis, G., Gliksman, D., Handorf, D., Haustein, K., Kornhuber, K., Kunstmann, H., Pinto, J. G., Warrach-Sagi, K., and Xoplaki, E.: The extremely hot and dry 2018 summer in central and northern Europe from a multi-faceted weather and climate perspective, Nat. Hazards Earth Syst. Sci., 23, 1699–1718, https://doi.org/10.5194/nhess-23-1699-2023, 2023.

Sun, G., Z. Hu, Y. Ma, Z. Xie, F. Sun, J. Wang, S. Yang, 2021: Analysis of local land atmosphere coupling characteristics over Tibetan Plateau in the dry and rainy seasons using observational data and ERA5, Science of The Total Environment, **774**, 145138, https://doi.org/10.1016/j.scitotenv.2021.145138.

---

## Referee Report (RR1)

**Soil moisture—atmosphere coupling strength over Central Europe in the recent warming climate**

Submitted for publication by Thomas Schwitalla, Lisa Jach, Volker Wulffmeyer and Kirsten Warrach-Sagi

**General comments**

*I am reviewing a revised version of the manuscript. As I have not seen the manuscript before, I comment on both the author responses of the previous review and the manuscript.*

The authors have provided point-by-point responses to comments raised by two referees. The coverage is comprehensive and the manuscript was amended accordingly. One key-point that has been raised by both referees is the choice of datasets (ERA5 + E-OBS) in relation to the scope of the study: RC1 and RC2 raise concerns regarding consistency of the reanalyzed model state and true precipitation fields and demand further explanation. While the authors have provided an analysis based on ERA5 precipitation (which is supposed to be a more consistent approach), the key points raised by RC2 are not covered adequately in my opinion:

- The fundamental question, 'what is the value of a feedback study based entirely on model data'? Is not answered. In other words, it remains unclear, what exactly is under investigation here: LA coupling in the ERA-5 reality (i.e. the ERA-5 land surface model and its interplay with the atmosphere), the actual physical coupling of the real atmosphere with the real surface or something hybrid in between?

- The authors advocate a rather simplistic view of model performance and representation of reality which I consider critical in this context: "An improvement of the representation of metrics must directly propagate in improved forecasts and vice versa". In general, this is not true: Sometimes, in particular with respect to the nonlinearity in surface-layer models, improvements in some metrics cause performance degradation, in particular for coupled set-ups where error-compensation is likely to occur.

After evaluating the manuscript on my own, I further identified related major issues that touch upon the core of the manuscript, including the definition and description of coupling metrics, selection of data and sampling issues (see major issues). These need to be covered for the manuscript to become publishable. I leave it to the editors whether this happens within major revisions of the manuscript or is better carried out outside the peer-review process which would result in a rejection.

**Major issues**

**M1** **The concept of land—atmosphere (LA) coupling as it is introduced in the manuscript is improper and incosistent.**

a) LA coupling is not trivially determined and I would wish for a framing of how it is defined/quantified (namely: by means of co-variability) already in the abstract as the determination of coupling is central to the entire manuscript. Regarding the definition of coupling by covariability, there is further the major

caveat of correlation vs. causality. Further, the definition of of coupling in terms of magnitude of the co-variability may hide compensating coupling mechanisms. A possible way to come around these conceptual limitations is considering surface-layer fluxes (as many of the studies cited in the manuscript do) and I urge the authors to also look at the fluxes acting as physical mediators between the land and atmosphere properties.

b)   The concepts of LA coupling and LA feedback are not the same (I understand that a coupling can also have dampening effect versus a feedback that is strictly amplifying) and the authors should exert greater care when defining these two concepts in lines 30-50 to avoid ambiguities.

c)   Definition of the coupling indices (Eq. 1) is incomplete. Operator σ is not defined; Variables used in calculating the ACI are not given explicitly.

d)   The analysis is constrained to approximate day time (06 – 18 UTC). While it has been shown that night-time failure of numerical models has little impact on convective boundary layer during the next day for idealized cases (van Stratum and Stevens, 2015), constraining of the LA coupling to daytime only is certainly biased and the impact on the results should not be underestimated as evidence from measurement shows that day-time and night-time effects compensate to a large degree. The fact that incorporating nighttime results into the analysis has "detrimental impact" (l. 150) may simply demonstrate that the coupling effects are far less important than the authors suggest as indeed much of the positive coupling feedback during daytime is reversed at night, which is a physical aspect of the problem.

**M2   The role of artificial and self-correlations is largely ignored.**

a)   As an example, consider TCI (according to Eq. 1): Latent heat flux crucially depends on soil-moisture availability (most likely via parameterization of the Bowen ratio). An index defined on modelling data as suggested by Eq. 1 suffera from two levels of unwanted correlation: First, it will reproduce the Bowen-ratio parametrization. Second, LH depends on soil-moisture and thus on the surface-layer closure. I wonder to what extent, we look at an analysis of the surface-layer and Bowen-ratio closures rather than a real physical-coupling analysis. This can easily be checked by evaluating the TCI theoretically based on the underlying parameterizations which will serve as a baseline. It would be fair to talk about physical coupling only whenever this baseline correlation is exceeded.

**M3   Statistics on [very] small samples**

a)   The reference period (30 years) is further split up into wet, dry, warm and cold summers which results in an extremely small sample size. In my view, this makes this study more a multi-case study rather than an assessment of climatological shifts as the abstract claims. While both approaches have merit, in my opinion the general conclusions that can be drawn from such a small sample (where individual events like the anomalous summer of 2021 etc.) do not reach as far as the authors suggest. The results, discussion and conclusion sections should be amended accordingly.

b)   The findings presented in the discussion section are actually interesting, but I do not see how – in the generality suggested by the authors – they can be hold up in view of the diverse results. It would be interesting to underpin these by a quantitative analysis (for instance a pdf analysis that strips of the geo-location and would isolate the physical processes at the core of coupling). [In this way, the sample size would be increased by also sampling space, on top of time!]

**M4   Presentation of results**

The number of figure panels is not appropriate given the limited interpretation. I suggest to rather focus on individual representative figures and give them more space rather than reproducing patterns for each year

and parameters. If relevant at all, they could be given in the supplement (as is already the case for some parameters).

**M5    The analysis is missing the final link from the surface to the atmosphere**

The authors claim to investigate land—atmosphere coupling. In fact, they, however, separately assess the terrestrial coupling (soil-to-surface layer) and the atmospheric coupling (surface-layer to atmosphere); the final link that one would expect based on title, abstract, and introduction (soil – atmosphere) is neither carried out nor discussed in the analysis.

**Minor issues**

- Abstract: "[…] for exacerbating durations": inappropriate verb (surface conditions may also act in the other direction + bad style → please rewrite!

- l. 106-111: I do not see why it is necessary to discuss the disadvantages of other reanalyses not used here. It diverts from the main path.

- The choice and weighting of the layers used for root-zone soil moisture has crucial impact on the time scales for which coupling can be investigated. In fact, it can be shown that higher weights on deeper layers act like a filter (Liu and Shao, 2013; van der Linden, 2022). This needs to be discussed an it should be carefully checked if the weighting and choice of layers selects appropriate time scales.

- The extensive discussion of TCI and introduction of SH—LH correlation belongs to the introduction rather than to the results section.

- Using CAPE as coupling metric is rather dangerous as it is a potential (and not an eventually released) kind of energy. The huge values over Easter Europe probably result from the fact that in these dry regions, convective inhibition is large and CAPE often "lives" for a relatively longer period of time.

- Fig. 9 and associated text: If I understand correctly, the LCL "deficit" is not really a deficit, but an anomaly.

- l. 389-394: This is not related to the present work and should either be removed or moved to the introduction.

**Technical comments**

- l. 76: "The in the preceding paragraph described" → The preceding paragraph describes

- l. 118: remove "seasons"

- l. 121-123: move predicate before object in the sentence!

- l. 264: why "could be"? Why spectulate? This can easily be checked based on available data

- l. 341: remove second period ('.').

- l. 342: "2018 [...]" → "The year 2018 [...]"

**Referencens**

van der Linden, S.J.A., Kruis, M.T., Hartogensis, O.K. et al (2022): Heat Transfer Through Grass: A Diffusive Approach. *Boundary-Layer Meteorol* **184**, 251–276 https://doi.org/10.1007/s10546-022-00708-7

Liu, S and Shao Y. Soil-layer (2013): configuration requirement for large-eddy atmosphere and land surface coupled modeling (2013). *Atmospheric Science Letters* **14**, 112–117 https://doi.org/10.1002/asl2.426

van Stratum, B and Stevens, B (2015): The influence of misrepresenting the nocturnal boundary layer on idealized daytime convection in large-eddy simulation. *J Adv Model Earth Syst* **7(2)**, 423–436 https://doi.org/10.1002/2014MS000370

---

## Referee Report (RR2)

**General overview**

The authors presented a valuable analysis focused on the influence of soil moisture on the land surface energy partitioning throughout Europe. They went a step further by providing an analysis of the consequences promoted by soil dry-out on the dynamics of the atmosphere, revealing a potential re-amplification of the LA feedbacks through the suppression of convection and further soil desiccation. In fact, there's a lack of studies performing an aggregation of several case studies, allowing for a detailed comparison of these LA feedbacks between different episodes characterized by the co-occurrence of extreme hot and dry conditions. This allows to get a time and spatial overview of the differences in the LA dynamics between episodes and I felt, particularly in the discussion chapter, that the authors missed the opportunity to underline and to explore more deeply the results in this context.

I think that a more detailed spatiotemporal integrated analysis under a climate change context is missing in the analysis (see major comments). It would be interesting to add some information about historical long-term changes in these LA feedbacks. The analysis was carried out using state of the art datasets and appropriate metrics. Thus, in terms of data/methods and the general conceptualization of the research problem I'm confident in the interest and robustness of the results presented here. As highlighted on the several following comments, my main concerns are focused on the poor quality of the English writing over some sections of the manuscript. The text has several typos and the English syntax is far from being ideal for a high-quality scientific publication. In addition, the author's argumentation is often poorly structured and presented in a confusing way. Therefore, a great effort should be made in order to improve the way authors expose their ideas and communicate with the reader. Moreover, the manuscript text often contains acronyms, abbreviations, scientific terms or metrics that are not properly defined in the data and methods sections. With this being said, I think the analysis has potential to be published, but only after authors have correctly addressed the following comments. This is my first round of reviews as I would like to check the manuscript once again to filter out other minor issues.

**Major Comments**

- My first major comment concerns the way the manuscript is structured. The section n. º 3 named as "Summer Season anomaly maps", presents the anomaly fields for several land surface and meteorological variables. These are, in fact, results that came from the analyses and they represent outcomes obtained by the authors. This section of the manuscript should therefore be included in the Results chapter. There's no reason to define a whole new chapter to present these findings.

- Throughout the manuscript, the argumentation, interpretation and discussion of the results is presented in a very vague an unclear way. An effort must be made in order to use a more appropriate scientific language and to adopt the correct scientific terms used in the literature to describe the processes and metrics under

consideration by the authors. More details about this aspect can be found in the minor comments below.

- Throughout the manuscript several abbreviations and acronyms used by the authors are not defined in the Material and Methods section (e.g. TCI η-LH; CTP-HI$_{low}$; TLCI). In fact, this section of the manuscript needs some adjustments to define more clearly the several metrics adopted in the analysis. For example, the ACI was computed using two different approaches: one using CAPE and the other using HLCL. However, the way the authors distinguish both throughout the manuscript text is far from being the ideal and it brings some confusion to the narrative. I suggest similarly to TCI, to present the ACI in its mathematical equation form, defining two new abbreviations to the different approaches: ACI$_{CAPE}$ for the first, and ACI$_{HLCL}$ for the second.

- As I'm sure the authors know, the term LA feedbacks addresses several processes between the land surface and the atmosphere that describe the connection between soil moisture and precipitation, evaporation, temperature and even other meteorological parameters. The analysis is focused on the inter-link between soil moisture and temperature through perturbations in the turbulent fluxes of latent and sensible heat. Thus, I'm not entirely confident on the use of such a generic term to describe such a particular process. I suggest considering another term such as "soil moisture–temperature coupling" and keep land–atmosphere feedback for the link between soil moisture and CAPE or the link between soil moisture and HLCL.

- Have the authors considered using ERA5-Land or even GLEAM instead of ERA5 to obtain soil moisture data? ERA5-Land and GLEAM are forced by meteorological fields provided by ERA5 and so a potential problem associated with some inconsistency in the data source would be avoided. In addition, ERA5-Land and GLEAM incorporate land surface models capable to improve the representation of the water and energy cycles over land, contributing to a better simulation of land surface variables (Muñoz-Sabater et al. 2021). Beck et al. (2021) evaluated the temporal dynamics of 18 state-of-the-art (quasi-)global near-surface soil moisture products, and concluded the following: "The ERA5 reanalysis, which assimilates ASCAT soil moisture (Hersbach et al., 2020), obtained a lower overall performance (median R = 0.68) than the open-loop models ERA5-Land (median R = 0.72) and HBV-ERA5 (median R = 0.74), which were both forced with ERA5 precipitation (Fig. 2a). This suggests that assimilating satellite soil moisture estimates (ERA5) was less beneficial than either increasing the model resolution (ERA5-Land) or improving the model efficiency (HBV)."

- The Figures already present an overall good quality. However, some extra adjustments would be welcome. The size of the panels could be increased a bit

more by removing the latitude and longitude ticks that are repeated unnecessarily in all panels (I would only keep them in the first panel). A larger font size would also be a good idea, especially for the panel title.

- Finally, I'm not sure if the authors, with results here presented, have successfully achieved the main objectives stated for the study: "this study investigates interannual variability of LA coupling strength"; "In this study, we therefore assess the temporal variability of LA coupling of the European summer seasons 1991- 2022 on the interannual time scale"; "This paper describes the variability of the LA coupling strength of the warm summer seasons 1991-2022 which became the dominant situation over Europe since 2010". Basically, the authors repeatedly stressed that the main goal of this study is to provide an interannual characterization of the LA coupling variability. However, in practice, what they presented here is an analysis focused on nine separated case study periods from a total of 32 years of data. This is not an interannual analysis even more when a time aggregation of these 9 years is lacking in the discussion. A narrative going, chronologically, throughout these 9 summer periods highlighting the effects of climate change, is not presented by the authors, which could be interesting. I think the authors have two options here: either they reformulate clearly the objectives of the analysis or they include a pure interannual analysis with a year-by-year evolution of the LA coupling and associated meteorological variables. The second option would be much more interesting, as it would also allow to get a temporal integrated overview of these parameters and see any possible trends in the soil moisture–temperature regimes throughout Europe and under a climate change context.

**Minor Comments**

**Lines 23-24:** Please change accordingly: "In the last decades, Europe experienced severe drought periods and heatwaves (WMO, 2015; C3S, 2018; Markonis et al., 2021; WMO, 2022a)  with 2022 being the hottest summer ever recorded over Europe (WMO, 2022a)."

**Lines 29-30:** "(…) who suggest that these extreme conditions will be more likely under climate change conditions where two out of three summer seasons will experience hot and dry conditions." This sounds a bit vague… This increase in hot and dry conditions under climate change conditions is estimated to occur for which period? Near future, far future? Is already happening? Authors should clarify.

**Line 30:** Please change accordingly: "(…) midlatitudes due to the occurrence of **a** double-jet stream **configuration associated to atmospheric blocking conditions**  (Kornhuber et al., 2017).

**Line 56:** What is the CTP-HI$_{low}$ framework? More information should be given here.

**Line 63**: Change accordingly: "soil moisture-temperature feedback was a key (…)".

**Lines 76-78:** It reads weird: "The in the preceding paragraph described shifts in the hydrological conditions from energy- to moisture-limited conditions originating from droughts and heatwaves (Dirmeyer et al., 2021; Duan et al., 2020) or severe flooding (Lo et al., 2021) imply temporal variability in LA coupling at sub-seasonal to interannual time-scales." Please rephrase it.

**Lines 82-84:** Please rephrase it to something like "However, a quantification of **the** temporal variability in different coupling relationships **and the associated impacts**  of the impact of the variability remain **still lacks**, as LA coupling strength **on other times scales than climate period**  **has been** barely investigated over Europe, so far".

**Lines 162-165:** The sentence is too long and confuse. Consider changing to something like: "As shown in Table 1 the warm and dry summer seasons **have** become **predominant**  since 2015**. This has been** associated with a strong reduction in annual and seasonal precipitation, combined with a reduced atmospheric water availability that led to a constant decline of the root zone soil moisture and**, thus, to** an agricultural drought  (van der Wiel et al., 2022)."

**Lines 166-169:** It reads weird. Consider changing to: "The following sections **present an analysis of the anomaly fields of**  ERA5 500 hPa geopotential, 2-m temperature**,** root zone soil moisture η **and**  precipitation **for the summer seasons chosen for evaluation (Table1)**.

**Line 175:** Change accordingly: "(…) However, the summer **seasons of** 2015 and 2020 are exceptions (…)". However, Figure 1 clearly shows, in contrast with 2020, that most of Europe was covered by exceptional high values of 500hPa geopotential during 2015. Can you please clarify this? Authors also wrote the following: "In 2015, a pronounced north-south anomaly gradient is visible with negative values over the British Isles and Scandinavia while in 2020 the 500 hPa geopotential is only slightly above the average 1991-2020". This is true, but a similar pattern is also observed during 2017… Why authors did not mention 2017 and highlight the similarities with 2015? Finally for both 2015 and 2017 summers, I'm not sure if a north-south anomaly gradient is the most accurate way to describe the geopotential anomalies… I would prefer to describe them as concentric nucleus of positive anomalies located over Central Europe associated to a strong meridional gradient of 500hPa Geopotential.

**Line 177:** "(…) while in 2020 the 500 hPa geopotential is only slightly above the average 1991-2020 **(bottom right panel in Fig. 1)**". The bottom right panel in Fig.1 shows the anomaly pattern during 2021… In order to avoid this and to help establish a better link

between the text and the figure, I recommend the authors to label with letters (a,b,c,d…) the several panels. This works for all the other figures in the manuscript.

**Line 182-183:** Change accordingly: "tThe highest 2-m temperature anomalies were present in **observed during the summers of** 2003, 2018, 2019, and 2022 (Fig. 2) which Is **and were spatially** associated with strong positive geopotential anomalies over Central Europe. **The summer of** 2022 was the hottest ever recorded so far (C3S, 2023)".

**Line 183 and 185:** This comment works for all the other sections of the manuscript where similar issues are observed. "**2022** was the hottest summer ever recorded so far (C3S, 2023). During 2006, the 2-m temperature anomalies **are** highest north of 51°N 185 while in 2017, the highest temperature anomalies **were** observed south of 50°N as the maximum geopotential anomaly is shifted to the north and south, respectively". It's not correct to start a sentence with 4 numeric digits… It would be more proper to start with something like "The summer of 2022…". Keep in mind that the analysis is focused only for the summer seasons, so make sure when analyzing and discussing the results that you're referring to the summer periods. Also, two different verbal tenses are used in the same sentence, which is not correct. In fact, the authors should adopt, when describing the results, the same verbal tense.

**Lines 201-203:** This sounds too vague… More information about the way these correlations were obtained (time and space dimensions) and what they exactly mean should be provided by the authors.

**Lines 209-210:** It would be interesting to represent in an extra panel the time series of the daily spatially averaged values of soil moisture over Europe from the early spring until the late summer time for all the summer seasons. It would allow to better catch in the results this effect that the authors are mentioning here. This new panel could be included not just here in Figure 5, but also in all the other figures.

**Lines 212-214:** "By using the median of the soil moisture anomalies, 2006 largely is an average summer with moderate positive anomalies over East Europe while 2015 and 2017 on average show moderate dry soil anomalies". Poor English syntax. Please rephrase it

**Lines 253-255:** Please change accordingly: "Apparently this **was** is related **explained by** to an already **a** moist spring season (Fig. S2) and the **a** heavy precipitation event **that occurred** occurring in June 2021 (Mohr 255 et al., 2023), leading to a soil moisture content close to field capacity (**top right** middle panel of Fig. S1)."

**Lines 291-294:** "Coupling hot spots are observed over East and Southeast Europe with ACI values of more than 250 J kg-1 occurring in connection with neutral or positive soil moisture anomalies in 2006, 2019, 2020, and 2021 (Fig. 8) which is connected to higher values of LH over these regions due to neutral or positive root zone soil moisture anomalies (Fig. 5)". Sounds repetitive. Please rewrite the sentence more clearly.

**Line 295-296:** There's a typo on the following sentence: "Over Germany and France, mostly only 14 weak coupling is seen with stronger signals during **e2003** and 2019"

**Lines 305-306:** There's a typo on the following sentence: "Over Central Europe the LCL deficit is comparatively small with values of up to 300 m, unlike the years **2003and** 2022 which show strong positive values". Also, strong positive values are also observed for the summer of 2015...

**Lines 306-307:** Change accordingly: "These are the summers with a pronounced negative soil moisture anomaly and a strong positive temperature anomaly of more than 3°C (Fig. 2 **and Fig.5**)"

**Line 319-320:** "At the same time, the high SH (not shown) leads to an increase of the PBL height and thus a higher LCL deficit as shown in Fig. 9" Considering that LCL deficit is defined as the difference between HLCL and PBLH and assuming that the HLCL was high during these summers and I'm no seeing how an increase in PBLH leads to an amplification of the LCL deficit. Can you please clarify this?

**Lines 321-324:** "During summer 2021, which showed record high temperatures over Europe, Central Europe shows a positive soil moisture anomaly (Fig. 5) connected to weak or negative coupling between η and LH (Fig. 6). This means that LH shows little variations and thus lowering HLCL (Wei et al., 2021) which is also reflected in a neutral LCL deficit Fig. 9)." This sentence is very confused, partially because it's written with a poor English syntax. An extra effort by the Authors is required in order to expose their ideas and the argumentation more clearly.

**Line 331:** TCI η-LH should be defined in the data and methods sections. See the major comment n. º 3

**Line 334-335:** "These regions are usually water-limited thus leading to limited evapotranspiration further reducing LH." Once again poor English writing quality… Also what limited evaporation means in the context of a water-limited regime? A scenario with a limited evaporation could also been seen in an energy-limited regime. I'm not sure I fully understood what the authors mean to say here. This goes in line the major comment n. º 2

**Line 356:** "(…) contrast to the cold and wet years 1997 and 2002 (Figs. S6, S7), the LCL deficit (not shown) is mostly positive" Not shown? The LCL deficit is represented in Figure 9, right?

**Line 364-365:** I understand what the authors are trying to say here, but they need to improve the writing quality… "A study of Denissen et al. (2020) found that LSMs tend to overestimate the critical soil moisture and **thus evaporation becomes soil moisture limited too early**."

**Line 380**: "This paper describes the variability of the LA coupling strength of the warm summer seasons 1991-2022 (…)". This is not entirely true. See the last major comment.

**Lines 389-394:** "According to Rousi et al. (2022) the frequency of the occurrence of heat waves has been accelerating over Europe 390 in the last 30-40 years where the large-scale

circulation pattern often features mid- and upper troposphere blocking situations leading to a split of the jet stream towards the Arctic and the Mediterranean. As the jet stream is an important feature for the European weather, it can also alter the near surface flow conditions in West and Central Europe (Laurila et al., 2021) while in other regions like the Mediterranean and East Europe, soil moisture preconditioning is more important as the impact of the jet stream becomes weaker (Prodhomme et al., 2022)." This text section looks a bit out of context in this summary chapter where the idea is to expose objectively and summarize the main outcomes from the analysis. Please consider to move it to the discussion chapter.

**Line 400-402:** An analysis based on 9 separate summer seasons is different from an interannual analysis. The authors need to reformulate this sentence. This goes in line with my last major comment

**Lines 401-404:** "Hot and dry conditions shift the terrestrial coupling to the moisture-limited regime, push the sensitivity of the HLCL on low LH, and through this switch gears to strongly positive LCL deficits which decreases the likelihood for locally triggered deep convection in this region". Super confuse

**Line 404-405:** "The increasing frequency of warm and dry years toward the second half of the study period hints toward a trend of extended periods of **moisture-limitations for evapotranspiration**" I get what the authors want to say but what is "moisture-limitations for evapotranspiration". Please rephrase it and try to adopt a more proper scientific language by using the right scientific terms to describe what you want to say. This goes in line with major comment n. º 2.

**References**

Beck HE, Pan M, Miralles DG, et al (2021) Evaluation of 18 satellite- and model-based soil moisture products using in situ measurements from 826 sensors. Hydrol Earth Syst Sci 25:17–40. https://doi.org/10.5194/hess-25-17-2021

Muñoz-Sabater J, Dutra E, Agust\'\i-Panareda A, et al (2021) ERA5-Land: a state-of-the-art global reanalysis dataset for land applications. Earth Syst Sci Data 13:4349–4383. https://doi.org/10.5194/essd-13-4349-2021

---

## Referee Report (RR3)

I'm pleased to note that the authors have made a great effort to improve some sections of the manuscript following my suggestions. The scientific quality and interest of the analysis has been improved and the conceptualization of the research problem refined. I'm pleased to see that they have included an interannual analysis that complements the results that were already in the manuscript. The metrics are now better described in the data and methods section and the Figures have also been improved.

However, the narrative is still very confusing. It's not only the poor English writing that represents a barrier for an effective and enjoyable reading process. The manuscript text lacks a fluid narrative, a structured reasoning and a solid storyline. Many sections throughout the manuscript appear completely out of context. Redundant statements are often found and the reader finds himself lost trying to follow the narrative several times. There isn't a proper connection between the results and conclusions and so, in the end, it's difficult to wrap up all things and to get an overall picture of the analysis. Unfortunately, these are all aspects that were pointed out in my last review.

Another major concern is related with the manuscript's size. An effort should be made to release the narrative from analysis and discussions that are not strictly necessary to fulfil the objectives. They can be either removed or moved to the supplementary material. More details about this issue are provided in my minor comments below.

My last major comment relates to the poor structuring of some chapters of the manuscript. For instance, the Abstract covers in a very superficial way the goals, the main results and the relevance of the paper. More details about the motivation and the importance of the analysis given the results obtained is absolutely essential. The second part of the Introduction chapter is supported in isolated paragraphs with a fragile connection between them. Some sentences are extremely confusing and written with a poor scientific language. The Discussion chapter presents a muddled narrative that fails to highlight the main findings of the analysis and their implications. Similarly to Introduction, it's not easy to find a connection between paragraphs. An effort must be made to reformulate all these chapters and to adopt a clear, fluid, convincing, objective and logical speech.

The authors mentioned that the text was proofread by a native English speaker, but many writing problems are still there. Considering all this, I think the manuscript is still not ready for publication. I've made a great effort to highlight all these problems in the following minor comments and so, I think it's fair to demand a last effort from the authors. The manuscript text must be deeply reformulated following the above describe suggestions and considering the minor issues described below. Otherwise, I will reject it for publication.

**Minor Comments:**

**Line 10**: Change accordingly: "(...) intensity of these events,  its influence is typically (...)"

**Line 13:** Consider the following suggestion: "(...) between 1991-2022 over Central Europe ."

**Lines 28-31:** "This was also shown by Rousi et al. (2023) and Dirmeyer et al.(2021) for 2018, who suggest that these extreme conditions will be more likely under climate change conditions during

2020-2049 where two out of three summer seasons will experience hot and dry conditions in a +1.5°C warmer world which is already the case" The sentence is too long.

**Lines 33-34:** "(e.g., planetary boundary layer (PBL) height, convective available potential energy (CAPE), lifted condensation level (LCL)" a closing bracket is missing.

**Lines 53-54:** Change accordingly: "(…) simulations for the  European summer seasons **between 1989 and 2008.**

**Lines 59-60:** Change accordingly: "They identified a  hot spot region for the surface coupling **between** sensible and latent heat fluxes and **between** latent heat flux and 2m temperature in South Europe . **A** transition zone is present over larger parts of Central Europe".

**Line 64:65:** "While there was only little sensitivity over the northern part of this area, Central Europe and the British Isles showed a change in the coupling regime based on the convective triggering potential and low level humidity index (CTP-HIlow)". This needs to be better explained.

**Lines 81-82:** "The analysis of Dirmeyer et al. (2021) for the 2018 European heatwave revealed enhanced soil moisture – near-surface feedback coupling under drought conditions". What is exactly a "soil moisture–near-surface" coupling?

**Lines 86-89:** "According to Ossó et al. (2022), Europe already faced an increase in climate extremes since 2000 and will remain a hot spot for severe droughts (Huebener et al., 2017; van der Wiel et al., 2022) impacting not only summer's crop yields (Toreti et al., 2022) but also affecting the generation of renewable energy". Out of context. This paragraph is focused on discussing the soil moisture-temperature coupling. Try to find a better way to fit this information in the Introduction.

**Lines 91-93:** "Shifts in the hydrological conditions from energy- to moisture-limited conditions originating from droughts and heatwaves (Dirmeyer et al., 2021; Duan et al., 2020) or severe flooding (Lo et al., 2021) imply temporal variability in LA coupling at sub-seasonal to interannual time -scales." What are these "hydrological conditions"? An effort must be made to make use of the right concepts.

**Lines94-96:** "Additionally, the critical soil moisture thresholds (Dirmeyer et al., 95 2021; Rousi et al., 2023) suggest not only an intensification of the heat and drought conditions by LA coupling over Europe but also a strengthening of the coupling itself" I'm not getting what you're trying so say here. Please clarify.

**Lines 101-102:** "However, a quantification of the temporal variability in different coupling relationships and the associated impacts of the variability still lack, as LA coupling strength on other time scales than climate periods has been barely investigated over Central Europe so far. The same applies to shifts between coupling regimes due to variability in the climatic conditions." Try to be more explicit here and to use a proper scientific language.

**Lines 125:** Remove the word "framework".

**Lines 130-131:** Change accordingly: "For our analysis, we used volumetric root zone soil moisture η, defined as weighted sum of the soil moisture in the top three soil layers of ERA5 , LH and SH, CAPE, and PBL height (PBLH).

**Lines 133-135:** "As HLCL was not available from ERA5, we used the approach from Georgakakos and Bras (1984) and Bolton (1980) which is based on surface pressure, 2m 135 temperature, and 2m dewpoint to derive HLCL which is also applied in Dirmeyer et al.." This needs to be rewritten.

**Lines 145-149:** Be more precise and objective. There's no need to give all these details. It only brings more confusion.

**Line 158:** Change accordingly: "(…) The summer seasons **of** 2003 and 2022 are (…)" This works for all the other parts of the manuscript where this issue occurs.

**Line 161-162:** "A trend towards larger dewpoint depression is also observed here since 2015". Visually I'm not sure about this. Also, you must have in your hands some results to prove this. I suggest the authors to compute the linear trends before getting these conclusions. This also applies for soil moisture and temperature.

**Line 166:** Remove "which will become more likely in the near future (Huebener et al., 2017; Rousi et al., 2022)". This sounds more like a discussion of your results.

**Lines 168-170:** Why do you start the analysis of Figure 2 with panel b)? I suggest changing the order of panel a) with panel b) if you want to keep the text as it is right now.

**Lines 172-173:** Change accordingly: "The median of TCIη-LH (Fig. 2a) shows higher values for the warm**er** summer seasons (see Fig. 1b)"

**Line 176:** "However, during the warm and dry **years a trend** of ACILH-HLCL approaching values around or below zero is evident." The word "trend" assumes a long-term changing pattern… Here you're just saying that for specific periods the ACILH-HLCL reaches low values…

**Figure 2 caption:** "The bold-faced numbers indicate the fraction of grid cells exceeding the 75th percentile of the respective index". The numbers that are not in bold refer to what?

**Lines 187-188:** "For the ACILH-CAPE (Fig. 2d) no clear trend for an increase or decrease can be observed which could give a hint that also the large-scale weather pattern can play a reasonable role in this case." I get what you're trying to say here but this needs to be explained more clearly.

**Lines 180-183:** Change accordingly: "(…) Based on the interannual variabilities shown in Figs 1 and 2, we therefore decided to focus on summer seasons which have a median 2m temperature anomaly of more than 0.5°C **(Table 1)**.

**Table 1:** These are annual anomalies or anomalies just for the summer periods?

**Lines 186-187:** "combined with a reduced atmospheric water availability" How do the authors know this?

**Lines 188-190:** Change accordingly: "Although the median 2m temperature anomaly for summer 2020 was only 0.4 °C, it **was** considered in our analysis **considering that it**  was the only summer **since 2015 witnessing a**  positive precipitation **anomaly according to both ERA5 and E-OBS datasets (Table 1)**

**Line 191:** Consider changing to: "3.2 Meteorological characterization of warm and dry summers".

**Lines 192-195:** Remove this text section.

**Line 196:** I would remove the analysis of the 500hPa Geopotential. The goal of this paper is to characterize the land-atmosphere interactions and not the anomalous circulation patterns associated to droughts and heatwaves. It only makes sense to keep this if a strong connection with the land-atmosphere processes is made (which is not the case). The contribution of these results for the overall analysis is residual. If you want to keep this, I suggest moving it to supplementary material. This would also contribute to a slight and welcome reduction of the manuscript size.

**Line 196:** It is the geopotential **height.**

**Figure 4 caption:** "The top left panel shows the mean summer 2m temperature 1991-2020 from ERA5". Rewrite this please.

**Line 224-225:** "with a median precipitation anomaly between -34 mm and -63 mm". This is a spatial median right? If so, It needs to be explicit.

**Lines 225-226:** I would move the E_OBS anomalies to supplementary Material. It only makes sense to keep in the main body text the results obtained using ERA5 considering that the LA coupling metrics were only computed using data from this reanalysis product. It's a matter of keeping some coherence. Of course, it's always nice to have results from E_OBS but they should only be used as a complement to prove that ERA5 follows quite well observations.

**Line 245:** Remove the word "amount".

**Lines 246-248:** Super confusing.

**Figure 7 caption:** Similarly to Figure 4 caption rewrite the following sentence please: "The top left panel denotes the summer mean root zone soil moisture 1991-2020 from ERA5"

**Line 259:** "spatial patterns and the spatial extent of warm or cool as well as moist or dry anomalies" Rewrite this please

**Lines 260-261:** "Firstly, 2003, 2015, 2018, 2019, and 2022 stand out the most" It reads weird

**Lines 261-262:** "They are characterized by large (**warm?)** temperature anomalies, dry anomalies in soil moisture and precipitation extent over most of the land areas in our study domain". The dry anomalies in SM and precipitation are regarding the absolute values or the spatial extent? It is not clear.

**Lines 270-272:** "A positive $TCI_{\eta\text{-}LH}$ denotes that LH is limited by the root zone soil moisture and the soil moisture variation results in LH variation while a negative $TCI_{\eta\text{-}LH}$ indicates that the

development of LH is energy limited, i.e., the incoming energy determines the LH development"
Very long sentence. Break in two pls.

**Lines 275-276:** Change accordingly: (…) **the analysis was**  base**d** on daytime means computed for the period 06 UTC and 18 UTC of each day (Yin et al., 2023)

**Lines 277-278:** Change accordingly: "Figure 8 shows the **mean spatial pattern of** TCI η-LH **observed for the previously selected**  warm **and dry** summer seasons.

**Lines 282-284:** The authors need to explain this better.

**Lines 300-301:** "This is related to the anomalously warm and dry conditions in the atmosphere and a soil moisture deficit during these" **(…???).**

**Lines 301-303:** "The SH increases due to a reduction of the evaporative cooling effect at the surface, and the consequent increase in the temperature gradient between land surface and atmosphere". Please structure your reasoning better.

**Lines 306-307:** "In 2017, the spring season showed a positive soil moisture anomaly over Germany, East Europe and the British Isles which is reflected in the strong correlation over these regions. What are the supplementary figures and the manuscript figure showing this?

**Line 315:** Remove the following: "building a bridge toward convective processes".

**Line 318-324:** Read carefully this section. I think there's an incorrect use of the acronyms LCL and HLCL.

**Line 327:** "(…) Simultaneously, the LCL deficit is negative (Fig. 12) leading (…). Should be Fig. 11 right?"

**Lines 331-332:** "This is the area in the study domain facing considerable interannual variability, which is reflected in sign changes, among other things". Explain this better. What are the "among other things"?

**Line 336:** (…) negative values in the ACILH-HLCL (Fig 11a, e, f, i)". Should be Figure 10 right?

**Line 341:** This is also shown by the negative values of the TLCIη-LH-HLCL. What is the TLCIη-LH-HLCL? I would suggest to remove this from the paper. Your results and already self-explanatory and the manuscript is super extensive.

**Line 342-348:** "Please note that the SH is always positively correlated with the PBLH over land and doesn't experience strong interannual variability (not shown). This implies that a strong increase in the SH due to the LH limitation causes strong PBL heating and growth. This in turn pushes both the PBLH and the HLCL upward. Due to the combination of strengthened PBL heating and decreased PBL moistening the HLCL rises further, which leads to an intensification of the LCL deficit and thus inhibiting deep moist convection (Santanello et al., 2011). The areas with the strongest changes in the signal converge with the regions experiencing the strongest warm and dry anomalies (compare Fig. 3j, Fig. 5j, and Fig. 7j)." Remove this. It only brings more confusion.

**Lines 357-370:** Please rewrite all this section. In fact, I would suggest a deep reformulation of all this chapter. The reader finds himself lost frequently and it's super hard to follow a logic narrative or a solid storyline. The authors really need to structure better their ideas and made a real effort to expose them in a more effective and organized way. There's a total confusion of Figures, metrics that are not defined and you're continuously switching the region or the period under discussion. This is not the right way to interpretate the analysis. Also, and as I suggest previously, there's a lot of supplementary material that honestly, I think it's unnecessary and it only brings more complexity.

**Line 373:** Change accordingly: "This index aims  **to assessing** the(…)"

**Lines 381-383:** Please remove the following sentence: "CAPE depends on the atmospheric humidity which is, among others, related to LH while LH is related to the atmospheric temperature, humidity, soil moisture and LAI."

**Lines 387-388:** "Together with a temperature gradient of up to 30 °C or more in the Mediterranean between 850 hPa and 500 hPa (not shown), this leads to stronger atmospheric instability and thus reduced coupling to LH." However, you mention that these regions are defined by large evaporation rates… Thus, a correlation with LH should be visible, right?

**Lines 394-395:** "Over Germany and France, mostly only weak coupling is seen with stronger signals during 2003 and 2019". Poor English

**Lines 395-405:** Rewrite all this section please. There's a total mixture of concepts, physical relations, etc. In fact, it sounds a bit out of context… An effort should be made to better connect this information with the link between LH and CAPE.

**Line 408:** remove the following sentence: "We now discuss the key findings".

**Line 409-415:** Please rewrite all this paragraph. Again, try to expose your arguments more clearly and in a logical way.

**Line 417 and 421:** No trend was discussed or presented in Fig.2 Please use another word to describe your point of view. This follows one of my previous comments.

**Lines 25-426:** "atmospheric stratification which is not only impacted by the surface conditions but also by the large-scale weather pattern and atmospheric stratification". Rewrite this pls.

**Line 424-426:** Rephrase it.

**Line 435-437:** These precipitation deficits could only be **partially** explained by changes in the soil moisture-evaporation regimes and soil moisture-precipitation coupling. Only a small fraction of precipitation results from local moisture recycling processes. The other fraction is explained by moisture convergence and transport of water vapour from remote regions. Also, this is something observed over a long-term period of for some specific years/summers? Please clarify this.

**Line 339-341:** How can the authors get this conclusion from what is said in the previous sentences? Is this supported by any kind of climate projections?

**Line 450:** "These regions are usually water-limited leading to limited evapotranspiration thus further reducing LH". In addition to the poor writing quality, I'm not seeing how a water-limited regime leads necessarily to lower LH….

**Line 454-455:** "Though not yet represented in the model, in reality, this results in a low LAI which is often the case in South Europe" I got lost here.

**Line 471-475:** Rewrite all this please.

**Lines 489-510:** All this sounds out of context and extremely confusing. Most of this information doesn't add anything relevant. The authors really need to reformulate this section and to find a better way to fit this information in the context of the analysis.

**Line 516-518:** "Firstly, the interannual variability between all years of the period was examined in the context of prevailing temperature and moisture anomalies in the light of a warming climate and a projected increase in hot and dry periods until 2100". Rewrite

---

## Referee Report (RR4)

I'm pleased to see that the authors have made a great effort to improve the manuscript in several aspects. The structure of the analysis is now more solid, objective and concise. The reader can now follow the entire narrative without getting lost with unnecessary, repetitive and confusing considerations and results. The manuscript finally follows a solid and robust storyline. The overall quality of the writting has also been improved, althought there are still some minor issues that need to be worked on (please see my minor comments). I suggest the authors to do a thorough review of the text, as there are still several typos and many sections that need to be written more clearly.

My final comment goes for the last two chapters (Discussion and Summary). I suggest merging the two sections into one. Some results discussed in the "Summary" chapter are not even mentioned in the "Discussion" and vice-versa. The ideas and the reasoning are much better organised in the "Summary" than in the "Discussion" chapter. I suggest putting togheter the considerations made in the two chapters and to basically wrap up the analysis following the formula used in the "Summary", adding some other observations that were included in the Discussion. As you will see in my following comments, many parts of the "Discussion" are still unclear. I think the key message here is to simplify, to be more objective and to find a more effective way of sharing the main outcomes of the analysis and to pass the message for the reader.

**Minor Comments:**

**Line 51:** "Soil moisture is essential (…)"

**Line 52:** Please change accordingly: "According to Osso et al. (2002) Europe has **been experiencing** an increase (…)".

**Line 61-62:** I would remove the following sentence. It sounds a bit vague… "Regions exhibiting strong LA coupling coincide with those previously identified through various coupling metrics (…)".

**Line 72-76:** I suggest to break this text section into two smalller sentences. "The analysis conducted (…) in Europe since 1979 (Becker et al., 2022)".

**Lines 119-124:** This sentence is too long. Please try to split it into two.

**Line 125:** Change accordingly: On average, LH in ERA5 tends to be overestimated by about 9Wm-2.

**Line 126:** These correlations were obtained in respect to which dataset? Also, a reference needs to be included here.

**Line 128-129:** Please change accordingly: in  **previous works.**

**Line 134:** There's an extra comma before the word "Additionally"

**Line 141:** Change accordingly: "The LCL deficit (m) is defined as **the** height difference (…)".

**Line 153:** Change accordingly: "(…)  **corresponds to the** (…)".

**Lines 177-179:** No need to describe so extensively what figure 1 shows. The figure caption already contains all this information.

**Lines 181-182:** Please rewrite this sentence: "Previously, there was a stronger interannual variability with mostly more than 50% of the grid cells with positive soil moisture anomalies". It reads weird.

**Lines 182-183:** Change accordingly: " Since 2015, positive temperature anomalies have been observed over more than 75% of the grid cells. Before 2015, only the years of 1994, 2003, 2006 and 2012 were characterized by having more than 50% of the grid cells covered by positive temperature anomalies.

**Lines 186-187:** "With the exception of 2016, the proportion of positive anomalies is more than 50%, while, as with the previous temperature, apart from 1994, 2003, 2006 and 2012, at least 50% of the grid cells show negative anomalies.", Again, it reads weird.

**Lines 188-189:** Can you please clarify this? I'm not getting what you're trying to say here…

**Lines 191:** Change accordingly. The evaporative demand of the atmosphere increases with higher temperatures resulting in a further reduction of soil moisture and an enhanced dewpoint depression. This relation pattern has been observed in the recent summer seasons, particularly after 2015.

**Line 193:** What do you exactly mean by "anomaly spread"? It is not clear.

**Lines 214-216:** I understand what you're trying to say here, but this needs to be written in a clear way. Pls rewrite.

**Line 217:** Change accordingly: "(…) with medians showing a short variation over time (…)".

**Line 218-221:** Again, I understand what you're trying to say here, but the authors need to find another way to describe and explain these results.

**Line 229:** Remove: "(…) and, thus, to an agricultural drought".

**Line 230:** Change accordingly: "(…) it was included in our analysis considering (…)"

**Line 237:** Change accordingly: "(…) During **the** summer **of** 2006, the 2m temperature  **were** highest north (…)". Please keep the same verb tense while describing the results. This applies to the entire manuscript.

**Figure 3 caption:** Change accordingly: "(…) The top left panel shows the mean summer 2m temperatures computed for the period between 1991 and 2020."

**Line 248:** Change accordingly: "**The year of** of 2006 (…)". This applies to the entire manuscript.

**Line 250:** Change accordingly: "(…) associated with **a** warm temperature**s** and (…)".

**Lines 268-269:** Rewrite the following sentence with an appropriate scientific description of the resuls. "The reason is the (...) of the summer".

**Line 359:** Change accordingly: "(...) and thus **a stronger negative** LCL deficit (...)"

**Line 370-371:** I got lost here: "In summer 2006, 2015, and 2017 the ACILH-HLCL is positive over large parts of Central Europe indicating that LH variations drive the evolution of HLCL". As you mentioned previously and correctly a potential physical relation betwen land surface and atmopshre only occurs when ACILH-HLCL is negative, right? Could you please clarify?

**Lines 372-373:** Please divide this sentence into two, as follows: "This implies that LH either has little variations or is high compared to ther summers seasons. This lead to a HLCL decrease and, ultimately to a residual LCL deficit over Central Europe as shown in Figure 9.

**Line 387-394:** This text section sounds a bit out of context... It doesn't provide any useful information for the LH-CAPE coupling.

**Lines 429-433:** Please consider changing this text section to the following: (...) In agreement with Jach et al. (2022), the Southeat/East Europe and the Baltic states were found to be regions marked by a strong $ACI_{LH-CAPE}$ coupling. However, when analysing the interannual variability of $ACI_{LH-CAPE}$, a weak connection is observed between this coupling mechanims and temperature and humidity conditions, suggesting that such variability might be driven by other atmospheric processes.

**Lines 436-437:** Change accordingly: "(...) despite higher temperature**,** strong LA coupling is largely limited to **South Europe**  as seen in the summer of 2021 (...)".

**Lines 437-438:** This reads weird: "This matches with the finding of Guo and Dirmeyer (2013), that areas with normally wet climate can experience a shift in coupling regimes under dry conditions".

**Lines 446-451:** This section of the text needs to be rewritten. Authors should find a better way to link all the sentences creating a more objective a clear narrative. The ideas and the reasoning are disconnected. The English writing quality should also be improved.

**Lines 451-454:** Please remove the following text section. Your results show nothing for the future: "(...) The increased frequency (...) in 2021 for instance".

**Lines 455-461:** Improve the English writing.

**Lines 462-472:** Following my comment above, this text section is very confusing. This needs to explained in a clear way. Try to simplify your message and to create a solid and objective narrative.

**Lines 481-482:** "This led to a stronger westerly flow air which allows for more humid air masses from the Atlantic". Again, try to keep the same verb tense. To correct this problem, the entire manuscript should be carefully proofread.

**Lines 519-521:** "In wet years, LH does not depend on the soil moisture availability as sufficient transpiration of the leaves is possible and the HLCL is not primarily controlled by the lack of

moisture at the surface". This is not correct. Under energy-limited conditions LH is controlled by the amount radiative energy. If you're referrring to something different, please clarify. Be carefull... this sentence might lead to a wrong interpretation. Clarify.

---

## Author Response (AR2)

**General overview**

The authors presented a valuable analysis focused on the influence of soil moisture on the land surface energy partitioning throughout Europe. They went a step further by providing an analysis of the consequences promoted by soil dry-out on the dynamics of the atmosphere, revealing a potential re-amplification of the LA feedbacks through the suppression of convection and further soil desiccation. In fact, there's a lack of studies performing an aggregation of several case studies, allowing for a detailed comparison of these LA feedbacks between different episodes characterized by the co-occurrence of extreme hot and dry conditions. This allows to get a time and spatial overview of the differences in the LA dynamics between episodes and I felt, particularly in the discussion chapter, that the authors missed the opportunity to underline and to explore more deeply the results in this context.

I think that a more detailed spatiotemporal integrated analysis under a climate change context is missing in the analysis (see major comments). It would be interesting to add some information about historical long-term changes in these LA feedbacks. The analysis was carried out using state of the art datasets and appropriate metrics. Thus, in terms of data/methods and the general conceptualization of the research problem I'm confident in the interest and robustness of the results presented here. As highlighted on the several following comments, my main concerns are focused on the poor quality of the English writing over some sections of the manuscript. The text has several typos and the English syntax is far from being ideal for a high-quality scientific publication. In addition, the author's argumentation is often poorly structured and presented in a confusing way. Therefore, a great effort should be made in order to improve the way authors expose their ideas and communicate with the reader. Moreover, the manuscript text often contains acronyms, abbreviations, scientific terms or metrics that are not properly defined in the data and methods sections. With this being said, I think the analysis has potential to be published, but only after authors have correctly addressed the following comments. This is my first round of reviews as I would like to check the manuscript once again to filter out other minor issues.

Thank you for carefully evaluating our manuscript. We really appreciate your suggestions to further improve our manuscript. Further a native English speaker proofread the manuscript.

Please find our detailed responses to your comments marked in blue below.

**Major Comments**

- My first major comment concerns the way the manuscript is structured. The section n. º 3 named as "Summer Season anomaly maps", presents the anomaly fields for several land surface and meteorological variables. These are, in fact, results that came from the analyses and they represent outcomes obtained by the authors. This section of the manuscript should therefore be included in the Results chapter. There's no reason to define a whole new chapter to present these findings.

  We followed your suggestion and incorporated the anomaly maps into the results chapter 3.2.

- Throughout the manuscript, the argumentation, interpretation and discussion of the results is presented in a very vague an unclear way. An effort must be made in order to use a more appropriate scientific language and to adopt the correct scientific terms used in the literature to describe the processes and metrics under

  consideration by the authors. More details about this aspect can be found in the minor comments below.

  We carefully went through all your minor comments you mentioned below. We kindly ask you to refer to our response to each of your minor comments below.

- Throughout the manuscript several abbreviations and acronyms used by the authors are not defined in the Material and Methods section (e.g. TCI $\eta$-LH; CTP- $HI_{low;}$ TLCI). In fact, this section of the manuscript needs some adjustments to define more clearly the several metrics adopted in the analysis. For example, the ACI was computed using two different approaches: one using CAPE and the other using HLCL. However, the way the authors distinguish both throughout the manuscript text is far from being the ideal and it brings some confusion to the narrative. I suggest similarly to TCI, to present the ACI in its mathematical equation form, defining two new abbreviations to the different approaches: $ACI_{CAPE}$ for the first, and $ACI_{HLCL}$ for the second.

  Thank you for your valuable suggestion. The TCI between root zone soil moisture and latent heat flux (LH) is now called "$TCI_{\eta\text{-}LH}$", the ACI between LH and CAPE is now called "$ACI_{LH\text{-}CAPE}$", and the ACI between LH and the height of the lifted condensation level is now called "$ACI_{LH\text{-}HLCL}$". The mathematical equations of $ACI_{LH\text{-}CAPE}$ and $ACI_{LH\text{-}HLCL}$ have been added to section 2.2.

- As I'm sure the authors know, the term LA feedbacks addresses several processes between the land surface and the atmosphere that describe the connection between soil moisture and precipitation, evaporation, temperature and even other meteorological parameters. The analysis is focused on the inter-link between soil moisture and temperature through perturbations in the turbulent fluxes of latent and sensible heat. Thus, I'm not entirely confident on the use of such a generic term to describe such a particular process. I suggest considering another term such as "soil moisture–temperature coupling" and keep land–atmosphere feedback for the link between soil moisture and CAPE or the link between soil moisture and HLCL.

  Our study does not analyze *soil moisture–temperature coupling* but the impact of soil moisture variability on atmospheric stability expressed by CAPE, HLCL and LCL via latent and sensible heat flux variability. We agree that by our applied metrics only land-atmosphere coupling is quantified, because it is only unidirectional and not the back and forth coupling between variables. Recent publications of, e.g., Seo et al. (2024) and Tak et al. (2024) appear to prefer the term "coupling" unless there is direct feedback between these variables.

Following this convention the TCI and ACI are coupling indices, because they combine
variables to describe the impact of the variability of one variable on the other. We
revised our manuscript accordingly.

• Have the authors considered using ERA5-Land or even GLEAM instead of ERA5 to
obtain soil moisture data? ERA5-Land and GLEAM are forced by meteorological
fields provided by ERA5 and so a potential problem associated with some
inconsistency in the data source would be avoided. In addition, ERA5- Land and
GLEAM incorporate land surface models capable to improve the representation of
the water and energy cycles over land, contributing to a better simulation of land
surface variables (Muñoz-Sabater et al. 2021). Beck et al. (2021) evaluated the
temporal dynamics of 18 state-of-the-art (quasi-)global near- surface soil moisture
products, and concluded the following: "The ERA5 reanalysis, which assimilates
ASCAT soil moisture (Hersbach et al., 2020), obtained a lower overall performance
(median R = 0.68) than the open-loop models ERA5-Land (median R = 0.72) and
HBV-ERA5 (median R = 0.74), which were both forced with ERA5 precipitation (Fig.
2a). This suggests that assimilating satellite soil moisture estimates (ERA5) was less
beneficial than either increasing the model resolution (ERA5-Land) or improving the
model efficiency (HBV)."

GLEAM and ERA5-land products were originally considered for this study, but their
use was rejected because they did not contain the necessary variables for the
analysis. ERA5-land and GLEAM do not provide diurnal cycles of high-resolution
vertical profiles of humidity and temperature or variables that characterize the
atmospheric boundary layer e.g., planetary boundary layer height (which is
required for the LCL deficit calculation) or CAPE, but these are required for our
analysis of land-atmosphere coupling.
Using ERA5-land and/or GLEAM in our study would lead to a mixture of data
sources and thus would prevent a seamless investigation.

We decided to include the following to the data and methods section of our
manuscript on page 3, line 114:

"Although a study of Beck et al. (2021) revealed that ERA5-Land (Muñoz-Sabater et
al., 2021) outperformed ERA5 with respect to in-situ soil moisture measurements
in the Carpathians and Southeast France during 2015-2019, data sets developed
solely for land surface studies like ERA5-land and the Global Land Evaporation
Amsterdam Model (GLEAM; Miralles et al., 2011) lack atmospheric boundary layer
variables required for studying land-atmosphere coupling and therefore were not
considered in this study to avoid mixing different models for the investigation of
the coupling chain."

• The Figures already present an overall good quality. However, some extra adjustments would be welcome. The size of the panels could be increased a bit
more by removing the latitude and longitude ticks that are repeated unnecessarily in
all panels (I would only keep them in the first panel). A larger font size would also be
a good idea, especially for the panel title.

We followed your suggestion and increased the font size of the panel title and the
size of the subfigures in each panel wherever possible.

• Finally, I'm not sure if the authors, with results here presented, have successfully
achieved the main objectives stated for the study: "this study investigates interannual
variability of LA coupling strength"; "In this study, we therefore assess the temporal
variability of LA coupling of the European summer seasons 1991- 2022 on the
interannual time scale"; "This paper describes the variability of the LA coupling
strength of the warm summer seasons 1991-2022 which became the dominant
situation over Europe since 2010". Basically, the authors repeatedly stressed that the
main goal of this study is to provide an interannual characterization of the LA
coupling variability. However, in practice, what they presented here is an analysis
focused on nine separated case study periods from a total of 32 years of data. This is
not an interannual analysis even more when a time aggregation of these 9 years is
lacking in the discussion. A narrative going, chronologically, throughout these 9
summer periods highlighting the effects of climate change, is not presented by the
authors, which could be interesting. I think the authors have two options here: either
they reformulate clearly the objectives of the analysis or they include a pure
interannual analysis with a year-by-year evolution of the LA coupling and associated
meteorological variables. The second option would be much more interesting, as it
would also allow to get a temporal integrated overview of these parameters and see
any possible trends in the soil moisture–temperature regimes throughout Europe and
under a climate change context.

In the first review iteration, both reviewers strongly suggested to focus only on
selected summer seasons as the manuscript would have become far too long.
Therefore, we decided to focus only on the summer seasons which show a median
temperature anomaly of more than +0.5 °C associated with a dry bias in precipitation
(with E-OBS as a reference).
However, we followed your suggestion and added timeseries of soil moisture and
temperature anomalies for all summer seasons 1991-2022. In addition, we also show
timeseries of the coupling indices to show interannual variabilities between the
summer seasons 1991-2020. This supports our choice to investigate the nine most
extreme summer seasons in more detail. The discussion and the summary sections
have been adjusted accordingly.

**Minor Comments**

**Lines 23-24:** Please change accordingly: "In the last decades, Europe experienced severe
drought periods and heatwaves (WMO, 2015; C3S, 2018; Markonis et al., 2021; WMO,

2022a)  with 2022 being the hottest summer ever recorded over Europe (WMO,
2022a)."

This has been corrected accordingly.

**Lines 29-30:** "(…) who suggest that these extreme conditions will be more likely under
climate change conditions where two out of three summer seasons will experience hot and
dry conditions." This sounds a bit vague… This increase in hot and dry conditions under
climate change conditions is estimated to occur for which period? Near future, far future? Is
already happening? Authors should clarify.

We changed the sentence accordingly. It now reads (line 27):

"This was also shown by Rousi et al. (2023) and Dirmeyer et al. (2021) for 2018, who suggest
that these extreme conditions will be more likely under climate change conditions during
2020-2049 where two out of three summer seasons will experience hot and dry conditions
in a +1.5°C warmer world which is already the case." (page 1, line 30)

**Line 30:** Please change accordingly: "(…) midlatitudes due to the occurrence of **a** double- jet
stream **configuration associated to atmospheric blocking conditions**
(Kornhuber et al., 2017).

This sentence has been changed according to your suggestion.

**Line 56:** What is the CTP-HI$_{low}$ framework? More information should be given here.

We added the following sentence to the introduction to explain the CTP-HI$_{low}$ framework
(lines 55-61):

"While there was only little sensitivity over the northern part of this area, Central Europe and
the British Isles showed a change in the coupling regime based on the convective triggering
potential and low-level humidity index (CTP-HIlow) framework (Findell and Eltahir, 2003a,
2003b). The combination of CTP and HIlow allows for a determination whether convection is
likely to occur (see Fig. 15 of Findell and Eltahir, 2003a). Jach et al. (2022) performed climate
change sensitivity tests using the CTP-HIlow framework. They found that Central Europe is in
a transition zone where the development of convection is more likely to be solely controlled
by a temperature increase."

**Line 63:** Change accordingly: "soil moisture-temperature feedback was
, a key (…)".

This sentence has been shortened according to your suggestion.

**Lines 76-78:** It reads weird: "The in the preceding paragraph described shifts in the
hydrological conditions from energy- to moisture-limited conditions originating from
droughts and heatwaves (Dirmeyer et al., 2021; Duan et al., 2020) or severe flooding (Lo et
al., 2021) imply temporal variability in LA coupling at sub-seasonal to interannual time-
scales." Please rephrase it.

This sentence has been shortened. It now reads (lines 80-82):

"Shifts in the hydrological conditions from energy- to moisture-limited conditions originating from droughts and heatwaves (Dirmeyer et al., 2021; Duan et al., 2020) or severe flooding (Lo et al., 2021) imply temporal variability in LA coupling at sub-seasonal to interannual time scales.

**Lines 82-84:** Please rephrase it to something like "However, a quantification of **the** temporal variability in different coupling relationships **and the associated impacts** as well as understanding of the impact of the variability remain **still lacksing**, as LA coupling strength **on other times scales than climate period** was **has been** barely investigated over Europe, and particularly on other time scales than climate periods, so far".

This sentence has been replaced by your suggestion. It now reads (lines 87-90):

"However, a quantification of the temporal variability in different coupling relationships and the associated impacts of the variability still lack, as LA coupling strength on other time scales than climate periods has been barely investigated over Europe so far."

**Lines 162-165:** The sentence is too long and confuse. Consider changing to something like: "As shown in Table 1 the warm and dry summer seasons **have** becaome **predominant the prevaileding situation** since 2015**. This has been** associated with a strong reduction in annual and seasonal precipitation, combined with a reduced atmospheric water availability that led to a constant decline of the root zone soil moisture and**, thus, to** an agricultural drought which was the case, e.g., in 2018-2020 over Europe (van der Wiel et al., 2022)."

We followed your suggestion and changed the sentences accordingly. It now reads (lines 185-188):

"As seen from Fig. 1 and Table 1, the warm and dry summer seasons have become predominant since 2015. This has been associated with a strong reduction in annual and seasonal precipitation, combined with a reduced atmospheric water availability that led to a constant decline of the root zone soil moisture and, thus, to an agricultural drought."

**Lines 166-169:** It reads weird. Consider changing to: "The following sections **present an analysis of the anomaly fields of** describe the characteristics of the summer seasons chosen for evaluation (Table 1) with respect to ERA5 500 hPa geopotential, 2-m temperature**,** root zone soil moisture η, **and** as well as observed and ERA5 simulated precipitation **for the summer seasons chosen for evaluation (Table1)**.

Thank you for your suggestion. The paragraph in section 3.2 now reads:

"This subchapter describes the synoptic conditions during each of the previously selected summers. The conditions comprise the 500 hPa geopotential, which informs about the large-scale weather pattern, the 2m temperature anomaly, the precipitation anomaly and the root zone soil moisture anomaly. A more detailed characterization of the summers will be used for the interpretation of the coupling indices later.

**Line 175:** Change accordingly: "(…) However, the summer **seasons of** 2015 and 2020 are exceptions (…)". However, Figure 1 clearly shows, in contrast with 2020, that most of Europe was covered by exceptional high values of 500hPa geopotential during 2015. Can you please clarify this? Authors also wrote the following: "In 2015, a pronounced north-south anomaly gradient is visible with negative values over the British Isles and Scandinavia while in 2020 the 500 hPa geopotential is only slightly above the average 1991-2020". This is true, but a similar pattern is also observed during 2017… Why authors did not mention 2017 and highlight the similarities with 2015? Finally for both 2015 and 2017 summers, I'm not sure if a north-south anomaly gradient is the most accurate way to describe the geopotential anomalies… I would prefer to describe them as concentric nucleus of positive anomalies located over Central Europe associated to a strong meridional gradient of 500hPa Geopotential.

Thank you for your suggestion. We rewrote the whole paragraph of section 3.2.1. It now reads:
"Figure 3 shows the 500 hPa geopotential height anomalies for the selected summer seasons. The 500 hPa geopotential height helps to determine mid-tropospheric troughs and ridges describing the large-scale weather pattern. Most of the investigated summer seasons are characterized by positive 500 hPa geopotential anomalies over large parts of Central Europe. The summer seasons 2003, 2019, and 2022 were characterized by a centric positive anomaly over central Europe with 2022 showing the highest positive anomalies of the investigated summer seasons. The summer seasons 2006 and 2017 were characterized by a meridional anomaly gradient around 50°N. In summer 2006, positive anomalies were present over the British Isles and South Scandinavia while in 2017, positive geopotential anomalies were observed over South Europe. Summer 2018 was characterized by strong positive anomalies north of 50°N and summer 2015 shows a moderate positive centric geopotential anomaly over Central Europe. During summer 2020, the 500 hPa geopotential shows a very weak zonal anomaly gradient so that it can be considered as an average summer compared with the climatology 1991-2020 (Fig. 3a). Summer 2021 was characterized by weak geopotential anomaly gradients while a higher anomaly was present over the British Isles."

**Line 177:** "(…) while in 2020 the 500 hPa geopotential is only slightly above the average 1991-2020 (**bottom right panel in Fig. 1**)". The bottom right panel in Fig.1 shows the anomaly pattern during 2021… In order to avoid this and to help establish a better link between the text and the figure, I recommend the authors to label with letters (a,b,c,d…) the several panels. This works for all the other figures in the manuscript.

We followed your suggestion and added subfigure labels in all panel plots including the supplement.

**Line 182-183:** Change accordingly: "t**T**he highest 2-m temperature anomalies were  **observed during the summers of** 2003, 2018, 2019, and 2022 (Fig. 2)  **and were spatially** associated with strong positive geopotential anomalies over Central Europe. **The summer of** 2022 was the hottest ever recorded so far (C3S, 2023)".

Thank you for your suggestion. We rewrote the whole paragraph in section 3.2.2. It now reads:

"The positive 500 hPa geopotential anomalies shown in Fig. 3 are associated with positive 2m temperature anomalies. The highest 2m temperature anomalies were observed during the summers 2003, 2018, 2019, and 2022 (Fig. 4b, f, g, j) and were spatially associated with strong positive geopotential anomalies over Central Europe. During summer 2006, the 2m temperature anomalies are highest north of 51°N while during the summer seasons 2015 and 2017, the highest temperature anomalies were observed south of 50°N. This coincides with the fact that maximum positive geopotential anomaly is observed south of 51°N (Fig. 3d, e). Summer 2020 shows positive temperature anomalies over a wide area of our study domain. However, the 500 hPa anomalies were very moderate indicating a constant flow of cooler and moist airmasses from the West to Central Europe. Summer 2021 showed a west-east anomaly gradient with temperatures slightly below the climatology over the western part of our investigation domain."

**Line 183 and 185:** This comment works for all the other sections of the manuscript where similar issues are observed. "**2022** was the hottest summer ever recorded so far (C3S, 2023). During 2006, the 2-m temperature anomalies **are** highest north of 51°N 185 while in 2017, the highest temperature anomalies **were** observed south of 50°N as the maximum geopotential anomaly is shifted to the north and south, respectively". It's not correct to start a sentence with 4 numeric digits… It would be more proper to start with something like "The summer of 2022…". Keep in mind that the analysis is focused only for the summer seasons, so make sure when analyzing and discussing the results that you're referring to the summer periods. Also, two different verbal tenses are used in the same sentence, which is not correct. In fact, the authors should adopt, when describing the results, the same verbal tense.

Thank you for your suggestion. As we indeed only focus on the summer seasons, we will make it clear throughout the manuscript. Further a native English speaker now checked the English to ensure language issues are solved.

**Lines 201-203:** This sounds too vague… More information about the way these correlations were obtained (time and space dimensions) and what they exactly mean should be provided by the authors.

We reformulated this paragraph and moved it to the discussion section on page 21, lines 450-460:

"As enough incoming solar energy is present in these regions, this further enhances SH and thus could further intensify drought periods (positive coupling). Together with the positive $TCI_{n\text{-}LH}$ the anticorrelation of SH-LH points to a strong limitation of evapotranspiration by insufficient root zone soil moisture. Though not yet represented in the model, in reality, this results in a low LAI which is often the case in South Europe (see Fig. S6c, d). Moisture-limitation of the LH in the warm and dry summers leads to a shift in the energy flux partitioning towards reduced PBL moistening and amplified PBL heating because of increased SH. This shift causes a drying throughout the PBL, which is shown by an increased HLCL (Fig. S5) and an intensified negative LCL deficit. Thus, the dry and warm conditions at the land surface propagate through the atmosphere and feed back in less favorable conditions for local convection."

**Lines 209-210:** It would be interesting to represent in an extra panel the time series of the daily spatially averaged values of soil moisture over Europe from the early spring until the late summer time for all the summer seasons. It would allow to better catch in the results this effect that the authors are mentioning here. This new panel could be included not just here in Figure 5, but also in all the other figures.

We decided to include a timeseries plot (Fig. 1) of soil moisture, 2m temperature and dewpoint depression anomalies to the new section 3.1. This figure nicely explains our decision to investigate only particular summer seasons. The following paragraph was added to the manuscript on page 5, lines 157-166:

"From the anomaly timeseries in Fig. 1a it is seen that from 2015 onwards the soil moisture content shows a tendency to decrease during summer except for 2016. The summer seasons 2003 and 2022 are the driest summer seasons since 1991. At the same time, a trend for a temperature increase of 0.5-1°C is observed from Fig. 1b since 2015.

Dewpoint depression anomalies (Fig. 1c) can be used as an indicator for the inhibition of cloud formation. A trend towards larger dewpoint depression is also observed here since 2015. As higher temperatures increase the evaporative demand of the atmosphere, this results in a further reduction of soil moisture and thus an enhanced dewpoint depression which is seen among the summer seasons after 2015 in Fig. 1. The anomaly spread of $\eta$ and 2m temperatures does not increase during these years pointing towards a general warming and drying over our region of interest which will become more likely in the near future (Huebener et al., 2017; Rousi et al., 2022)."

**Lines 212-214:** "By using the median of the soil moisture anomalies, 2006 largely is an average summer with moderate positive anomalies over East Europe while 2015 and 2017 on average show moderate dry soil anomalies". Poor English syntax. Please rephrase it

We reformulated the complete paragraph of the new subsection 3.2.4 for a better readability. It now reads:

"Figure 7 displays the ERA5 derived root zone soil moisture anomalies. The summer seasons 2003, 2018, and 2022 show the lowest root zone soil moisture availability over Germany, Benelux, and France. This relates to the strong positive temperature bias and the precipitation dry bias shown both by E-OBS and ERA5. An evaluation of the median of the soil moisture anomalies over Central Europe revealed that summer 2006 is an average summer with moderate positive anomalies over East Europe. The negative soil moisture anomaly during summer 2015 is related to missing precipitation over large parts of Central Europe. Summer 2017 shows a strong positive soil moisture anomaly over North Germany and North

Poland related to the higher-than-average rainfall amount (see Figs. 5 and 6). Interestingly,
although summer 2019 was among of the warmest and driest summers, the soil moisture dry
bias is less pronounced as in the other three hot and dry summer seasons 2003, 2018, and
2022 related to a higher soil moisture content during spring (Fig. S2f). Summer 2020 shows
drier than average soils over France and Germany while soil moisture in the other regions is
around or even above the climatological average. The summer season 2021 shows strong
positive soil moisture anomalies over Benelux and Germany which was related to colder than
average April and May 2021 (C3S, 2022) as well as due to the Ahr flood event (Mohr et al.,
2023)."

**Lines 253-255:** Please change accordingly: "Apparently this **was**  **explained by**
**a** moist spring season (Fig. S2) and  **a** heavy precipitation event **that occurred**
in June 2021 (Mohr 255 et al., 2023), leading to a soil moisture content close to
field capacity (**top right**  panel of Fig. S1)."

This has been changed according to your suggestion.

**Lines 291-294:** "Coupling hot spots are observed over East and Southeast Europe with ACI
values of more than 250 J kg-1 occurring in connection with neutral or positive soil moisture
anomalies in 2006, 2019, 2020, and 2021 (Fig. 8) which is connected to higher values of LH
over these regions due to neutral or positive root zone soil moisture anomalies (Fig. 5)".
Sounds repetitive. Please rewrite the sentence more clearly.

We modified the sentence a bit and it now reads (page 19, line 390):

"Coupling hot spots are observed over East and Southeast Europe with $ACI_{LH-CAPE}$ values of
more than 250 J kg$^{-1}$ in summer 2006, 2019, 2020, and 2021 (Fig. 12). They are related to
higher values of LH over these regions (not shown) due to neutral or positive root zone soil
moisture anomalies (Fig. 7)."

**Line 295-296:** There's a typo on the following sentence: "Over Germany and France, mostly
only 14 weak coupling is seen with stronger signals during **e2003** and 2019"

This typo is corrected.

**Lines 305-306:** There's a typo on the following sentence: "Over Central Europe the LCL
deficit is comparatively small with values of up to 300 m, unlike the years **2003and** 2022
which show strong positive values". Also, strong positive values are also observed for the
summer of 2015...

A blank was added here.

**Lines 306-307:** Change accordingly: "These are the summers with a pronounced negative soil
moisture anomaly and a strong positive temperature anomaly of more than 3°C (Fig. 2 **and**
**Fig.5**)"

As this section was completely rewritten, this sentence is no longer present.

**Line 319-320:** "At the same time, the high SH (not shown) leads to an increase of the PBL height and thus a higher LCL deficit as shown in Fig. 9" Considering that LCL deficit is defined as the difference between HLCL and PBLH and assuming that the HLCL was high during these summers and I'm no seeing how an increase in PBLH leads to an amplification of the LCL deficit. Can you please clarify this?

Thanks for pointing this out. Indeed, our explanation is not correct here. Therefore, this sentence starting line 359 has been changed to:

"This indicates that the very dry soil during these summers (Fig. 7) caused the low LH which in turn initiated a considerable increase of the HLCL (Fig. S5) and thus a higher LCL deficit as shown in Fig. 12. This is also shown by the negative values of the $TLCI_{\eta-LH-HLCL}$ (Fig. S3) showing feedback between $\eta$, LH and HLCL while only weak feedback between $\eta$, LH, and CAPE is present (Fig. S4)"

**Lines 321-324:** "During summer 2021, which showed record high temperatures over Europe, Central Europe shows a positive soil moisture anomaly (Fig. 5) connected to weak or negative coupling between $\eta$ and LH (Fig. 6). This means that LH shows little variations and thus lowering HLCL (Wei et al., 2021) which is also reflected in a neutral LCL deficit Fig. 9)." This sentence is very confused, partially because it's written with a poor English syntax. An extra effort by the Authors is required in order to expose their ideas and the argumentation more clearly.

We made this sentence clearer. It now reads in line 364:
"During summer 2021, the positive soil moisture anomaly (Fig. 7) is connected to weak or negative coupling between $\eta$ and LH (Fig. 8). This implies that LH either has little variations or is high compared to other summer seasons and thus lowering HLCL (not shown, e.g., Wei et al., 2021) which is also reflected in a mostly neutral LCL deficit over Central Europe as shown in Fig. 11."

**Line 331:** TCI $\eta$-LH should be defined in the data and methods sections. See the major comment n. º 3

We followed your major comment #3 to improve the readability of our manuscript with respect to the applied coupling indices.

**Line 334-335:** "These regions are usually water-limited thus leading to limited evapotranspiration further reducing LH." Once again poor English writing quality… Also what limited evaporation means in the context of a water-limited regime? A scenario with a limited evaporation could also been seen in an energy-limited regime. I'm not sure I fully understood what the authors mean to say here. This goes in line the major comment n. º 2

We replaced this sentence by a paragraph starting one page 21, line 442:

"The coupling signals remain stable throughout the summer seasons over North Europe and the Mediterranean region (Seneviratne et al., 2006; Knist et al., 2017; Jach et al., 2020; Jach et al., 2022). The correlation between SH and LH is mainly positive over the British Isles, indicating that evapotranspiration is limited by the incoming energy (Knist et al., 2017) which is also the case over France, Benelux, and Germany for summer 2021 (not shown)."

**Line 356:** "(…) contrast to the cold and wet years 1997 and 2002 (Figs. S6, S7), the LCL deficit (not shown) is mostly positive" Not shown? The LCL deficit is represented in Figure 9, right?

Thank you for spotting this. Indeed, the sentence is a bit confusing. The sentence on page 14, line 306 is now rewritten, and it now reads:

"In 2017, the spring season showed a positive soil moisture anomaly over Germany, East Europe and the British Isles which is reflected in the strong correlation over these regions. The correlation pattern for summer 2021 is similar as during the cold and wet seasons 1997 or 2002 (not shown) where enough soil moisture is available for evapotranspiration."

**Line 364-365:** I understand what the authors are trying to say here, but they need to improve the writing quality… "A study of Denissen et al. (2020) found that LSMs tend to overestimate the critical soil moisture and **thus evaporation becomes soil moisture limited too early**."

Thank you for your valuable suggestion. As we rewrote almost the complete discussion chapter, the sentence on page 22, line 498 now reads:

"On the other hand, Denissen et al. (2020) found that LSMs tend to overestimate the critical soil moisture (Hsu and Dirmeyer, 2023)."

**Line 380**: "This paper describes the variability of the LA coupling strength of the warm summer seasons 1991-2022 (…)". This is not entirely true. See the last major comment.

Indeed, we only focus on nine selected summer seasons between 2003 and 2022. As we restructured large parts of the manuscript, this sentence does not exist anymore.

**Lines 389-394:** "According to Rousi et al. (2022) the frequency of the occurrence of heat waves has been accelerating over Europe 390 in the last 30-40 years where the large-scale circulation pattern often features mid- and upper troposphere blocking situations leading to a split of the jet stream towards the Arctic and the Mediterranean. As the jet stream is an important feature for the European weather, it can also alter the near surface flow conditions in West and Central Europe (Laurila et al., 2021) while in other regions like the Mediterranean and East Europe, soil moisture preconditioning is more important as the impact of the jet stream becomes weaker (Prodhomme et al., 2022)." This text section looks a bit out of context in this summary chapter where the idea is to expose objectively and summarize the main outcomes from the analysis. Please consider to move it to the discussion chapter.

Following your suggestion, we moved this paragraph to the discussion section on page 21, line 465.

**Line 400-402:** An analysis based on 9 separate summer seasons is different from an interannual analysis. The authors need to reformulate this sentence. This goes in line with my last major comment

Following your earlier suggestion, we added time series of 2m temperature, soil moisture and the coupling indices used in our study. The timeseries shows that the nine summer seasons we investigated in depth have a different pattern than the other summer seasons and that these nine summer seasons show a trend for a behavior which is more likely in the (near) future.

We modified the paragraph in the summary, and it now reads on page 23, line 539:

"The interannual variability of the summer seasons revealed a temperature increase which is accompanied by a decline in soil moisture and an increased in the dewpoint depression which is most prominent in the especially warm and dry summers 2003, 2015, 2018, 2019, and 2022.

The warm and dry conditions lead to an intensification or even the onset of statistically measurable coupling in the various processes along the LoCo process chain. In wet years, LH does not depend on the soil moisture availability as sufficient transpiration of the leaves is possible (see Fig. S5d) and also the HLCL is not primarily controlled by the lack of moisture at the surface."

**Lines 401-404:** "Hot and dry conditions shift the terrestrial coupling to the moisture- limited regime, push the sensitivity of the HLCL on low LH, and through this switch gears to strongly positive LCL deficits which decreases the likelihood for locally triggered deep convection in this region". Super confuse

This connection between LH and HLCL is now discussed in the discussion section in more detail and is therefore removed from the summary.

**Line 404-405:** "The increasing frequency of warm and dry years toward the second half of the study period hints toward a trend of extended periods of **moisture-limitations for evapotranspiration**" I get what the authors want to say but what is "moisture-limitations for evapotranspiration". Please rephrase it and try to adopt a more proper scientific language by using the right scientific terms to describe what you want to say. This goes in line with major comment n. º 2.

We reformulated this paragraph at the end of the summary starting line 546. It now reads:

"The increasing frequency of warm and dry summers from 2015 onwards hints toward a trend of extended periods of reduced soil moisture available- for evapotranspiration and the likelihood of locally triggered convection. This leads to a growing influence of soil moisture variability on the meteorological conditions which was not as pronounced before 2003 due to cooler and moister conditions. Markonis et al. (2021) found a considerable increase in drought events over Central Europe since 2010 which they relate to increasing temperature and a lack of rainfall which together cause a soil moisture depletion due to
excessive evapotranspiration."

# References

Beck HE, Pan M, Miralles DG, et al (2021) Evaluation of 18 satellite- and model-based soil
moisture products using in situ measurements from 826 sensors. Hydrol Earth Syst Sci
25:17–40. https://doi.org/10.5194/hess-25-17-2021

Muñoz-Sabater J, Dutra E, Agust\'\i-Panareda A, et al (2021) ERA5-Land: a state-of-the- art
global reanalysis dataset for land applications. Earth Syst Sci Data 13:4349– 4383.
https://doi.org/10.5194/essd-13-4349-2021

Seo, E., P. A. Dirmeyer, M. Barlage, H. Wei and M. Ek, 2024: Evaluation of land-atmosphere
coupling processes and climatological bias in the UFS global coupled model. J. Appl.
Meteor. Clim., **25**, 161–175, doi:10.1175/JHM-D-23-0097.1

Tak, S., E. Seo, P. A. Dirmeyer and M.-I. Lee, 2024: The role of soil moisture-temperature
coupling for the 2018 Northern European heatwave in a subseasonal forecast. Weather
and Climate Extremes, 44, https://doi.org/10.1016/j.wace.2024.100670

---

## Author Response (AR3)

I'm pleased to note that the authors have made a great effort to improve some sections of the manuscript following my suggestions. The scientific quality and interest of the analysis has been improved and the conceptualization of the research problem refined. I'm pleased to see that they have included an interannual analysis that complements the results that were already in the manuscript. The metrics are now better described in the data and methods section and the Figures have also been improved.

Dear Reviewer,

Thank you for carefully evaluating our manuscript again. Please find our answers to your comments in blue.

However, the narrative is still very confusing. It's not only the poor English writing that represents a barrier for an effective and enjoyable reading process. The manuscript text lacks a fluid narrative, a structured reasoning and a solid storyline. Many sections throughout the manuscript appear completely out of context. Redundant statements are often found and the reader finds himself lost trying to follow the narrative several times. There isn't a proper connection between the results and conclusions and so, in the end, it's difficult to wrap up all things and to get an overall picture of the analysis. Unfortunately, these are all aspects that were pointed out in my last review.

*Based on this we decided to completely restructure and rewrite larger parts of the manuscript without changing the scientific content instead of inserting and deleting here and there.*

Another major concern is related with the manuscript's size. An effort should be made to release the narrative from analysis and discussions that are not strictly necessary to fulfil the objectives. They can be either removed or moved to the supplementary material. More details about this issue are provided in my minor comments below.

*We followed this suggestion and shortened the introduction by removal of some sentences.*

My last major comment relates to the poor structuring of some chapters of the manuscript. For instance, the Abstract covers in a very superficial way the goals, the main results and the relevance of the paper. More details about the motivation and the importance of the analysis given the results obtained is absolutely essential.

*We rewrote the abstract accordingly*

The second part of the Introduction chapter is supported in isolated paragraphs with a fragile connection between them. Some sentences are extremely confusing and written with a poor scientific language.

*We rewrote the introduction accordingly*

The Discussion chapter presents a muddled narrative that fails to highlight the main findings of the analysis and their implications. Similarly to Introduction, it's not easy to find a connection between paragraphs. An effort must be made to reformulate all these chapters and to adopt a clear, fluid, convincing, objective and logical speech.

*The discussion has been revised accordingly (see also the answers to the comments below.*

The authors mentioned that the text was proofread by a native English speaker, but many writing problems are still there. Considering all this, I think the manuscript is still not ready for publication.

I've made a great effort to highlight all these problems in the following minor comments and so, I think it's fair to demand a last effort from the authors. The manuscript text must be deeply reformulated following the above describe suggestions and considering the minor issues described below. Otherwise, I will reject it for publication.

*We decided to revise the manuscript because only language and structural arguments but no scientific reasons are subject of concern.*

**Minor Comments:**

*Some references to lines belong to the revised manuscript with track mode and some to the manuscript without track mode, which sometimes made it hard to understand the comments and requirements. We tried our best, when no text was cited to find the correct lines.*

**Line 10**: Change accordingly: "(...) intensity of these events,  its influence is typically (...)"

*The whole abstract has been rewritten.*

**Line 13:** Consider the following suggestion: "(...) between 1991-2022 over Central Europe ."

*The whole abstract has been rewritten.*

**Lines 28-31:** "This was also shown by Rousi et al. (2023) and Dirmeyer et al.(2021) for 2018, who suggest that these extreme conditions will be more likely under climate change conditions during 2020-2049 where two out of three summer seasons will experience hot and dry conditions in a +1.5°C warmer world which is already the case" The sentence is too long.

*Changed to "This phenomenon was also observed by Rousi et al. (2023) and Dirmeyer et al. (2021) in relation to the extreme conditions of 2018, suggesting that such events are likely to become more frequent under climate change."*

**Lines 33-34:** "(e.g., planetary boundary layer (PBL) height, convective available potential energy (CAPE), lifted condensation level (LCL)" a closing bracket is missing.

*Changed to " (e.g., planetary boundary layer (PBL) height, convective available potential energy (CAPE), lifted condensation level (LCL)) "*

**Lines 53-54:** Change accordingly: "(...) simulations for the  European summer seasons **between 1989 and 2008.**

*Due to rewriting of the introduction (see above) this sentence is deleted. Knist is now cited only as "Regions exhibiting Strong LA coupling coincide with those previously identified through various coupling metrics (e.g., Koster et al. (2004), Dirmeyer (2011), Guo and Dirmeyer (2013), Knist et al. (2017) and Jach et al. (2022))"*

**Lines 59-60:** Change accordingly: "They identified a  hot spot region for the surface coupling **between** sensible and latent heat fluxes and **between** latent heat flux and 2m temperature in South Europe  **.** **A** transition zone is present over larger parts of Central Europe".

*Due to rewriting of the introduction (see above) this sentence is deleted. Knist is now cited only as "Regions exhibiting Strong LA coupling coincide with those previously identified through various coupling metrics (e.g., Koster et al. (2004), Dirmeyer (2011), Guo and Dirmeyer (2013), Knist et al. (2017) and Jach et al. (2022))"*

*Further the following was added: "Using water isotopes, precipitation, humidity, air temperature, and soil moisture data from 2006 to 2009, Yuan et al. (2023) identified the Central and Eastern Europe region in summer as one of 11 global hotspots for LA coupling, exhibiting varying pathways (e.g., soil moisture-precipitation, soil moisture-evapotranspiration, and soil moisture-temperature) and seasonality of LA coupling strength."*

**Line 64:65:** "While there was only little sensitivity over the northern part of this area, Central Europe and the British Isles showed a change in the coupling regime based on the convective triggering potential and low level humidity index (CTP-HIlow)". This needs to be better explained.

*Due to rewriting of the introduction (see above) this sentence is deleted. The details of the method are not needed in the introduction.*

**Lines 81-82:** "The analysis of Dirmeyer et al. (2021) for the 2018 European heatwave revealed enhanced soil moisture – near-surface feedback coupling under drought conditions". What is exactly a "soil moisture–near-surface" coupling?

*Changed to "The analysis conducted by Dirmeyer et al. (2021) for the 2018 European heatwave revealed enhanced soil moisture-maximum temperature coupling under drought conditions"*

**Lines 86-89:** "According to Ossó et al. (2022), Europe already faced an increase in climate extremes since 2000 and will remain a hot spot for severe droughts (Huebener et al., 2017; van der Wiel et al., 2022) impacting not only summer's crop yields (Toreti et al., 2022) but also affecting the generation of renewable energy". Out of context. This paragraph is focused on discussing the soil moisture-temperature coupling. Try to find a better way to fit this information in the Introduction.

*We moved this text as part of restructuring the introduction*

**Lines 91-93:** "Shifts in the hydrological conditions from energy- to moisture-limited conditions originating from droughts and heatwaves (Dirmeyer et al., 2021; Duan et al., 2020) or severe flooding (Lo et al., 2021) imply temporal variability in LA coupling at sub-seasonal to interannual time -scales." What are these "hydrological conditions"? An effort must be made to make use of the right concepts.

*Rewritten to "The critical soil moisture threshold defines the boundary between energy-limited and water-limited regimes for evapotranspiration. Shifts from energy- to soil moisture-limited conditions due to droughts and heatwaves (Dirmeyer et al., 2021; Duan et al., 2020) or vice versa in the case of severe flooding (Lo et al., 2021) imply temporal variability in LA coupling over sub-seasonal to interannual timescales."*

**Lines94-96:** "Additionally, the critical soil moisture thresholds (Dirmeyer et al., 95 2021; Rousi et al., 2023) suggest not only an intensification of the heat and drought conditions by LA coupling over Europe but also a strengthening of the coupling itself" I'm not getting what you're trying so say here. Please clarify.

*Rewritten together with previous comments. The whole paragraph now reads:*

*"Guo and Dirmeyer (2013) reported interannual variability in soil moisture-precipitation coupling, resulting from differing soil moisture availability. The critical soil moisture threshold defines the boundary between energy-limited and water-limited regimes for evapotranspiration. Shifts from energy- to soil moisture-limited conditions due to droughts and heatwaves (Dirmeyer et al., 2021; Duan et al., 2020) or vice versa in the case of severe flooding (Lo et al., 2021) imply temporal variability in LA coupling over sub-seasonal to interannual timescales. Below these critical soil moisture thresholds, intensification of heat and drought conditions occurs through LA coupling over Europe, alongside a strengthening of the coupling itself. Jach et al. (2022) identified Central Europe as a transition zone where the development of convection appears to be primarily influenced by temperature increases."*

**Lines 101-102:** "However, a quantification of the temporal variability in different coupling relationships and the associated impacts of the variability still lack, as LA coupling strength on other time scales than climate periods has been barely investigated over Central Europe. The same applies to shifts between coupling regimes due to variability in the climatic conditions." Try to be more explicit here and to use a proper scientific language.

*Rewritten to "Despite significant advancements in understanding land-atmosphere (LA) coupling, a crucial aspect of this complex phenomenon remains poorly understood: the temporal variability of LA coupling strength and its associated impacts. Specifically, the investigation of LA coupling across timescales beyond climate periods has been largely neglected in Central Europe, and shifts between coupling regimes driven by variability in climatic conditions remain an ongoing research topic (Barriopedro et al., 2023). To address this knowledge gap, the current study aims to quantify the variability of LA coupling strength over Central Europe during the summer seasons from 1991 to 2022, focusing on the relationships between temperature, soil moisture, precipitation, and large-scale weather patterns. By leveraging high-resolution data from the fifth generation of the European Centre for Medium -Range Weather Forecasting (ECMWF) atmospheric reanalysis (ERA5; Hersbach et al., 2020), this study seeks to provide new insights into the dynamics of LA coupling and its implications for climate extremes, agriculture, and ecosystems in the region. Ultimately, this research aims to enhance our understanding of the complex interactions between the land surface and the atmosphere and to inform the development of more effective strategies for mitigating the impacts of climate change in Central Europe."*

**Lines 125:** Remove the word "framework".

*Done*

**Lines 130-131:** Change accordingly: "For our analysis, we used volumetric root zone soil moisture η, defined as weighted sum of the soil moisture in the top three soil layers of ERA5 , LH and SH, CAPE, and PBL height (PBLH).

*Land surface models have different numbers and thicknesses of soil layers and different root zone depths. This impacts the potential transpiration. Therefore it is necessary to mention the total root zone depth of ERA5 data used here. We change the sentence to: "For our analysis, we used volumetric root zone soil moisture η, defined as weighted sum of the soil moisture in the top three soil layers of ERA5 (i.e. the top 1 meter ), LH and SH, CAPE, and PBL height (PBLH)"*

**Lines 133-135:** "As HLCL was not available from ERA5, we used the approach from Georgakakos and Bras (1984) and Bolton (1980) which is based on surface pressure, 2m 135 temperature, and 2m dewpoint to derive HLCL which is also applied in Dirmeyer et al.." This needs to be rewritten.

*Rewritten to: "Since HLCL was not directly available from ERA5, we applied the approach proposed by Georgakakos and Bras (1984) and Bolton (1980) , which derive HLCL based on surface pressure, 2m temperature, and 2m dew point, a method also employed by Dirmeyer et al. (2014):"*

**Lines 145-149:** Be more precise and objective. There's no need to give all these details. It only brings more confusion.

*It is not clear to us if you are referring to the revised manuscript in track mode :*

==*"To categorize the summer seasons during 1991-2022,this period into warm and wet, warm and dry, and cold summer seasons, seasonal mean anomalies of 2-m2m temperatures and precipitation from ERA5 and as well as precipitation from the ENSEMBLES daily gridded observational dataset for precipitation (E-OBS; Cornes et al., 2018) version V26.0e were calculated."*==

*Or without track mode:*

==*"To derive the strength of the coupling between the land surface and the atmosphere (ACI), the standard deviation of η can, e.g., be substituted by surface fluxes in Eq. 2 while LH in Eq. 2 can be substituted by PBLH or CAPE (Dirmeyer et al., 2014).*==

==*ACIs are computed 1) between LH and CAPE ($ACI_{LH-CAPE}$), and 2) between LH and HLCL ($ACI_{LH-HLCL}$): "*==

*The second paragraph was added due to previous reviewer requests, so we think it is the prior paragraph. We rewrote it and moved it to the new subchapter 2.3: "Seasonal mean anomalies of 2m temperatures and precipitation from ERA5 as well as precipitation from the ENSEMBLES daily gridded observational dataset for precipitation (E-OBS; Cornes et al., 2018) version V26.0e were calculated to categorize the summer seasons in Central Europe between 1991 and 2022 into dry to wet and warm to cold or moderate years. "*

**Line 158:** Change accordingly: "(…) The summer seasons **of** 2003 and 2022 are (…)" This works for all the other parts of the manuscript where this issue occurs.

*Changed throughout the manuscript*

**Line 161-162:** "A trend towards larger dewpoint depression is also observed here since 2015". Visually I'm not sure about this. Also, you must have in your hands some results to prove this. I

suggest the authors to compute the linear trends before getting these conclusions. This also applies for soil moisture and temperature.

*Firstly we added chapter 2.3 and explain what is shown in figures 1 and 2:*

*"The investigation of interannual variability of anomalies in various variables and metrics, including their spatial distribution, involved the calculation of time series of the spatial variability of anomalies as follows. For each land grid cell, the average anomaly for the months of June to August was computed for each year. Box-whisker plots were then utilized to represent the data from all land grid cells, facilitating a comparison of the spatial variability of summer anomalies across different years." Further we rewrote the analyses of figures 1, it now reads:*

*"Figure 1 shows box-whisker plots of the summer mean values of soil moisture, 2m temperature and 2m dew point temperature depression from 1991 to 2022 of the land grid cells in the study area between 40°N and 60°N and between 5°W and 25°E. The anomalies refer to the mean values of the respective grid cells from 1991-2020. Since 2015, apart from 2016 and 2021, more than 75% of the grid cells in the study area show negative soil moisture anomalies (Fig. 1a), in 2021 it is more than 50%. Previously, there was a stronger interannual variability with mostly more than 50% of the grid cells with positive soil moisture anomalies. The temperature anomaly (Fig. 1b) has been positive in more than 75% of the grid cells since 2015, in some cases more than 1 K. Before that, only 1994, 2003, 2006 and 2012 show more than 50% of the grid cells with a positive anomaly; in the other years, more than 75% of the grid cells are usually cooler than the mean value. There has also been a change in the dew point temperature depression since 2015. With the exception of 2016, the proportion of positive anomalies is more than 50%, while, as with the previous temperature, apart from 1994, 2003, 2006 and 2012, at least 50% of the grid cells show negative anomalies. It is also noticeable that the anomalies in at least 50% of the grid cells have spanned the same or a larger range of values since 2015, meaning that the spatial variability of the size of the anomalies is increasing."*

*Further we rewrote the analyses of figures 2, it now reads:*

*Figure 2 shows box-whisker plots of the summer mean values of LA coupling indices from 1991 to 2022 of all land grid cells in the study area between 40°N and 60°N and between 5°W and 25°E. They represent the value range across Europe for each index and summer. Variations between the years denote both interannual variability in the number of grid cells (i.e. spatial extent) with potential for physical coupling, and differences in the strength of the coupling (higher or lower values for the index).*

*The distributions of the TCIη-LH display strong interannual variability in terms of the expansion of the area with potential for physical coupling given as the number in each box. The fraction of land cells with positive TCIη-LH ranges between 0.54 in 2011 and 0.92 in 2022. This points to a variability in the land area with potential for coupling of up to 38%, showing substantial interannual variability in the spatial extent of the coupling region. At the same time, the median of TCIη-LH (Fig. 2a) shows higher values for the warm summer seasons (see Fig. 1b), which implies that also the strength of the coupling increases during these years. During the different summer seasons CORRSH-LH is mostly positive across Europe (Fig. 2b), which means that the LH and SH co-vary. Negative correlations, where the limitation of the LH causes an exaggeration of the SH, mostly occur in the Mediterranean. However, there are few exceptions for the very warm and dry summer seasons of 2003, 2018, 2019, and 2022 where the median of CORRSH-LH drops to less than 0.2 due to less positive correlation coefficients and a larger land area with negative correlations. The interannual variability in ACILH-HLCL (Fig. 2c) is less pronounced than that of the TCIη-LH and CORR. The land area with potential for physical coupling ranges between 5% in the early 1990s and 33% in 2003, where also the median TCI dropped below 100m. However, with exception of 2003, all summers*

*with the largest expansion of the potential coupling region and the lowest median ACI assemble in the warm and dry years of the last decade (bold-numbers inFig. 2c). For the ACILH-CAPE (Fig. 2d) the median index does not show strong interannual variability, but the land area fraction with positive ACILH-CAPE varies between 0.48 and 0.8. Additionally, differences in the interquartile range (height of the boxes) and length of the whiskers suggests larger spatial variability in some years. Generally, the years with the highest median index and the largest potential coupling regions does not resemble with the temperature and humidity conditions as it does for the other indices.*

**Line 166:** Remove "which will become more likely in the near future (Huebener et al., 2017; Rousi et al., 2022)". This sounds more like a discussion of your results.

*Deleted.*

**Lines 168-170:** Why do you start the analysis of Figure 2 with panel b)? I suggest changing the order of panel a) with panel b) if you want to keep the text as it is right now.

*Done*

**Lines 172-173:** Change accordingly: "The median of TCIη-LH (Fig. 2a) shows higher values for the warm**er** summer seasons (see Fig. 1b)"

*Done*

**Line 176:** "However, during the warm and dry **years a trend** of ACILH-HLCL approaching values around or below zero is evident." The word "trend" assumes a long-term changing pattern… Here you're just saying that for specific periods the ACILH-HLCL reaches lowvalues…

*Changed to "However, during the warm and dry years (see Fig. 1b), more grid cells have small or negative $ACI_{LH-HLCL}$, i.e. their LH variability is not or only weakly coupled with the variability of the HLCL."*

**Figure 2 caption:** "The bold-faced numbers indicate the fraction of grid cells exceeding the 75th percentile of the respective index". The numbers that are not in bold refer to what?

*Changed to "The numbers indicate the fraction of land cells in the value range of the index potentially indicating a physical relationship, i.e. $TCI_{\eta-LH} > 0$, $CORR_{SH-LH}$ and $ACI_{LH-HLCL} < 0$ and $ACI_{LH-CAPE} > 0$. Bold-face numbers mark the 8 years (i.e. 25% of the examined years) with the highest share in the period."*

**Lines 187-188:** "For the ACILH-CAPE (Fig. 2d) no clear trend for an increase or decrease can be observed which could give a hint that also the large-scale weather pattern can play a reasonable role in this case." I get what you're trying to say here but this needs to be explained more clearly.

*Changed to "For the $ACI_{LH-CAPE}$ (Fig. 2d) there is also no change in the interannual variability evident, but the strength of $ACI_{LH-CAPE}$ shows a larger spatial variability expressed in a wider range of the $25^{th}$ to $75^{th}$ percentile in each year since 2003. It is worth noting that 2019 shows the largest variability*

*of ACI_LH-CAPE where 78 % of the grid cells exceed the 75th percentile." The large scale circulation is discussion, so deleted here.*

**Lines 180-183:** Change accordingly: "(…) Based on the interannual variabilities shown in Figs 1 and 2, we therefore decided to focus on summer seasons which have a median 2m temperature anomaly of more than 0.5°C **(Table 1)**.

*Done*

**Table 1:** These are annual anomalies or anomalies just for the summer periods?

*Summer, it is changed to "Selected summer seasons based on a positive summer temperature anomaly larger than 0.5°C with respect to the climatological summer mean of 1991-2020."*

**Lines 186-187:** "combined with a reduced atmospheric water availability" How do the authors know this?

*From Fig. 1c, we added this in the text*

**Lines 188-190:** Change accordingly: "Although the median 2m temperature anomaly for summer 2020 was only 0.4 °C, it **was** considered in our analysis **considering that it** was the only summer **since 2015 witnessing a** positive precipitation **anomaly according to both ERA5 and E-OBS datasets (Table 1)**bias since 2015

*Changed*

**Line 191:** Consider changing to: "3.2 Meteorological characterization of warm and dry summers".

*We want to emphasize that we describe the selected summer, so now the heading reads "Meteorological characterization of the selected warm and dry summers"*

**Lines 192-195:** Remove this text section.

*Done*

**Line 196:** I would remove the analysis of the 500hPa Geopotential. The goal of this paper is to characterize the land-atmosphere interactions and not the anomalous circulation patterns associated to droughts and heatwaves. It only makes sense to keep this if a strong connection with the land-atmosphere processes is made (which is not the case). The contribution of these results for the overall analysis is residual. If you want to keep this, I suggest moving it to supplementary material. This would also contribute to a slight and welcome reduction of the manuscript size.

*It is moved to the supplement.*

**Line 196:** It is the geopotential **height.**

*Done*

**Figure 4 caption:** "The top left panel shows the mean summer 2m temperature 1991-2020 from ERA5". Rewrite this please.

*Done*

**Line 224-225:** "with a median precipitation anomaly between -34 mm and -63 mm". This is a spatial median right? If so, It needs to be explicit.

*Yes, "spatial" is added.*

**Lines 225-226:** I would move the E_OBS anomalies to supplementary Material. It only makes sense to keep in the main body text the results obtained using ERA5 considering that the LA coupling metrics were only computed using data from this reanalysis product. It's a matter of keeping some coherence. Of course, it's always nice to have results from E_OBS but they should only be used as a complement to prove that ERA5 follows quite well observations.

*E_OBS is moved to the supplement and the paragraph and also in the soil moisture paragraph is rewritten accordingly.*

**Line 245:** Remove the word "amount".

*Done*

**Lines 246-248:** Super confusing.

*Rewritten: "Interestingly, although summer 2019 was among of the warmest and driest summers, the soil moisture dry anomaly is less pronounced as in the other three hot and dry summer seasons of 2003, 2018, and 2022. The reason is the higher soil moisture content during spring 2019 (Fig. S2f), that was not used by the beginning of the summer."*

**Figure 7 caption:** Similarly to Figure 4 caption rewrite the following sentence please: "The top left panel denotes the summer mean root zone soil moisture 1991-2020 from ERA5"

*Changed, now figure 6*

**Line 259:** "spatial patterns and the spatial extent of warm or cool as well as moist or dry anomalies" Rewrite this please

*Rewritten: "While all years indicated that most of the cells experienced a significant warm anomaly, the spatial patterns and the extent of warm or cool, as well as moist or dry anomalies varied between the years"*

**Lines 260-261:** "Firstly, 2003, 2015, 2018, 2019, and 2022 stand out the most" It reads weird

*Changed to "Firstly, the years that stand out the most are 2003, 2015, 2018, 2019, and 2022."*

**Lines 261-262:** "They are characterized by large (**warm?)** temperature anomalies, dry anomalies in soil moisture and precipitation extent over most of the land areas in our study domain". The dry anomalies in SM and precipitation are regarding the absolute values or the spatial extent? It is not

clear.

*As in temperature. It now reads: "They are characterized by warm temperature anomalies and dry anomalies in soil moisture and precipitation across most of the land areas in our study domain."*

**Lines 270-272:** "A positive TCIη-LH denotes that LH is limited by the root zone soil moisture and the soil moisture variation results in LH variation while a negative TCI η-LH indicates that the development of LH is energy limited, i.e., the incoming energy determines the LH development" Very long sentence. Break in two pls.

*Done*

**Lines 275-276:** Change accordingly: (…) **the analysis was**  base**d** on daytime means computed for the period 06 UTC and 18 UTC of each day (Yin et al., 2023)

*Done*

**Lines 277-278:** Change accordingly: "Figure 8 shows the **mean spatial pattern of** TCI η-LH **observed for the previously selected**  warm **and dry** summer seasons. .

*Done*

**Lines 282-284:** The authors need to explain this better.

*It now reads "In the wettest regions during both years, the index changes its sign. The now neutral to negative values indicate that there is enough soil moisture available (see Fig. 7). This implies that in these areas and during these years, the variations in latent heat (LH) flux are not directly linked to changes in soil moisture (refer to Fig. 6 and Fig. 7)." In the previous paragraph it was already said, why only positive TCLI indicate that the soil moisture is below the critical soil moisture threshold. Otherwise the regime is energy limited. So this here is not repeated.*

**Lines 300-301:** "This is related to the anomalously warm and dry conditions in the atmosphere and a soil moisture deficit during these" **(…???).**

*"years" was added*

**Lines 301-303:** "The SH increases due to a reduction of the evaporative cooling effect at the surface, and the consequent increase in the temperature gradient between land surface and atmosphere". Please structure your reasoning better.

*Now reads: "The soil moisture deficit limits LH and due to the resulting reduction of evaporative cooling SH is further increased. Consequently, the temperature gradient between land surface and atmosphere increases."*

**Lines 306-307:** "In 2017, the spring season showed a positive soil moisture anomaly over Germany, East Europe and the British Isles which is reflected in the strong correlation over these regions. What are the supplementary figures and the manuscript figure showing this?

*The figures are added: "In 2017, the spring season showed a positive soil moisture anomaly over Germany, East Europe and the British Isles (Fig. S4) which is reflected in the strong positive correlations between LH and SH during the summer over these regions (Fig. 7d)."*

**Line 315:** Remove the following: "building a bridge toward convective processes".

*Done*

**Line 318-324:** Read carefully this section. I think there's an incorrect use of the acronyms LCL and HLCL.

*They are used correctly.*

**Line 327:** "(…) Simultaneously, the LCL deficit is negative (Fig. 12) leading (…). Should be Fig. 11 right?"

*Yes, done*

**Lines 331-332:** "This is the area in the study domain facing considerable interannual variability, which is reflected in sign changes, among other things". Explain this better. What are the "among other things"?

*We deleted ", among other things".*

**Line 336:** (…) negative values in the ACILH-HLCL (Fig 11a, e, f, i)". Should be Figure 10 right?

*Yes, done*

**Line 341:** This is also shown by the negative values of the TLCIη-LH-HLCL. What is the TLCIη-LH-HLCL? I would suggest to remove this from the paper. Your results and already self-explanatory and the manuscript is super extensive.

*We followed your suggestion and deleted the correseponding text regarding the TLCI. Also, the two supplementary figures were deleted.*

**Line 342-348:** "Please note that the SH is always positively correlated with the PBLH over land and doesn't experience strong interannual variability (not shown). This implies that a strong increase in the SH due to the LH limitation causes strong PBL heating and growth. This in turn pushes both the PBLH and the HLCL upward. Due to the combination of strengthened PBL heating and decreased PBL moistening the HLCL rises further, which leads to an intensification of the LCL deficit and thus inhibiting deep moist convection (Santanello et al., 2011). The areas with the strongest changes in the signal converge with the regions experiencing the strongest warm and dry anomalies (compare Fig. 3j, Fig. 5j, and Fig. 7j)." Remove this. It only brings more confusion.

*Done*

**Lines 357-370:** Please rewrite all this section. In fact, I would suggest a deep reformulation of all this

chapter. The reader finds himself lost frequently and it's super hard to follow a logic narrative or a solid storyline. The authors really need to structure better their ideas and made a real effort to expose them in a more effective and organized way. There's a total confusion of Figures, metrics that are not defined and you're continuously switching the region or the period under discussion. This is not the right way to interpretate the analysis. Also, and as I suggest previously, there's a lot of supplementary material that honestly, I think it's unnecessary and it only brings more complexity.

*Changed to "Over Germany, France, and Benelux, the $ACI_{LH-HLCL}$ shows low or negative values during the extreme warm and dry summer seasons of 2003, 2018, and 2022 (Fig. 8a, e, i).This indicates that the very dry soil during these summers (Fig. 5) caused low LH which in turn initiated a considerable increase of the HLCL (Fig. S5) and thus a higher LCL deficit as shown in Figure 9.*

*In summer 2006, 2015, and 2017 the $ACI_{LH-HLCL}$ is positive over large parts of Central Europe indicating that LH variations drive the evolution of HLCL. During summer 2021, the positive soil moisture anomaly (Fig. 5) is connected to weak or negative coupling between η and LH (Fig. 6). This implies that LH either has little variations or is high compared to other summer seasons and thus lowering HLCL (not shown, e.g., Wei et al., 2021) which is also reflected in a mostly neutral LCL deficit over Central Europe as shown in Figure 9.*

*As the $TCI_{η-LH}$ is mostly positive over these regions during these summers, while the $ACI_{LH-CAPE}$ is neutral to slightly positive, this indicates that soil moisture variation impacts LH variations but with weak feedback to the atmosphere. "*

**Line 373:** Change accordingly: "This index aims  **to assessing** the(…)"

*Done*

**Lines 381-383:** Please remove the following sentence: "CAPE depends on the atmospheric humidity which is, among others, related to LH while LH is related to the atmospheric temperature, humidity, soil moisture and LAI."

*Done*

**Lines 387-388:** "Together with a temperature gradient of up to 30 °C or more in the Mediterranean between 850 hPa and 500 hPa (not shown), this leads to stronger atmospheric instability and thus reduced coupling to LH." However, you mention that these regions are defined by large evaporation rates… Thus, a correlation with LH should be visible, right?

*Changed to "Together with a temperature gradient of up to 30 °C or more in the Mediterranean between 850 hPa and 500 hPa (not shown), this can leads to a strong atmospheric instability in ERA5 and thus to an overestimation of CAPE in the Mediterranean (Taszarek et al., 2018)."*

**Lines 394-395:** "Over Germany and France, mostly only weak coupling is seen with stronger signals during 2003 and 2019". Poor English

*Changed to "Over Germany and France, coupling is generally weak, although stronger signals were observed in 2003 and 2019."*

**Lines 395-405:** Rewrite all this section please. There's a total mixture of concepts, physical relations, etc. In fact, it sounds a bit out of context… An effort should be made to better connect this information with the link between LH and CAPE.

*We reorganised and rewrote the section as follows: "3.4.2 Coupling LH-CAPE*

[revised manuscript text omitted]

**Line 417 and 421:** No trend was discussed or presented in Fig.2 Please use another word to describe your point of view. This follows one of my previous comments.

*Changed to "However, the last decade shows the largest spatial extent and highest coupling strengths due to more the warm and dry summers (Fig. 2a). "*

**Lines 25-426:** "atmospheric stratification which is not only impacted by the surface conditions but also by the large-scale weather pattern and atmospheric stratification". Rewrite this pls.

*The paragraph including the commented sentence reads now as follows: "The $ACI_{LH-CAPE}$ shows coupling hot-spots over Southeast and East Europe as well as over the Baltic states which coincides with the hotspot observed in Jach et al. (2022) who studied surface fluxes influences on the potential for deep convection triggering. However, the interannual variability of $ACI_{LH-CAPE}$ shows little connection with the temperature and humidity conditions. CAPE results from a complex interplay of atmospheric stratification, synoptic circulation and moistening and heating by the land surface. The results suggest that rather the atmospheric factors drive the interannual variability."*

**Line 424-426:** Rephrase it.

*Same as previous, done*

**Line 435-437:** These precipitation deficits could only be **partially** explained by changes in the soil moisture-evaporation regimes and soil moisture-precipitation coupling. Only a small fraction of precipitation results from local moisture recycling processes. The other fraction is explained by moisture convergence and transport of water vapour from remote regions. Also, this is something observed over a long-term period of for some specific years/summers? Please clarify this.

*In this case, we did not aim to discuss local moisture recycling, but rather a growing occurrence of heatwaves and droughts due to drought-induced warming. Koster et al. (2009) used precipitation-temperature correlations based on observations and global simulations as surrogates for soil moisture and evaporation to analyze this. We revised the paragraph to make this point more clear:*

*"From the interannual variability  of the different variables shown in Figs. 1 and 2, it can be concluded that warm and dry summer seasons are associated with a differing behavior of LA coupling strength across Europe. During summer seasons with enough moisture, despite higher temperatures strong LA coupling is largely limited to the European South as seen in the summer of 2021. This matches with the finding of Guo and Dirmeyer (2013), that areas with normally wet climate can experience a shift in coupling regimes under dry conditions. On the seasonal time-scale, Lo et al. (2021) also found regime shifts due to an extreme flood in a semi-arid region. According to Rousi et al. (2022) the frequency of occurrence of heat waves is accelerating over Europe in the last 30-40 years where the large scale circulation pattern often features mid- and upper troposphere blocking situation leading to a split of the jet stream towards the Arctic and the Mediterranean. As the position of the jet stream has a decisive effect on European weather, it can also alter the near surface flow conditions in West and Central Europe (Laurila et al., 2021) while in other regions like*

*the Mediterranean and East Europe, soil moisture preconditioning is more important as the impact of the jet stream becomes weaker (Prodhomme et al., 2022). Dirmeyer et al. (2021) showed the causal connection between the hot and dry conditions during the 2018 extreme summer. The spring already started with a warm anomaly and slightly drier conditions over Germany (Xoplaki et al., 2023) turning into a severe drought due to a strong soil moisture depletion (Rousi et al., 2023). Dirmeyer et al. (2021) also showed that the drought conditions intensified the 2018 heatwave, because when the volumetric soil moisture content fell below a critical value, surface fluxes and temperatures became highly sensitive to the further declining soil moisture. The concept of drought-induced warming through evaporative controls was also found by Koster et al. (2009)."*

**Line 339-341:** How can the authors get this conclusion from what is said in the previous sentences? Is this supported by any kind of climate projections?

*We revised the paragraph. It discusses our finding of increased coupling strength and spatial extent of the coupling region within hot and dry years in the context of an increased frequency of heat waves as well as mechanisms for the co-occurrence of drought and heat conditions reported in literature. On this basis, the sentence was revised to: "The increased frequency of hot and dry extremes together with our findings suggests that greater coupling strength can occur more often over a larger extent of Europe in the future. Despite the warmer temperatures variations in humidity can cause variability in coupling as seen in 2021 for instance."*

**Line 450:** "These regions are usually water-limited leading to limited evapotranspiration thus further reducing LH". In addition to the poor writing quality, I'm not seeing how a water-limited regime leads necessarily to lower LH….

*Rewritten to "The available net radiation energy is divided between LH and SH according to the energy required for evapotranspiration. LH and SH are correlated as long as evapotranspiration is not limited by the available soil moisture. LH in the regions south of 44 °N (Fig. 8) is usually water-limited. Therefor a common feature of the warm and dry summer seasons is the anticorrelation of LH and SH."*

**Line 454-455:** "Though not yet represented in the model, in reality, this results in a low LAI which is often the case in South Europe" I got lost here.

*Deleted*

**Line 471-475:** Rewrite all this please.

*It now reads: "During warm and humid or moderate summer seasons, the local LA system is characterized by sufficient moisture, which leads to a decoupling in several links along the local coupling (LoCo; Santanello et al., 2018) chain. Specifically, the terrestrial coupling index $TCI_{\eta\text{-}LH}$ is negative, indicating that variations in $\eta$ do not drive LH. Additionally, LH and SH co-vary, suggesting that evapotranspiration is not limited by soil moisture availability."*

**Lines 489-510:** All this sounds out of context and extremely confusing. Most of this information doesn't add anything relevant. The authors really need to reformulate this section and to find a better way to fit this information in the context of the analysis.

*We agree and deleted Lines 489-504. Lines 505-510 were rewritten and moved to the dataset description: "While ERA5 is a robust data set, it has some limitations. LH in ERA5 tend to be*

*overestimated on average by about 9 W m$^{-2}$ (Muñoz-Sabater et al., 2021). ERA5 soil moisture shows reasonable correlations of up to 0.7 over Europe, but may be overestimated on wet days and underestimated on sub-daily precipitation rates. Despite its limitations, ERA5 is a reliable data set for studying LA coupling and has been successfully applied in various studies. Its hourly estimates and high horizontal resolution make it a valuable tool for understanding the complex interactions between the atmosphere and land surface.*"

**Line 516-518:** "Firstly, the interannual variability between all years of the period was examined in the context of prevailing temperature and moisture anomalies in the light of a warming climate and a projected increase in hot and dry periods until 2100". Rewrite

*Changed to "Firstly, the interannual variability of these relationships was examined across all years of the period taking into account the prevailing temperature and moisture anomalies in the context of a warming climate and a projected increase in hot and dry periods until 2100 (Huebener et al., 2017)."*

---

## Author Response (AR4)

I'm pleased to see that the authors have made a great effort to improve the manuscript in several aspects. The structure of the analysis is now more solid, objective and concise. The reader can now follow the entire narrative without getting lost with unnecessary, repetitive and confusing considerations and results. The manuscript finally follows a solid and robust storyline. The overall quality of the writting has also been improved, althought there are still some minor issues that need to be worked on (please see my minor comments). I suggest the authors to do a thorough review of the text, as there are still several typos and many sections that need to be written more clearly.

Dear Reviewer,
Thank you for appreciating our revisions. Please find our answers to your comments in blue.

My final comment goes for the last two chapters (Discussion and Summary). I suggest merging the two sections into one. Some results discussed in the "Summary" chapter are not even mentioned in the "Discussion" and vice-versa. The ideas and the reasoning are much better organised in the "Summary" than in the "Discussion" chapter. I suggest putting togheter the considerations made in the two chapters and to basically wrap up the analysis following the formula used in the "Summary", adding some other observations that were included in the Discussion. As you will see in my following comments, many parts of the "Discussion" are still unclear. I think the key message here is to simplify, to be more objective and to find a more effective way of sharing the main outcomes of the analysis and to pass the message for the reader.

**Minor Comments:**

**Line 51:** "sSoil moisture is essential (...)"

Corrected

**Line 52:** Please change accordingly: "According to Osso et al. (2002) Europe has **been experiencing** an increase (...)".

Corrected

**Line 61-62:** I would remove the following sentence. It sounds a bit vague... "Regions exhibiting strong LA coupling coincide with those previously identified through various coupling metrics (...)".

Deleted.

**Line 72-76:** I suggest to break this text section into two smalller sentences. "The analysis conducted (...) in Europe since 1979 (Becker et al., 2022)".

The sentence was split into two parts. The two sentences now read:

"The analysis conducted by Dirmeyer et al. (2021) for the 2018 European heatwave revealed enhanced soil moisture-maximum temperature coupling under drought conditions, where exceptionally low soil moisture limited evapotranspiration and consequently amplified heatwave conditions due to reduced evaporative cooling (Santanello et al., 2018). This led to one of the most severe heatwaves recorded in Europe since 1979 (Becker et al., 2022)."

**Lines 119-124:** This sentence is too long. Please try to split it into two.

The long sentence is now split into two parts:

"Although a study of Beck et al. (2021) revealed that ERA5-Land (Muñoz-Sabater et al., 2021) outperformed ERA5 with respect to in-situ soil moisture measurements in the Carpathians and Southeast France during 2015-2019, data sets developed solely for land surface studies like ERA5-land and the Global Land Evaporation Amsterdam Model (GLEAM; Miralles et al., 2011) lack atmospheric boundary layer variables required for studying LA coupling. Therefore, ERA5-Land and GLEAM were not considered in this study to avoid mixing different models for the investigation of the coupling chain."

**Line 125:** Change accordingly: On average, LH in ERA5 tends to be overestimated by about 9Wm-2.

Changed.

**Line 126:** These correlations were obtained in respect to which dataset? Also, a reference needs to be included here.

The sentence now reads:

"ERA5 soil moisture shows reasonable correlations of up to 0.7 with in situ-measurements from the International Soil Moisture Network (Dorigo et al., 2021) over Europe, but may be overestimated on wet days and underestimated on sub-daily precipitation rates."

**Line 128-129:** Please change accordingly: in  **previous works.**

Corrected

**Line 134:** There's an extra comma before the word "Additionally"

The comma is deleted.

**Line 141:** Change accordingly: "The LCL deficit (m) is defined as **the** height difference (...)".

Corrected.

**Line 153:** Change accordingly: "(...)  **corresponds to the** (...)".

Corrected.

**Lines 177-179:** No need to describe so extensively what figure 1 shows. The figure caption already contains all this information.

These lines were deleted.

**Lines 181-182:** Please rewrite this sentence: "Previously, there was a stronger interannual variability with mostly more than 50% of the grid cells with positive soil moisture anomalies". It reads weird.

The sentence is reformulated. It now reads:

"Before 2015, there was a stronger interannual variability where often more than 50% of the grid cells have positive soil moisture anomalies."

**Lines 182-183:** Change accordingly: " Since 2015, positive temperature anomalies have been

observed over more than 75% of the grid cells. Before 2015, only the years of 1994, 2003, 2006 and 2012 were characterized by having more than 50% of the grid cells covered by positive temperature anomalies.

Changed according to your suggestion.

**Lines 186-187:** "With the exception of 2016, the proportion of positive anomalies is more than 50%, while, as with the previous temperature, apart from 1994, 2003, 2006 and 2012, at least 50% of the grid cells show negative anomalies.", Again, it reads weird.

We rephrased the sentence. It now reads:

"Except for 2016, the proportion of grid cells with positive dew point depression anomalies is larger than 50% as indicated by the median line inside the boxes. Before 2015, apart from 1994, the hot and dry summer 2003, 2006 and 2012, at least 50% of the grid cells show negative dew point depression anomalies."

**Lines 188-189:** Can you please clarify this? I'm not getting what you're trying to say here…

We reformulated this sentence and split it into two parts. It now reads:

"It is also noticeable that the anomalies have spanned the same or a larger range of values since 2015 as indicated by the upper quartile. This implies that the spatial variability and the magnitude of the anomalies is increasing."

**Lines 191:** Change accordingly. The evaporative demand of the atmosphere increases with higher temperatures resulting in a further reduction of soil moisture and an enhanced dewpoint depression. This relation pattern has been observed in the recent summer seasons, particularly after 2015.

Changed accordingly.

**Line 193:** What do you exactly mean by "anomaly spread"? It is not clear.

We meant the range of the anomalies. Therefore "spread" was replaced by "range".

**Lines 214-216:** I understand what you're trying to say here, but this needs to be written in a clear way. Pls rewrite.

We reformulated this sentence. It now reads:

"The land area with potential for physical coupling ranges between 5% in the early 1990s and 33% in 2003. However, except for 2003, all summers with the largest spatial extent of the potential coupling region and the lowest median $ACI_{LH-HCLC}$ occur in the warm and dry years of the last decade (bold-numbers in Fig. 2c)."

**Line 217:** Change accordingly: "(…) with medians showing a short variation over time (…)".

Corrected.

**Line 218-221:** Again, I understand what you're trying to say here, but the authors need to find another way to describe and explain these results.

The modified sentences now read:

*"However, the land area with potential for coupling (positive ACILH-CAPE) varies between 0.48 and 0.8, showing variability in the spatial extent of the coupling region for this relationship. Unlike the other indices, the greatest coupling strength (represented by the median index) and the largest extent of the coupling region do not occur in the warm and dry years."*

**Line 229:** Remove: "(…) and, thus, to an agricultural drought".

*Removed.*

**Line 230:** Change accordingly: "(…) it was included in our analysis considering (…)"

*Corrected.*

**Line 237:** Change accordingly: "(…) During **the** summer **of** 2006, the 2m temperature  **were** highest north (…)". Please keep the same verb tense while describing the results. This applies to the entire manuscript.

*Corrected.*

**Figure 3 caption:** Change accordingly: "(…) The top left panel shows the mean summer 2m temperatures computed for the period between 1991 and 2020."

*Corrected.*

**Line 248:** Change accordingly: "**The year of** of 2006 (…)". This applies to the entire manuscript.

*Corrected.*

**Line 250:** Change accordingly: "(…) associated with  warm temperature**s** and (…)".

*Corrected.*

**Lines 268-269:** Rewrite the following sentence with an appropriate scientific description of the resuls. "The reason is the (…) of the summer".
*This sentence had an error. It now reads:*

*"The reason for the less pronounced dry anomaly is the higher soil moisture availability during spring 2019 (Fig. S4f)"*

**Line 359:** Change accordingly: "(…) and thus **a stronger negative** LCL deficit (…)"

*Corrected.*

**Line 370-371:** I got lost here: "In summer 2006, 2015, and 2017 the ACILH-HLCL is positive over large parts of Central Europe indicating that LH variations drive the evolution of HLCL". As you mentioned previously and correctly a potential physical relation betwen land surface and atmopshre only occurs when ACILH-HLCL is negative, right? Could you please clarify?

*Yes, that is right. In this case LH variations do not drive the evolution of HLCL. The text has been adjusted.*

**Lines 372-373:** Please divide this sentence into two, as follows: "This implies that LH either has little variations or is high compared to ther summers seasons. This lead to a HLCL decrease and, ultimately to a residual LCL deficit over Central Europe as shown in Figure 9.

*Corrected.*

**Line 387-394:** This text section sounds a bit out of context... It doesn't provide any useful information for the LH-CAPE coupling.

We agree. We decided to delete this text section.

**Lines 429-433:** Please consider changing this text section to the following: (...) In agreement with Jach et al. (2022), the Southeat/East Europe and the Baltic states were found to be regions marked by a strong $ACI_{LH-CAPE}$ coupling. However, when analysing the interannual variability of $ACI_{LH-CAPE}$, a weak connection is observed between this coupling mechanims and temperature and humidity conditions, suggesting that such variability might be driven by other atmospheric processes.

Corrected.

**Lines 436-437:** Change accordingly: "(...) despite higher temperature**,** strong LA coupling is largely limited to **South Europe**  as seen in the summer of 2021 (...)".

Corrected.

**Lines 437-438:** This reads weird: "This matches with the finding of Guo and Dirmeyer (2013), that areas with normally wet climate can experience a shift in coupling regimes under dry conditions".

We reformulated this sentence. It now reads:

"This agrees with the results of Guo and Dirmeyer (2013), who showed that areas with normally wet climate can experience a shift in coupling regimes under dry conditions."

**Lines 446-451:** This section of the text needs to be rewritten. Authors should find a better way to link all the sentences creating a more objective a clear narrative. The ideas and the reasoning are disconnected. The English writing quality should also be improved.

We reformulated this text section. It now reads:

The spring of 2018 showed a warm temperature anomaly and slightly drier soil moisture conditions over Germany (Xoplaki et al., 2023). These turned into a severe drought due to a strong soil moisture depletion during summer (Rousi et al., 2023). Dirmeyer et al. (2021) also showed that in 2018 the drought conditions further intensified the heatwave. The reason is that when the volumetric soil moisture content fell below a critical value, surface fluxes and temperatures became highly sensitive to the further declining soil moisture. This concept of drought-induced warming through evaporative controls was also found by Koster et al. (2009).

**Lines 451-454:** Please remove the following text section. Your results show nothing for the future: "(...) The increased frequency (...) in 2021 for instance".

This section is removed.

**Lines 455-461:** Improve the English writing.

We reformulated this text section. It now reads:

"The coupling signals remain stable throughout the evaluated summer seasons over North Europe and the Mediterranean region (Seneviratne et al., 2006; Knist et al., 2017; Jach et al., 2020; Jach et al., 2022). It is worth to note that the correlation between SH and LH is mainly positive over the British Isles, indicating that evapotranspiration is limited by the incoming energy (Knist et al.,

2017). This is also the case over France, Benelux, and Germany for summer 2021 where a positive soil moisture anomaly was present during spring. Over Central and East Europe changes in the coupling regimes occur between the individual summers as indicated by switches in the sign of different indices. This area coincides with the transition zones observed in the studies of Knist et al. (2017) and Jach et al. (2022)."

**Lines 462-472:** Following my comment above, this text section is very confusing. This needs to explained in a clear way. Try to simplify your message and to create a solid and objective narrative.

The handling editor wrote in his comment, that he does not demand major changes in the discussion section. However, we modified this text section:

"The available net radiation is partitioned between LH and SH according to the energy required for evapotranspiration. LH and SH are correlated as long as evapotranspiration is not limited by the available soil moisture. Our study revealed that LH is often water-limited (reddish colors in Fig. 6) which is associated with an anticorrelation of LH and SH. As enough incoming solar energy is present , this further enhances SH and thus could further intensify drought periods (positive coupling). Together with the positive TCIη-LH, the anticorrelation of SH-LH points to a strong limitation of evapotranspiration by insufficient root zone soil moisture (Fig. 5).

Moisture-limitation of the LH in the warm and dry summers leads to a shift in the energy flux partitioning towards reduced PBL moistening and amplified PBL heating because of increased SH. This shift causes a drying throughout the PBL, which is shown by an increased HLCL (Fig. S5) and an intensified negative LCL deficit (Fig. 9). Thus, the warm and dry conditions at the land surface propagate through the atmosphere leading to less favorable conditions for local convection."

**Lines 481-482:** "This led to a stronger westerly flow air which allows for more humid air masses from the Atlantic". Again, try to keep the same verb tense. To correct this problem, the entire manuscript should be carefully proofread.

Corrected. The final proofread will be performed during the publication process in NHESS.

**Lines 519-521:** "In wet years, LH does not depend on the soil moisture availability as sufficient transpiration of the leaves is possible and the HLCL is not primarily controlled by the lack of moisture at the surface". This is not correct. Under energy-limited conditions LH is controlled by the amount radiative energy. If you're referrring to something different, please clarify. Be carefull... this sentence might lead to a wrong interpretation. Clarify.

We reformulated this text section to avoid misunderstanding. It now reads:

"In wet years, LH is not soil moisture limited, i.e., HLCL is not primarily controlled by the lack of moisture at the surface but by the available energy from radiation."